# Dynamic mitochondrial transcription and translation in B cells control germinal center entry and lymphomagenesis

Yavuz F. Yazicioglu [1], Eros Marin[1], Ciaran Sandhu [1,2], Silvia Galiani[3], Iwan G. A. Raza[2], Mohammad Ali[4,5], Barbara Kronsteiner [4,5], Ewoud B. Compeer[1], Moustafa Attar [1], Susanna J. Dunachie [4,5,6], Michael L. Dustin [1] & Alexander J. Clarke [1]✉

Germinal center (GC) B cells undergo proliferation at very high rates in a hypoxic microenvironment but the cellular processes driving this are incompletely understood. Here we show that the mitochondria of GC B cells are highly dynamic, with significantly upregulated transcription and translation rates associated with the activity of transcription factor A, mitochondrial (TFAM). TFAM, while also necessary for normal B cell development, is required for entry of activated GC precursor B cells into the germinal center reaction; deletion of *Tfam* significantly impairs GC formation, function and output. Loss of TFAM in B cells compromises the actin cytoskeleton and impairs cellular motility of GC B cells in response to chemokine signaling, leading to their spatial disorganization. We show that B cell lymphoma substantially increases mitochondrial translation and that deletion of *Tfam* in B cells is protective against the development of lymphoma in a c-Myc transgenic mouse model. Finally, we show that pharmacological inhibition of mitochondrial transcription and translation inhibits growth of GC-derived human lymphoma cells and induces similar defects in the actin cytoskeleton.

The germinal center (GC) reaction is a highly spatially organized process in secondary lymphoid tissue essential for humoral immunity[1]. B cells responding to antigen captured by follicular dendritic cells introduce random mutations in their immunoglobulin genes in a process known as somatic hypermutation (SHM), which occurs in the anatomically defined dark zone, and are then selected through competitive interaction with follicular helper T cells (T$_{FH}$) in the light zone. GC B cells cycle between these zones, leading to antibody affinity maturation and eventually formation of memory B or plasma cells. During SHM, mutations are introduced in immunoglobulin gene loci through the action of activation-induced cytidine deaminase (AID encoded by *AICDA*). This can give rise to oncogenic mutations, for example, the translocation of *MYC* with the *IGH* or *IGL* loci[2]. GC B cells are the origin of most diffuse large B cell lymphomas (DLBCL), the most common non-Hodgkin lymphoma.

The metabolic processes that support GC B cell homeostasis are incompletely understood. GC B cells are highly proliferative, with division times as short as 4–6 h, and reside within a hypoxic

[1]Kennedy Institute of Rheumatology, University of Oxford, Oxford, UK. [2]Medical Sciences Division, University of Oxford, Oxford, UK. [3]Medical Research Centre Human Immunology Unit, Weatherall Institute of Molecular Medicine, University of Oxford, Oxford, UK. [4]Nuffield Department of Medicine Centre For Global Health Research, Nuffield Department of Clinical Medicine, University of Oxford, Oxford, UK. [5]Mahidol-Oxford Tropical Medicine Research Unit, Mahidol University, Bangkok, Thailand. [6]National Institute for Health and Care Research Oxford Biomedical Research Centre, Oxford University Hospitals NHS Foundation Trust, Oxford, UK. ✉e-mail: alexander.clarke@kennedy.ox.ac.uk

microenvironment[3–5]. Typically, rapidly proliferating immune cells mainly use aerobic glycolysis, but GC B cells rely on fatty acid oxidation (FAO) and oxidative phosphorylation, although to what extent is somewhat controversial[4–10]. This metabolic phenotype is carried over into a substantial proportion of DLBCLs, which are also oxidative phosphorylation-dependent[11]. In vitro, mitochondria have been shown to be regulators of B cell signaling through redox-related mechanisms, although whether these studies can be generalized to the hypoxic microenvironment of the GC found in vivo is uncertain[12–14].

Regulation of mitochondrial transcription and translation is important in other immune cell types, including cytotoxic CD8[+] T cells, where it controls cell killing independent of the effects on metabolism, and cytokine production in CD4[+] T cells[15–17]. A key regulator of mitochondrial transcription and translation is transcription factor A, mitochondrial (TFAM), a DNA-binding high mobility box group protein that aids in the packaging of the mitochondrial genome into nucleoids, analogous to the role of histone proteins, and also controls transcription and translation of mitochondrial DNA, serving as a regulator of mitochondrial biogenesis[18,19]. In T cells, TFAM controls immunosenescence by restraining the production of inflammatory cytokines[20]; in fibroblasts and myeloid cells, it acts to regulate an antiviral immune state[21]. Recent work has shown that the mitochondrial DNA helicase TWINKLE is required for GC formation and plasma cell differentiation[22]. However, the dynamics, function and regulation of mitochondria in GC B cell biology remain incompletely understood.

We show that GC B cell mitochondria are highly dynamic organelles, undergoing profound structural changes as they transition through the GC reaction. We find that TFAM is dynamically regulated in B cells and is required for their development and transcriptional and spatial entry into the GC reaction by modulating cellular motility. We also demonstrate, using genetically modified mouse models, that TFAM is essential for the development of lymphoma and that pharmacological inhibition of mitochondrial transcription and translation in human lymphoma cells is a potential treatment target for human disease.

## Results

### GC B cells undergo active mitochondrial remodeling

To determine mitochondrial density and structure in GC B cells, we first immunized mitochondrial reporter mice expressing green fluorescent protein (GFP) and mCherry (mito-QC)[23] with sheep red blood cells (SRBCs) and examined spleen by confocal microscopy and flow cytometry. The mitochondrial GFP signal was strongly localized to GCs and significantly higher than the surrounding B cell follicle (Fig. 1a,b),

which we confirmed by flow cytometry (Fig. 1c). We found that the mitochondrial GFP signal was highest in the gray zone, followed by the light zone and then the dark zone (Fig. 1c). We identified highly distinct mitochondrial morphology between naive and GC B cells, with small, fragmented mitochondria in naive B cells and large fused mitochondria in GC B cells (Fig. 1d and Extended Data Fig. 1a). The mito-QC reporter mouse also allows detection of autophagy of mitochondria (mitophagy), as mCherry is resistant to quenching in the acid environment of the autolysosome. Using multispectral imaging flow cytometry, we screened GC B cells for the presence of GFP−mCherry+ punctae. Although autophagy has been reported in GC B cells[24], we did not identify evidence of active mitophagy (Extended Data Fig. 1b–d). We hypothesized that for B cells entering the GC reaction to acquire the high mitochondrial mass we observed, activation of mitochondrial transcription and translation would be required. Using Aicda-Cre × Rosa26[STOP]tdTomato reporter mice to label GC B cells, we found markedly increased levels of cytochrome c oxidase subunit 1 (COXI), part of the electron transport chain (ETC) complex IV, in GC B cells (Fig. 1e,f).

To examine mitochondrial transcription, we next labeled B cells in vivo or ex vivo with 5-ethynyl uridine (5-EU), which is incorporated into actively synthesized RNA. We detected colocalization between 5-EU and COXI, with 5-EU incorporation highly prominent in GC B cells and most often seen in areas of high COXI signal (Fig. 1g).

To understand the regulation of this high rate of mitochondrial protein transcription and translation, we examined the expression and distribution of TFAM. We found TFAM protein organized into punctate structures representing mitochondrial nucleoids, which colocalized with 5-EU (Fig. 1g). These were more numerous and larger in GC B cells (Fig. 1h) and had a more elliptical morphology, which is associated with elevated transcriptional activity (Extended Data Fig. 1e)[25]. We confirmed the increase in nucleoid number using stimulated emission depletion (STED) super-resolution microscopy (Fig. 1i). Mitochondria are therefore highly dynamic in GC B cells, with prominent transcriptional activity associated with increased TFAM-nucleoid content.

### *Tfam* is essential during B cell development and differentiation

TFAM has a role in the prevention of long-term inflammatory immunosenescence in T cells but is dispensable for their development[20,26]. Whether TFAM is important for B cell development or function is unknown, as are the dynamics of ETC component expression in B cell developmental trajectories. To address these questions, we first conditionally deleted *Tfam* in B cells using *Cd79a*-Cre (hereafter B-Tfam),

**Fig. 1 | GC B cells undergo active mitochondrial remodeling. a**, Spleen sections from mito-QC mice immunized with SRBC and analyzed on day 12. Scale bar, 50 μm. Representative of three independent experiments. **b**, Quantification of mitochondrial GFP signal intensity and area (μm²) in GCs. Each point represents an individual GC (n = 9) and surrounding B cell follicle (n = 9), pooled from three individual mito-QC mice immunized as in **a**. Representative of two independent experiments. **c**, Gating strategy for GC B cells and their subsets in the spleen. GC and naive B cells were gated as CD19+CD38−GL-7+ and CD19+IgD+GL-7−, respectively. GC B cells from the dark zone, gray zone and light zone were identified based on CXCR4 and CD86 expression signatures as depicted. Quantification and comparison of gMFI of mitochondrial GFP in the indicated subsets from immunized mito-QC mice (n = 7), pooled from two independent experiments. **d**, Maximum intensity projection of three-dimensional (3D) Airyscan ICC images depicting the mitochondrial network (GFP) in MACS-enriched and flow-sorted GC B cells (light zone, gray zone and dark zone) and magnetic bead-sorted naive B cells collected from immunized (enhanced SRBC immunization protocol at day 12) and unimmunized mito-QC spleens, respectively. Scale bar, 3 μm. Three-dimensional mitochondrial volume (μm³) quantification in naive (n = 10), light zone (n = 49), gray zone (n = 29) and dark zone (n = 28) cells isolated from four mice. Representative of two independent experiments. **e**, Spleen sections from Aicda-Cre × Rosa26[STOP]tdTomato

mice immunized with SRBC and analyzed on day 12, with IHC for COXI and mitochondrial import receptor subunit TOM20 homolog (TOMM20). Scale bar, 50 μm. Representative of two independent experiments. **f**, Flow cytometry histogram plots and quantification of COXI protein levels in GC B cells normalized to IgD+ naive B cells from the same mice (n = 4). Representative of three independent experiments. **g**, Representative Airyscan confocal images of in vivo 5-EU incorporation (indicating RNA synthesis), with COXI and TFAM in ex vivo-isolated naive and GC B cells. Quantification of mitochondrial 5-EU-integrated signal density in ex vivo-labeled naive (n = 23) and GC B (n = 32) cells isolated from two mice. Scale bar, 3 μm. Representative of two independent experiments. **h**, Two-dimensional Airyscan ICC images of TFAM in naive and GC B cells, with 3D reconstruction indicating individual mitochondrial nucleoids. Quantification of nucleoid number and volume (μm³) in naive (n = 22) and GC B (n = 24) cells isolated from two mice. Scale bar, 1 μm. Representative of two independent experiments. **i**, Deconvoluted confocal and STED super-resolution ICC images of TFAM in naive and GC B cells. Scale bar, 1 μm. Representative of two imaging experiments. Statistical significance was calculated using an unpaired two-tailed *t*-test (**b**,**g**,**h**), a two-tailed Mann–Whitney *U*-test (**f**) or an ordinary one-way analysis of variance (ANOVA) with Tukey's multiple comparisons test (**c**,**d**). Data are presented as the mean ± s.e.m. DZ, dark zone; LZ, light zone; GZ, gray zone.

which is active from the early pro-B cell stage[27]. TFAM was efficiently deleted in mature splenic B cells (Extended Data Fig. 2a,b). B-Tfam mice appeared healthy, with no clinical signs of immunosenescence or overt autoimmunity. However, B-Tfam mice had a profound reduction in the peripheral B cell compartment (Fig. 2a and Extended Data Fig. 2c–e). Analysis of B cell development in the bone marrow indicated a failure of progression from the pro- to the pre-B cell stage (Fig. 2b,c and Extended Data Fig. 2f). We noted that heterozygous B-Tfam mice had normal B cell development and an intact peripheral B cell compartment (Extended Data Fig. 2g,h).

To understand the expression of ETC proteins during B cell development, and the effect of *Tfam* deletion, we quantified a panel of ETC proteins by high-dimensional spectral flow cytometry in B wild-type (B-WT) and B-Tfam bone marrow and spleen (Fig. 2d,e). We found that in B-WT bone marrow, there was a progressive increase in the expression of most ETC proteins from the pre-pro-B stage (Hardy fraction A[28]), peaking at the earliest pre-B subset (fraction C′) and then substantially falling in the later stages (fractions D–F). Interestingly, TFAM expression peaked early (fraction A), suggesting that it might initiate a mitochondrial transcription program leading to upregulation of ETC proteins. The maximum expression of ETC proteins corresponded to the developmental block seen in B-Tfam mice (pro- to pre-B cell stage). B-Tfam mice deleted TFAM from fraction A onwards in the bone marrow and TFAM remained deleted in splenic B cell subsets;

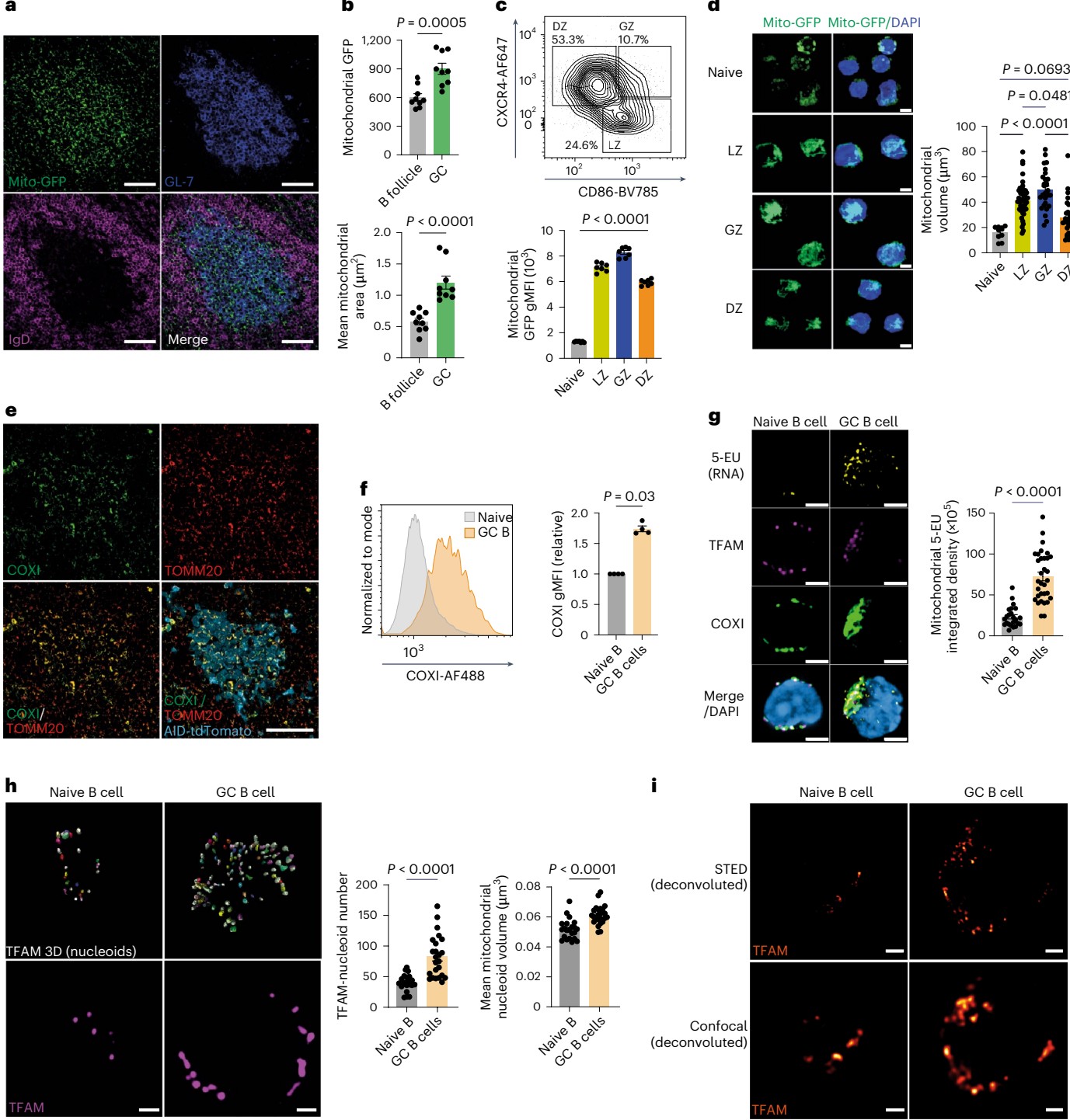

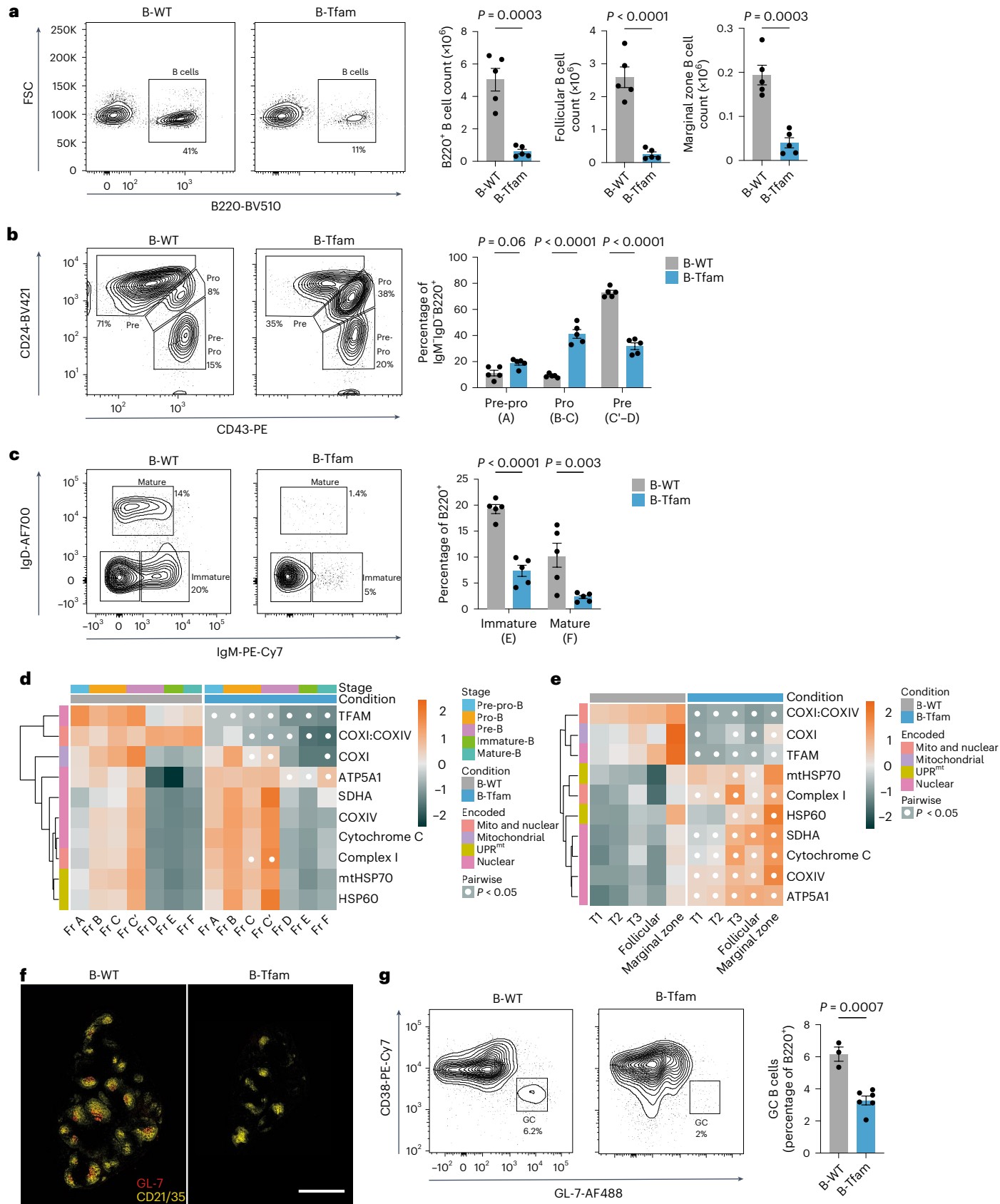

accordingly, mitochondrially encoded COXI was downregulated. The nuclear-encoded ETC components COXIV, cytochrome C, ATP5A1 and succinate dehydrogenase A (SDHA) were increased.

In the periphery, ETC component expression was similar across transitional and follicular B cells but increased in marginal zone B cells (Fig. 2e). TFAM deletion led to a much more marked upregulation of

**Fig. 2 | *Tfam* is essential during B cell development and differentiation.**
**a**, Representative flow cytometry plots of B220[+] B cells from the spleens of unimmunized 6-week-old B-Tfam and B-WT mice, with quantification of total splenic B cell counts and absolute counts of follicular (CD23[+]CD21[int]) and marginal zone (CD23[-]CD21[+]) B cell subsets (*n* = 5 mice). Representative of two independent experiments. FSC, forward scatter. **b**, Representative flow cytometry plots and quantification of bone marrow IgM[-]IgD[-]B220[+] pre-pro-B, pro-B and pre-B cells from B-Tfam and B-WT mice (*n* = 5 mice). Results are representative of two independent experiments. Equivalent Hardy stages are shown in parentheses. **c**, Representative flow cytometry plots and quantification of bone marrow-resident B220[+]IgM[+]IgD[-] immature and B220[+]IgD[+] mature B cell subsets from B-WT and B-Tfam mice (*n* = 5 mice). Results are representative of two independent experiments. Equivalent Hardy stages are shown in parentheses. **d**, Heatmap of row *z*-scores for the gMFI of indicated mitochondrial proteins, measured by spectral flow cytometry in bone marrow B cell subsets (Hardy stages) of unimmunized B-WT and B-Tfam mice (mean of *n* = 2). Results are representative of two independent experiments with *n* = 4 mice per group

in total. Fr A, pre-pro-B; Fr B-C, pro-B; Fr C′-D, pre-B; Fr E, immature-B; Fr F, mature-B. **e**, Heatmap of row *z*-scores for the gMFI of indicated mitochondrial proteins, measured by spectral flow cytometry in splenic B cell subsets of unimmunized B-WT and B-Tfam mice (mean of *n* = 2). Results are representative of two independent experiments with *n* = 3 mice per group in total. **f,g**, B-WT and B-Tfam mice were immunized with SRBCs (days 0 and 5) intraperitoneally; on day 12 spleens were analyzed by IHC and flow cytometry. **f**, Representative tile-scan images of spleen sections from B-Tfam and B-WT mice depicting GL-7[+] GCs and CD21/35[+] B cell follicles. Scale bar, 500 μm. Results are representative of two independent experiments. **g**, Flow cytometry gating strategy and quantification of CD38[-]GL-7[+] GC B cells in the spleens of immunized B-Tfam (*n* = 6) and B-WT (*n* = 3) mice. Results are representative of two independent experiments. Statistical significance was calculated using an unpaired two-tailed *t*-test (**a,g**) or a two-way ANOVA with Šidák's correction (**b,c**) or Tukey's (**d,e**) multiple comparisons tests, with experimental batch as a covariate (**d,e**). Data are presented as the mean ± s.e.m.

nuclear-encoded ETC proteins. This was evident in a mismatch in the ratio of proteins encoded by mitochondrial or nuclear genomes in the same ETC complex, that is, COXI and COXIV of complex IV. Extracellular flux analysis of unstimulated peripheral B cells showed no difference in oxygen consumption rate (OCR) but a significant increase in extracellular acidification rate in B-Tfam mice (Extended Data Fig. 2i).

Also increased in B-Tfam B cells were the mitochondrial unfolded protein response (UPR[mt])-associated proteins heat shock protein 60 (HSP60) and mitochondrial 70-kDa heat shock protein (mtHSP70) (Fig. 2d,e). These results demonstrate that ETC components are dynamically regulated across B cell developmental trajectories and that TFAM is required for the expression of mitochondrially encoded proteins.

We next examined the ability of B-Tfam mice to generate GCs. Examination of immunized B-Tfam spleens revealed smaller and fewer B cell follicles and an almost complete absence of GL-7[+] GCs compared to B-WT mice (Fig. 2f,g). *Tfam* heterozygosity in B cells also led to a significantly reduced GC response after SRBC immunization (Extended Data Fig. 2j). These results collectively suggest that TFAM is essential for normal B cell differentiation and that in situations of low-energy demand, partial respiratory compensation is possible despite substantially reduced translation of mitochondrial proteins.

## GC B cells require TFAM

Having established a role for TFAM in B cell development and ETC complex balance, we next focused on its function in the GC reaction. To specifically delete *Tfam* in GC B cells, we generated

Aicda-Cre × Tfam[loxP] × Rosa26[STOP]tdTomato × Blimp1-mVenus mice (hereafter Aicda-Tfam), which allow simultaneous identification of cells that have expressed *Aicda* (activated, GC, memory B and plasma cells) and/or currently express Blimp1 (encoded by *Prdm1*) (by detecting the fluorescent reporter proteins tdTomato and mVenus, respectively) (Fig. 3a). Aicda-Tfam mice were immunized with SRBCs and analyzed on day 12. Immunofluorescence staining of Aicda-Tfam spleen sections demonstrated small GCs in reduced numbers compared with Aicda-WT controls (Fig. 3b). Clusters of plasma cells in the splenic cords were much smaller in Aicda-Tfam mice and Blimp1-mVenus[+]tdTomato[+] cells were poorly represented, suggesting impairment of GC output (Fig. 3c). The proportion of CD138[+]tdTomato[+] long-lived plasma cells in bone marrow was also significantly lower (Extended Data Fig. 3a). We found that the generation of CD19[+]IgD[lo]CD38[-]GL-7[+] (and correspondingly CD19[+]IgD[lo]CD38[-]tdTomato[+]) GC B cells was substantially reduced in Aicda-Tfam mice as measured by flow cytometry (Fig. 3d,e). We also confirmed a reduction in Blimp1-mVenus[+]tdTomato[+] post-GC plasma cells (Fig. 3f).

Next, we examined antigen-specific GC formation by immunizing mice with (4-hydroxy-3-nitrophenyl) acetyl-chicken gamma globulin (NP-CGG). The proportion and absolute numbers of NP-binding GC B cells were significantly reduced in Aicda-Tfam mice, as were those of plasma cells measured in situ; the plasma cell to GC B cell ratio was correspondingly decreased (Fig. 3g and Extended Data Fig. 3b–d). Affinity maturation was also compromised, with decreased binding of IgG1 antibodies to NP[1–4] compared with NP[>20] (Fig. 3h and Extended

**Fig. 3 | GC B cells require TFAM. a**, Schematic of Aicda-Tfam mouse (Aicda-Cre × Tfam[loxP] × Rosa26[STOP]tdTomato × Blimp1-mVenus). **b–f**, Aicda-WT and Aicda-Tfam mice were immunized with SRBCs intraperitoneally; on day 12, spleens were analyzed using flow cytometry or IHC. **b,c**, Representative Airyscan IHC confocal images of spleen sections (**b**) and tdTomato[+]Blimp1-mVenus[+] plasma cell clusters in red pulp (**c**) from Aicda-Tfam and Aicda-WT mice. Scale bars, 500 μm and 75 μm, respectively. Representative of three independent experiments. **d,e**, Representative flow cytometry plots for the identification of GC B cells from Aicda-WT (*n* = 6) and Aicda-Tfam (*n* = 4) mice. Quantification of CD38[-]GL-7[+] GC B cell (**d**) and CD38[-]tdTomato[+] GC B cell (**e**) percentages within the CD19[+] B cell compartment. Representative of four independent experiments. **f**, Representative flow cytometry plots and quantification of tdTomato[+] post-GC plasma cells within the Dump[-] (IgD[-], Gr1[-], CD3[-]) Blimp1-mVenus[+] cell compartment (*n* = 3 mice per group). Data are representative of four independent experiments. **g,h**, Aicda-WT and Aicda-Tfam mice were immunized with 50 μg NP-CGG in alum (at a 1:1 ratio) intraperitoneally; on day 14, spleens and sera were collected for flow cytometry, IHC and enzyme-linked immunosorbent assay analyses. **g**, Representative flow cytometry plots depicting CD19[+]CD38[-]GL-7[+] GC B cells binding NP-PE or allophycocyanine (NP-APC) from

Aicda-Tfam (*n* = 10) or Aicda-WT mice (*n* = 4). Quantification of NP-specific GC B cell absolute counts is shown. Data are representative of three independent experiments. **h**, Enzyme-linked immunosorbent assay quantification and comparison of the ratio of IgG1 NP-specific high-affinity antibodies to low-affinity antibodies detected by binding to NP[1–4] and NP[>20] antigens, respectively, from Aicda-WT (*n* = 6) and Aicda-Tfam mice (*n* = 5). Data were pooled from two independent experiments. **i–k**, Aicda-WT and Aicda-Tfam mice were immunized with 50 μg NP-CGG in alum (at a 1:1 ratio) intraperitoneally; on day 30, they were boosted with 50 μg NP-CGG in PBS. Spleens and bone marrow were collected for flow cytometry on day 70. **i**, Representative flow cytometry plots depicting tdTomato[+]IgG1[+]CD38[+] memory B cells binding NP-APC from Aicda-WT and Aicda-Tfam mice. **j**, Proportional comparison and absolute count quantification of NP-binding within CD38[+]IgG1[+]tdTomato[+] memory B cells from Aicda-WT and Aicda-Tfam mice (*n* = 4 per group). Representative of two independent experiments. **k**, Absolute plasma cell counts in the bone marrow of Aicda-WT and Aicda-Tfam mice (*n* = 6 per group). Data were pooled from two independent experiments. Statistical significance was calculated using a two-tailed Mann–Whitney *U*-test (**d,e**) and an unpaired two-tailed *t*-test (**f–h,j–k**). Data are presented as the mean ± s.e.m.

Data Fig. 3e,f). However, there was no difference in IgM anti-NP antibodies, which is in keeping with the extrafollicular origin of this response, produced by plasmablasts that have not expressed *Aicda* (Extended Data Fig. 3c,g).

To understand if loss of *Tfam* compromised B cell memory, we immunized Aicda-Tfam and Aicda-WT mice with NP-CGG in alum and then boosted them on day 30 with NP-CGG in PBS. On day 70 there were substantially fewer NP-binding tdTomato⁺CD38⁺IgG1⁺ memory B cells (Fig. 3i,j) and on day 49 a reduction in IgG1 anti-NP antibodies reactive against NP$_{1-4}$ and NP$_{>20}$ (Extended Data Fig. 3h). The number of plasma cells in the bone marrow was also significantly reduced on day 70 (Fig. 3k).

Mitochondria play a central role in the regulation of apoptosis, but surprisingly the apoptosis rate detected by means of active caspase 3 staining in Aicda-Tfam GC B cells was comparable with that of Aicda-WT controls (Extended Data Fig. 3i). In situ TUNEL also demonstrated unaltered apoptosis (Extended Data Fig. 3j). There were no significant differences in cell cycle dynamics between Aicda-Tfam and Aicda-WT mice (Extended Data Fig. 3k,l). We also confirmed that Tfam deletion did not affect proliferation or cell viability in vitro (Extended Data Fig. 3m–p).

These data demonstrate that loss of *Tfam* markedly compromises GC B cell differentiation and output but without detectable effects on proliferation and survival in those cells already committed to the GC fate.

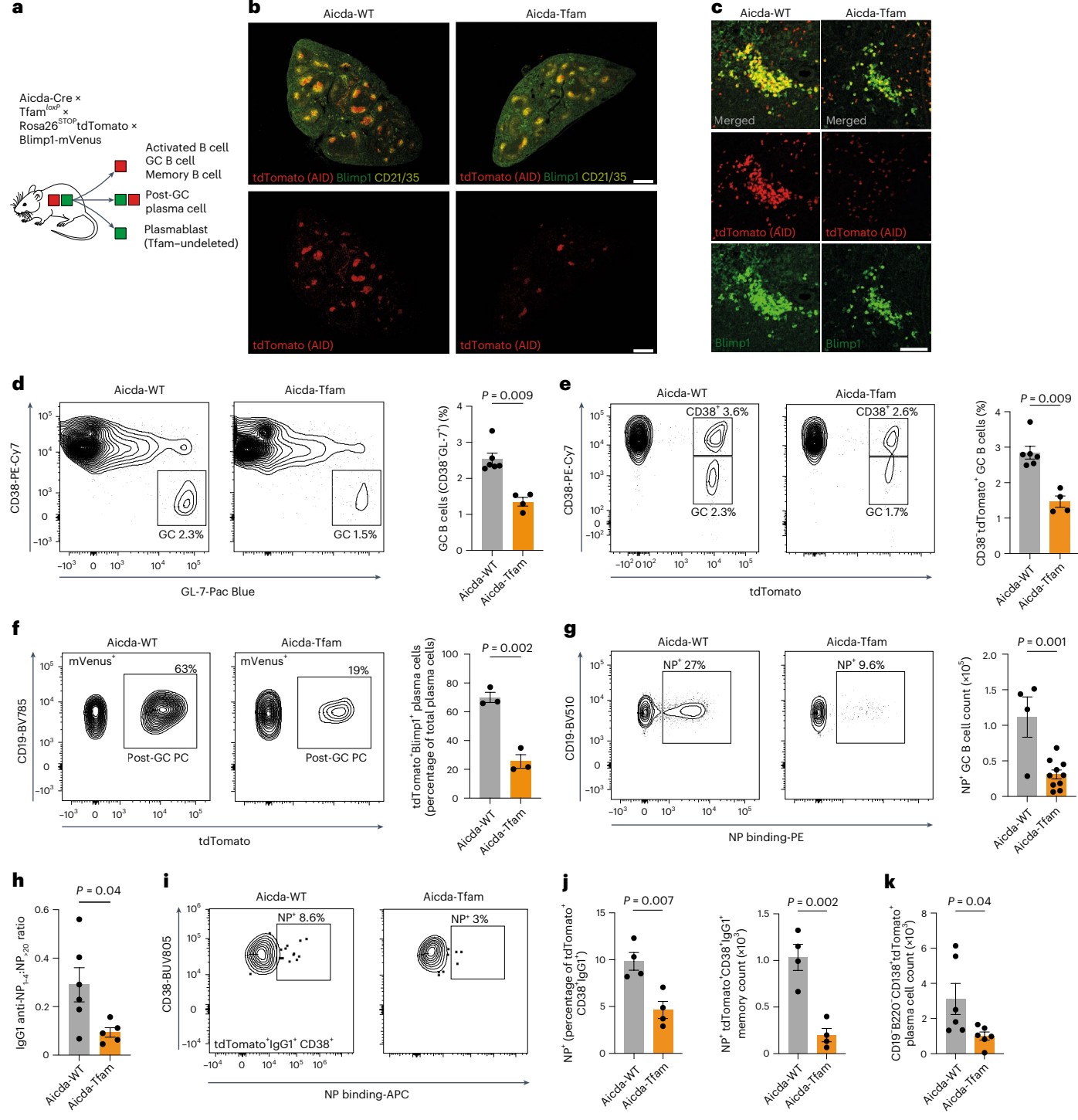

## TFAM controls transcriptional entry into the GC program

To examine the effects of *Tfam* deletion on the transcriptional program of GC B cells, we performed combined single-cell gene expression profiling and V(D)J sequencing. We immunized Aicda-Tfam and Aicda-WT mice with NP-CGG and sorted tdTomato[+] cells on day 14 (Fig. 4a). This population includes any B cells that have expressed *Aicda* and so will include pre-GC, GC, memory and plasma cells. We identified nine shared clusters, broadly separated into B cells from GC, CD38[+] non-GC and plasma cell populations (Fig. 4b,c). Cluster 0 (C0), which expressed markers of immaturity suggestive of an activated precursor (AP) state (*Ighd*, *Ccr6*, *Gpr183*, *Cd38*, *Sell* and low levels of *Bcl6*), was significantly expanded in Aicda-Tfam compared with Aicda-WT mice (Fig. 4d).

Examination of differential gene expression in C0 revealed, as expected, broad dysregulation of mitochondrial gene expression in Aicda-Tfam mice, in keeping with the function of TFAM as a regulator of mitochondrial transcription (Fig. 4e). Pathway analysis[29] demonstrated substantial downregulation of translation initiation and elongation gene sets (Fig. 4f). Gene components of the activator protein 1 (AP-1) signaling pathway *Jun*, *Junb*, *Jund* and *Fos* were all significantly upregulated (Fig. 4g). The AP-1 pathway is broadly upregulated by cellular stress signaling, reactive oxygen species and in response to environmental cues[30,31].

Analysis of the main GC cluster (C1) demonstrated as before dysregulation of mitochondrial and ribosomal gene transcription, with upregulation of *Jun*, *Fos* and *Junb* (Fig. 4h,i). Notably increased in Aicda-Tfam cells was *Rgs1* (regulator of G-protein signaling-1), a GTPase-activating protein that has an important role in the negative regulation of cell movement in response to chemokines (for example, CXCL12)[32,33] (Fig. 4j). Downregulated were *Coro1a* (coronin-1), which encodes an actin-binding protein required for cell migration[34], and *Arpc3* (actin-related protein 2/3 complex subunit 3) (Fig. 4h), which mediates branched actin polymerization and actin foci formation[35].

We next examined B cell clonality by evaluating V(D)J sequences. Because we sorted all tdTomato[+] B cells, the overall clonal diversity of all samples was high (Fig. 4k). As expected in the anti-NP immune response, V$_H$1-72 usage was dominant (Fig. 4l). However, there was more diversity in Aicda-Tfam mice, with fewer larger clones, suggesting that there was less ability for the evolution of dominant clones.

There was significantly less SHM in Aicda-Tfam B cells (Extended Data Fig. 4a), including in the *Ighv1-72* gene in GC B cells (Extended Data Fig. 4b,c). The W33L mutation in CDR1 and substitution of K59 of the V$_H$1-72 heavy chain confer increased affinity for NP[36,37]. There were significantly lower W33L and K59 substitution rates in the Aicda-Tfam GC B cell cluster, in keeping with our observation that high-affinity NP binding was reduced (Fig. 4m). There was no W33L mutation and negligible K59 substitution in the AP cluster (C0) of either Aicda-WT or Aicda-Tfam mice, reflecting their pre-GC state (Extended Data Fig. 4d).

Therefore, transcriptional analysis revealed evidence of failure of Aicda-Tfam B cells to transition from an AP to a committed GC B cell phenotype.

## TFAM is required for GC B cell commitment

Given the relative accumulation in immunized Aicda-Tfam mice of AP B cells with high expression of *Sell* (selectin L), *Ccr6* and a GC

transcriptional profile suggestive of altered cell trafficking and cytoskeleton dynamics, we hypothesized that *Tfam* is required for activated B cells to enter the GC and remain appropriately spatially positioned.

We first confirmed the proportional expansion of APs, defined as tdTomato[+]CD38[+]IgD[+], in Aicda-Tfam mice after NP-CGG immunization (Fig. 5a). Despite a significant numerical reduction in GC B cells, the AP population was maintained (Extended Data Fig. 5a). This was also seen to a more striking extent in B-Tfam mice immunized with SRBC, with APs defined as IgD[+]GL-7[int] (Extended Data Fig. 5b). We found that in Aicda-WT mice, a lower proportion of AP B cells bound NP compared to GC B cells, and did so with lower affinity, in keeping with their lack of a W33L mutation and low levels of SHM (Extended Data Fig. 5c).

We then examined the splenic sections of immunized Aicda-Tfam mice under high magnification. We found a highly disorganized GC architecture, with poor GC B cell compartmentalization; within the follicle, there were relatively many more tdTomato[+]IgD[+] B cells in Aicda-Tfam mice (Fig. 5b). The level of B cell lymphoma 6 (BCL6) protein, a master transcriptional regulator of GC commitment and entry, was lower in Aicda-Tfam GC B cells (Extended Data Fig. 5d).

We next asked whether it was possible to overcome the failure of AP B cells to enter the GC by adoptively transferring preactivated Tfam[−/−] AP-like B cells into primed congenically marked mice. We used the induced GC B cell (iGB) culture system[38] and TAT-Cre to delete *Tfam* in *Tfam*[loxP] × Rosa26[STOP]tdTomato B cells, or with WT congenically marked CD45.1/2 control B cells. This experimental design allows competitive transfer of activated B cells to take place (Fig. 5c). The resulting Tfam[−/−] iGB cells effectively deleted TFAM on day 4 of culture and this resulted in decreased expression of mitochondrially encoded ETC proteins and upregulation of nuclear-encoded proteins, as seen in splenic B cells from B-Tfam mice (Extended Data Fig. 5e). Loss of *Tfam* did not affect cell expansion over 4 days of culture (Extended Data Fig. 5f). However, after 5 days, Tfam[−/−] iGB cells were at a substantial competitive disadvantage in GC participation after adoptive transfer (Extended Data Fig. 5g and Fig. 5d–f).

We reasoned that the defects we observed on deletion of *Tfam* might be due to abnormalities in T$_{FH}$-B cell interaction, or a defective T$_{FH}$ pool. Although there was no numerical defect in the T$_{FH}$ compartment (Fig. 5g,h), and antigen presentation was intact (Extended Data Fig. 5h), we noted impairment of T$_{FH}$-induced in vitro GC B cell differentiation (Fig. 5i,j and Extended Data Fig. 5i).

To determine whether the defect we saw in Aicda-Tfam mice was B cell-intrinsic, we generated mixed competitive bone marrow chimeras, in which T$_{FH}$ generation would be intact. Aicda-Tfam GC B cells were outcompeted by WT cells compared with Aicda-WT controls in the spleen and Peyer's patches (Extended Data Fig. 5j). Overall these data suggest that TFAM promotes the entry or maintenance of activated B cells into the GC and that this function is principally cell-intrinsic.

## TFAM regulates mitochondrial translation in activated B cells

We next examined the expression of TFAM and mitochondrial ETC components after immunization. There was dynamic expression of TFAM, which peaked at the AP stage and then subsided after GC

---

**Fig. 4 | TFAM controls transcriptional entry into the GC program. a**, Schematic of the experimental design. Aicda-Tfam and Aicda-WT mice (*n* = 3 mice per group) were immunized with 50 µg NP-CGG with alum (at a 1:1 ratio) intraperitoneally; live Dump[−] (CD3[−], Gr1[−], CD11c[−]) tdTomato[+] cells were flow-sorted before bead capture and 10X library preparation and sequencing. A total of 9,948 cells from Aicda-Tfam and 9,667 cells from Aicda-WT mice were sequenced. **b**, UMAP and clustering of integrated Aicda-WT and Aicda-Tfam (*n* = 3 mice per group) datasets. IFN, interferon. **c**, Heatmap of selected differentially expressed marker genes by cluster for all samples. **d**, Cluster proportions between groups. **e**, Volcano plot of differentially expressed genes in C0 (APs). **f**, SCPA of Gene Ontogeny biological processes (GO BPs) in the AP cluster. **g**, *Jund* gene expression projected onto

clustered data. **h**, Volcano plot of differentially expressed genes in cluster 1 (C1) (GC). **i**, SCPA of GO BPs in the GC cluster (C1). **j**, *Rgs1* gene expression projected onto clustered data. **k**, Plot of clonotype abundance distribution for each sample. **l**, Plot of proportions of top clonotypes for each sample. Clonotypes with IGHV1-72 usage are highlighted. **m**, Quantification of substitution at *Ighv1-72* amino acid coding positions 33 and 59 for the GC B cell cluster. The number of GC B cells with Ighv1-72 usage analyzed are indicated on the plot. Statistical significance was calculated with a two-tailed *t*-test with correction for multiple comparisons using the Bonferroni method (**d**,**e**,**h**), a multisample Mahalanobis crossmatch test, with correction for multiple comparisons using the Bonferroni method (**f**,**i**) or a two-tailed chi-squared test (**m**).

entry, and a progressive increase in COXI, which was maximal in GC B cells from Aicda-WT mice (Fig. 6a,b). Other ETC proteins were also highly expressed in GC B cells, including those encoded in nuclear and mitochondrial DNA. Deletion of *Tfam* led to a program of alteration in

mitochondrial ETC protein expression in AP and GC B cells in B-Tfam mice (Fig. 6c), much like that observed during B cell development.

To directly confirm the activity of mitochondrial translation, we then measured incorporation of the amino acid analog

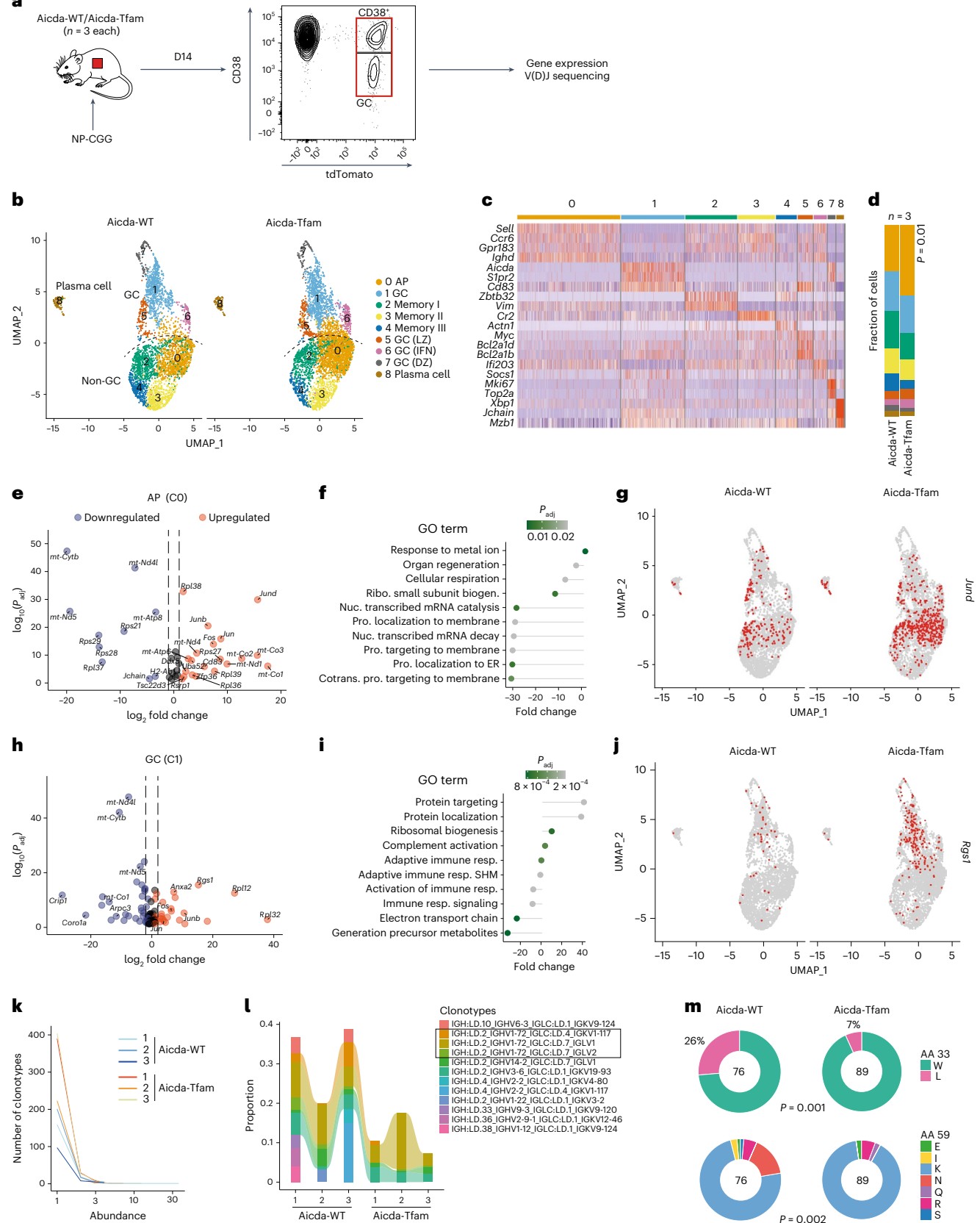

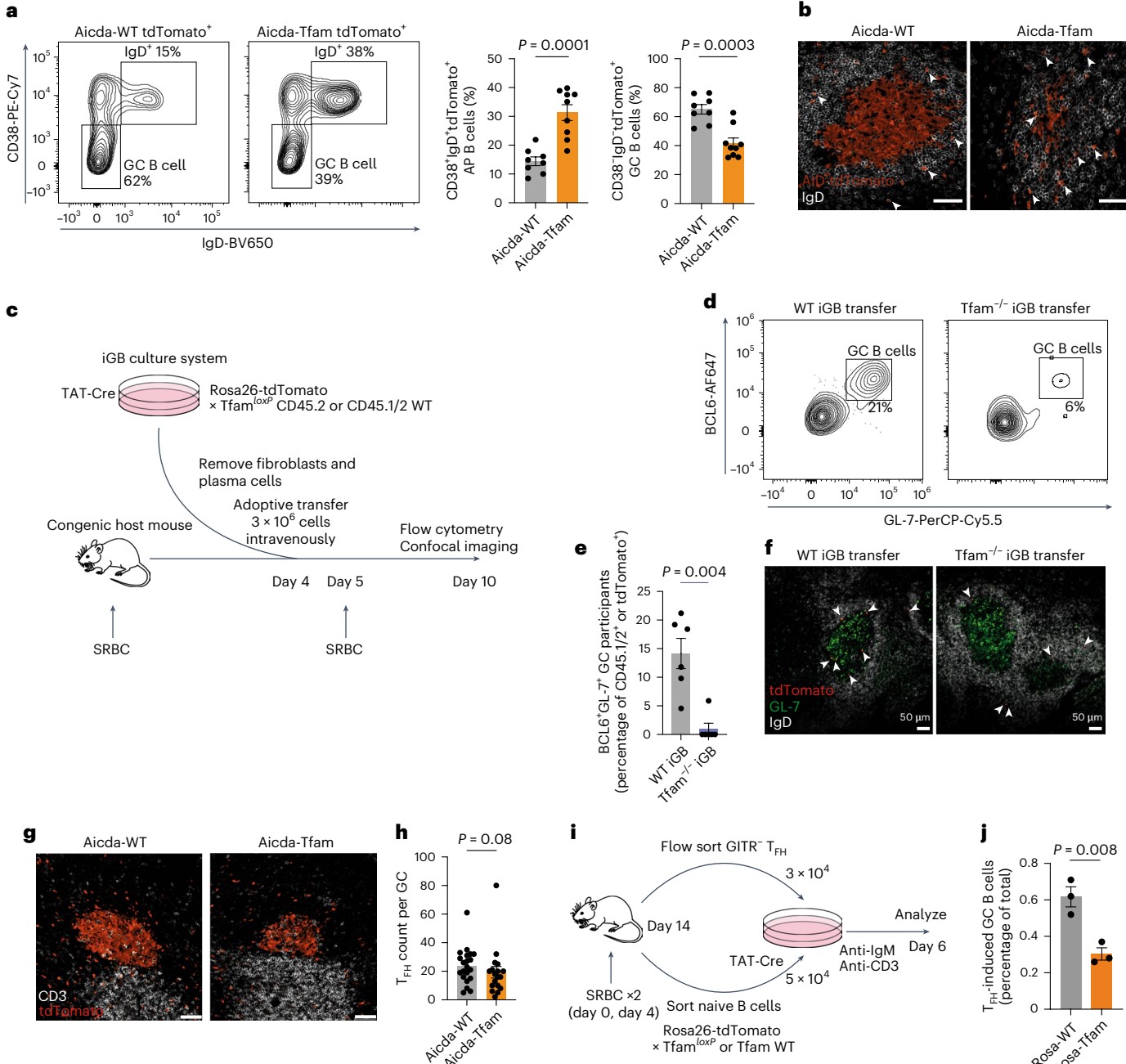

**Fig. 5 | TFAM is required for GC B cell commitment. a**, Representative flow cytometry plots depicting the gating of CD38⁺IgD⁺ APs and CD38⁻IgD⁻ GC B cells in the CD19⁺tdTomato⁺ population. Quantification of AP and GC B cell percentages from NP-CGG immunized Aicda-WT (*n* = 8) and Aicda-Tfam (*n* = 9) mice. Data were pooled from two independent experiments and are representative of four independent experiments. **b**, Representative Airyscan IHC images of GCs from Aicda-WT and Aicda-Tfam mice. The arrowheads indicate tdTomato⁺IgD⁺ APs. Scale bar, 50 μm. Representative of two independent experiments. **c**, Schematic of the experimental design for the iGB culture and adoptive transfer to assess in vivo GC entry and recruitment. **d**, Flow cytometric characterization of GC entry of adoptively transferred WT and Tfam⁻/⁻ iGB cells in SRBC-immunized CD45.1 congenic hosts on day 10. GC B cells were defined as BCL6⁺GL-7⁺. Representative of two independent experiments. **e**, Quantification of **d** (*n* = 6 mice per group). **f**, Confocal images of splenic sections depicting tdTomato⁺ WT and Tfam⁻/⁻

iGB cells adoptively transferred as in **c** but into separate congenic hosts. The arrowheads indicate transferred tdTomato⁺ iGB cells. Representative of two independent experiments. **g**, Representative confocal images of the splenic T_FH cell compartment in NP-CGG immunized Aicda-WT and Aicda-Tfam mice. Scale bar, 50 μm. Representative of three independent experiments. **h**, T_FH cell counts per GC from immunized Aicda-WT (*n* = 21 GCs) and Aicda-Tfam (*n* = 18 GCs) mice (as in **g**). Each symbol represents one GC pooled from *n* = 4 mice. Representative of three independent experiments. **i**, Schematic of the experimental design of the T_FH-B coculture. **j**, Percentages of in vitro T_FH-induced Rosa26-tdTom Tfam WT (Rosa-WT) or *Tfam^loxP* (Rosa-Tfam) GC B cells (GL-7⁺tdTomato⁺IgG1⁺) of the total viable cells. Technical replicates of *n* = 2 mice are shown. Representative of two independent experiments with *n* = 3 mice in total. Statistical significance was calculated using an unpaired two-tailed *t*-test (**a**,**j**) and a two-tailed Mann–Whitney *U*-test (**e**,**h**). Data are presented as the mean ± s.e.m.

O-propargyl-puromycin (OPP) in isolated ex vivo mitochondria from Aicda-WT and Aicda-Tfam B cells by flow cytometry (Fig. 6d,e). The proportion of red fluorescent protein (RFP)⁺ mitochondria (originating from

tdTomato⁺ AP/GC B cells) (Extended Data Fig. 6a) was substantially lower in Aicda-Tfam mice, reflecting their defective GC formation (Fig. 6e). We detected significantly more OPP incorporation in RFP⁺ mitochondria

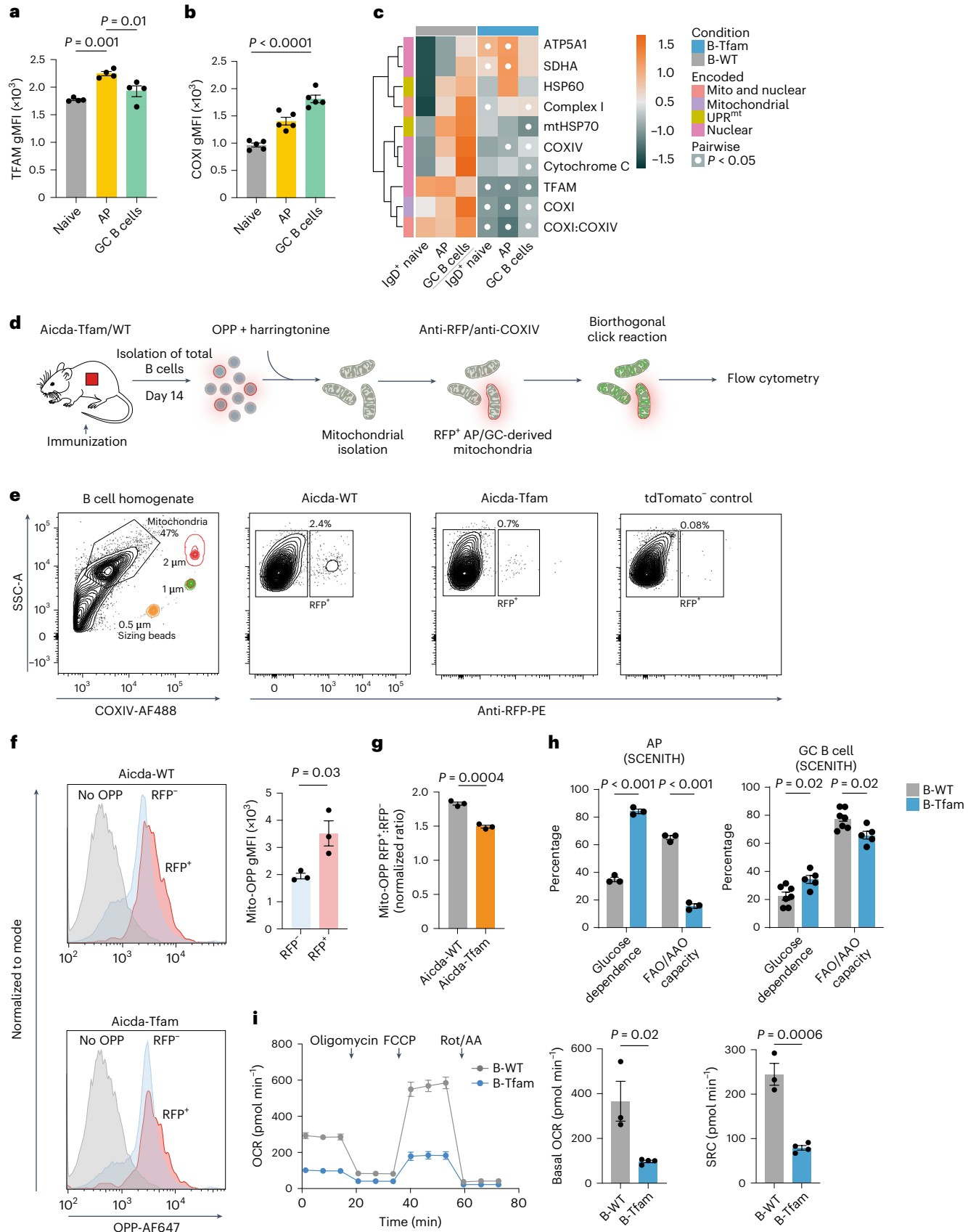

than those of non-AP or GC B cell origin (RFP⁻) (Fig. 6f). Translation was decreased in RFP⁺ mitochondria from Aicda-Tfam B cells compared to Aicda-WT cells (Fig. 6g); using Tfam⁻/⁻ iGB cells, treatment with the

mitochondrial translation inhibitor and oxazolidinone antibiotic chloramphenicol (CHL)[39] did not lead to additional suppression, confirming the defective translation in mitochondria (Extended Data Fig. 6b).

**Fig. 6 | TFAM regulates mitochondrial translation in activated B cells.**
**a,b**, Quantification of TFAM (**a**) ($n = 4$) and COXI (**b**) ($n = 5$ mice per group) gMFI in tdTomato$^+$CD38$^-$ GC B, tdTomato$^+$IgD$^+$ AP and tdTomato$^-$IgD$^+$ naive B cell compartments from SRBC-immunized Aicda-WT mice. Data are representative of three independent experiments. **c**, Heatmap of row $z$-scores for the gMFI of the indicated mitochondrial proteins, measured using spectral flow cytometry in the indicated B cell subsets of NP-CGG-immunized (day 14) B-WT and B-Tfam mice (mean of $n = 3$ mice per group). Results are representative of two independent experiments with $n = 4$ per group in total. **d**, Schematic of single-mitochondrion translation assay. **e**, Gating strategy for isolated mitochondria based on side scatter area (SSC-A), sizing beads and COXIV-AF488 signal, and for tdTomato$^+$ (RFP$^+$) AP/GC-derived mitochondria. **f**, Flow cytometry histogram plots depicting OPP-Alexa Fluor (AF) 647 signal in AP/GC-derived RFP$^+$ and naive B cell-derived RFP$^-$ mitochondria from immunized Aicda-Tfam and Aicda-WT mice, and no OPP (DMSO) vehicle control from pooled samples. Quantification of OPP gMFI in RFP$^+$ versus RFP$^-$ mitochondria from immunized Aicda-WT mice ($n = 3$ mice) is shown. Data were pooled from three independent experiments. **g**, Normalized OPP in RFP$^+$ AP/GC-derived mitochondria from Aicda-Tfam and Aicda-WT mice ($n = 3$ per group). The experiment was performed in three batches and values were pooled after batch correction. **h**, Quantification of glucose dependence and FAO or AAO capacity of cells based on OPP incorporation into splenic IgD$^+$GL-7$^{int}$ APs ($n = 3$ mice per group) and CD38$^-$IgD$^-$GL-7$^+$ GC B cells from B-WT ($n = 7$) and B-Tfam mice ($n = 5$) treated ex vivo with metabolic inhibitors (oligomycin or 2-deoxy-D-glucose). Data were pooled from two independent experiments. See Supplementary Methods for details of the SCENITH calculations. **i**, Real-time OCR measurements (from the Mito Stress test) of $3 \times 10^5$ B cells stimulated overnight (anti-CD40 and IL-4) from B-Tfam ($n = 4$) and B-WT ($n = 3$) mice, with quantification of basal OCR and spare respiratory capacity. Results were pooled from two independent experiments. FCCP, carbonyl cyanide-$p$-trifluoromethoxyphenylhydrazone; Rot/AA, rotenone/antimycin A. Statistical significance was calculated using a one-way ANOVA with Tukey's multiple comparisons test (**a,b**), an unpaired two-tailed $t$-test (**f,i**), an unpaired two-tailed $t$-test after batch correction (**g**) or a two-way ANOVA with Tukey's multiple comparisons test (**c**) or Šidák's correction (**h**). Data are presented as the mean ± s.e.m.

We then examined the metabolic consequences of *Tfam* deletion on APs and GC B cells, quantifying cytoplasmic protein translation rate by the incorporation of OPP as a proxy for metabolic capacity, as recently reported[40]. GC B cells had significantly higher basal OPP incorporation than AP B cells, reflecting their high levels of metabolic activity (Extended Data Fig. 6c). Loss of *Tfam* led to an increase in glucose dependence and a marked reduction in FAO/amino acid oxidation (AAO) capacity in AP B cells, seen to a lesser extent in GC B cells (Fig. 6h and Extended Data Fig. 6c,d). These results were mirrored by Seahorse extracellular flux analysis in activated B cells and iGBs (Fig. 6i and Extended Data Fig. 6e,f).

Therefore, loss of TFAM compromised GC B cell mitochondrial translation and impaired metabolic homeostasis.

### *Tfam* deletion disrupts B cell mobility

We found that Aicda-Tfam GCs were poorly compartmentalized, with smaller dark zones (Fig. 7a) and a disrupted dark zone to light zone ratio measured using flow cytometry (Fig. 7b).

Positioning of B cells in the GC is controlled by the chemokines CXCL12 and CXCL13, which promote migration to the dark zone and light zone, respectively[41]. As our preceding data indicated that AP B cells need TFAM to enter the GC, and given that the transcriptional profile of Aicda-Tfam GC B cells was suggestive of cytoskeletal and mobility defects, we hypothesized that TFAM was required for proper cellular positioning in GCs.

We examined the cellular actin network of TFAM-deficient B cells and found a significant increase in filamentous actin (F-actin) in Aicda-Tfam GC B cells (Fig. 7c,d), which was also evident in B-Tfam B cells (Extended Data Fig. 7a,b). Rearrangement of the actin cytoskeleton is critical for B cell migration[42]; to understand if Tfam deletion compromised GC B cell motility, we performed a transwell migration assay to determine chemotaxis in response to the chemokines CXCL12 and CXCL13. We found that Aicda-*Tfam* GC B cells migrated poorly, with significantly reduced chemotaxis compared to Aicda-WT cells (Fig. 7e).

B cell receptor (BCR) and chemokine-driven cytoplasmic calcium mobilization critically regulates F-actin organization in B cells[43]. As mitochondria are an important reservoir of intracellular calcium[44], we asked whether Tfam deletion led to dysregulated intracellular calcium levels. After CXCL12 stimulation, B cells from B-Tfam mice failed to sustain peak cytoplasmic calcium levels (Fig. 7f), which we also observed after BCR stimulation with anti-IgM (Fig. 7g). Interestingly, we also detected a significant upregulation in the levels of the mitochondrial calcium uniporter (MCU), suggesting elevated mitochondrial calcium uptake capacity in B-Tfam B cells, whereas CD3$^+$ T cells showed comparable MCU expression (Fig. 7h and Extended Data Fig. 7c).

Mitochondrial reactive oxygen species (mtROS) were also substantially increased in B-Tfam naive and APs (Fig. 7i and Extended Data Fig. 7d). ROS activate the AP-1 signaling pathway[31] and this observation was therefore consistent with the transcriptional profile of Aicda-Tfam AP B cells.

**Fig. 7 | *Tfam* deletion disrupts B cell mobility. a**, Representative Airyscan IHC images of spleen sections depicting GCs. Quantification of the dark zone area normalized to the GC area in SRBC-immunized Aicda-WT ($n = 41$ GCs) and Aicda-Tfam ($n = 17$ GCs) mice. Image analyses were performed including all GCs identified on two sections from $n = 2$ mice of each genotype. Scale bar, 100 μm. Representative of four independent experiments. **b**, Representative flow cytometry plots of the dark zone, gray zone and light zone subpopulations of tdTomato$^+$CD38$^-$GL-7$^+$ GC B cells. Quantification of the dark zone to light zone ratio from NP-CGG-immunized Aicda-WT ($n = 8$) and Aicda-Tfam ($n = 9$) mice is shown. Representative of four independent experiments. **c**, 3D Airyscan confocal images of magnetic bead-sorted and F-actin phalloidin + 4,6-diamidino-2-phenylindole (DAPI)-stained GC B cells from Aicda-WT and Aicda-Tfam mice. Scale bar, 6 μm. Representative of two independent experiments. **d**, Representative flow cytometry histogram of F-actin phalloidin fluorescence of tdTomato$^+$GL-7$^+$ GC B cells and tdTomato$^-$IgD$^+$ naive B cells from immunized Aicda-WT ($n = 6$) and Aicda-Tfam ($n = 8$) mice. Quantification of F-actin phalloidin gMFI in GC B cells normalized to naive B cells from the same host is shown. Data were pooled from three independent experiments. **e**, Quantification of chemotaxis to CXCL12 and CXCL13 in GC B cells from Aicda-WT ($n = 3$) and Aicda-Tfam ($n = 5$) mice. Data are representative of four independent experiments. **f**, Representative Fluo-4 AM geometric mean signal intensity kinetics (moving average) of B-WT ($n = 4$) and B-Tfam ($n = 4$ mice) B220$^+$ B cells stimulated with CXCL12 for 120 s. Quantification of the area under the curve (AUC) between 60 and 90 s (corresponding to the peak response) is shown. **g**, Representative Fluo-4 AM gMFI signal kinetics (moving average) of B-WT ($n = 4$) and B-Tfam ($n = 4$ mice) B220$^+$ B cells stimulated with anti-IgM for 300 s. Quantification of the AUC between 60 and 300 s is shown. In **f** and **g**, experiments were run as technical duplicates in four independent replicate experiments consisting of one B-Tfam and one WT mouse in each. The data points from the B-Tfam mice were normalized to the WT data run in the same batch. **h**, Representative flow cytometry histogram of MCU fluorescence in B220$^+$ IgD$^+$ B cells from unimmunized B-Tfam and B-WT mice. Quantification of MCU gMFI in IgD$^+$ B cells from B-Tfam and B-WT mice ($n = 3$). Data are representative of three independent experiments. **i**, Representative flow cytometry histogram of mtROS-Deep Red fluorescence in IgD$^+$ B cells from B-WT ($n = 4$) and B-Tfam ($n = 5$) mice. Data are representative of two independent experiments. **j**, Quantification of chemotaxis to CXCL12 in B cells from unimmunized B-Tfam ($n = 3$) and B-WT ($n = 5$) mice in the presence of mitoTEMPO or Ru265. Data are representative of two independent experiments and were pooled after batch correction. Statistical significance was calculated using an unpaired two-tailed $t$-test (**b,d,h,i**), a two-tailed Mann–Whitney $U$-test (**a,f,g**) or a two-way ANOVA with Šidák's correction (**e**), with batch as a covariate (**j**). Data are presented as the mean ± s.e.m.

Given these observations, we next tested whether either suppression of mtROS with the scavenger mitoTEMPO[45] or inhibition of MCU function with the ruthenium compound Ru265 (ref. 46) improved cell motility in B-Tfam B cells in response to CXCL12. MCU inhibition did not improve transwell migration but strikingly mitoTEMPO largely rescued the defect in B-Tfam cells (Fig. 7j).

Our data therefore collectively suggest that TFAM is required for proper cellular motility to enable entry into and movement within the

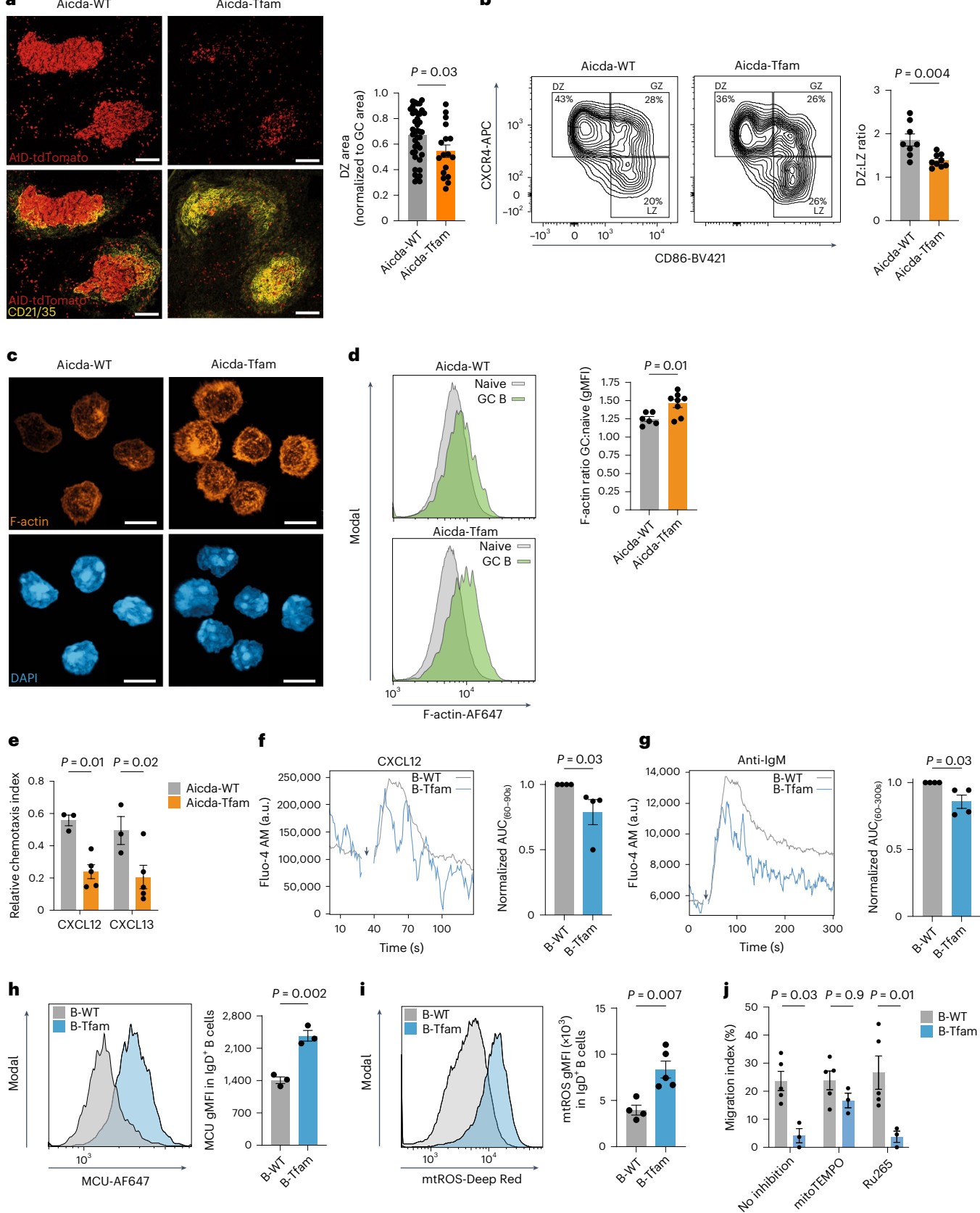

GC, and that this effect is associated with regulation of ETC function and mtROS generation.

### *Tfam* deletion in B cells prevents lymphoma

Having demonstrated an essential role for TFAM in B cell development and entry into the GC, we next asked whether it was required for the development of lymphoma. One of the most common mutations giving rise to DLBCL is translocation of *MYC* to immunoglobulin gene loci, leading to its unregulated expression. c-Myc is a key transcription factor regulating cell cycle and growth, cellular metabolism and mitochondrial biogenesis[47]. We reasoned that deletion of *Tfam* would counter the oncogenic effects of c-Myc overexpression. We employed the well-established Eµ-Myc transgenic mouse model of lymphoma, in which *Myc* is expressed under the control of the *Igh* enhancer[48]. Eµ-Myc mice develop lymphoma with high penetrance from a median age of 11 weeks. To understand what effects overexpression of *Myc* had on mitochondrial translation, we transferred established lymphoma cells from Eµ-Myc mice into WT CD45.1 congenic hosts to allow us to compare B cells from the same environment (Fig. 8a). We found very markedly higher expression of COXI in transferred lymphoma cells compared with WT B cells, with significantly upregulated TFAM (Fig. 8b,c).

We next generated *Cd79a*-Cre × *Tfam*^loxP × Eµ-Myc mice (B-Tfam-Myc). B-Tfam-Myc mice were completely protected from the development of lymphoma during the observation period of 30 weeks, compared with Eµ-Myc controls, which had a median survival of 12.5 weeks (Fig. 8d).

Finally, to establish whether inhibition of mitochondrial transcription and translation might be a therapeutic target in human lymphoma, we treated Daudi B cell lymphoma cells (originally arising from Burkitt lymphoma) with IMT1, a specific inhibitor of mitochondrial RNA polymerase (POLRMT), which functions along with TFAM to initiate mitochondrial RNA transcription[49]. We found that treatment with IMT1 led to a progressive reduction of COXI levels, with increasing COXI:succinate dehydrogenase B (SDHB) mismatch (Fig. 8e,f). Imaging of Daudi cells treated with IMT1 revealed mitochondrial enlargement and confirmed the loss of COXI (Fig. 8g). IMT1 reduced cell growth, inhibited cell cycle progression, increased mtROS levels and recapitulated the F-actin dysregulation we observed with *Tfam* deletion (Fig. 8h–k and Extended Data Fig. 8a). To inhibit mitochondrial translation, we used CHL. CHL reduced COXI expression and increased expression of F-actin, and the

UPR^mt protein LON peptidase 1 (LONP1) in Daudi cells (Fig. 8l,m and Extended Data Fig. 8b). Cell growth was substantially reduced (Fig. 8n).

These results define high rates of mitochondrial translation enabled by *Tfam* expression as an essential requirement for the development of B cell lymphoma and show the therapeutic potential of POLRMT and mitochondrial translation inhibition in human disease.

## Discussion

In this study, we show that on entry to the GC, B cells dramatically remodel their mitochondria, increasing their mass and radically altering their morphology. As part of this transition, mitochondrial translation is highly active and we demonstrate that the nuclear-encoded mitochondrial transcriptional and translational regulator TFAM is not only required for B cell development but also for their entry into the GC program, proper spatial anchoring and for the subsequent development of lymphoma.

GC B cells have a highly distinct metabolism, predominantly relying on oxidative phosphorylation despite their very rapid rate of proliferation in a hypoxic microenvironment[6,8]. For rapidly dividing cells to maintain high mitochondrial mass in the face of dilution to their daughter cells, a high rate of mitochondrial protein translation and division must be maintained. We were surprised to find that TFAM is required for entry into the GC program itself; when deleted, there is a proportional accumulation of B cells with an AP phenotype, which have expressed *Aicda* but maintain markers of immaturity. Recently, the TCA metabolite α-ketoglutarate was shown to be upregulated by interleukin-4 (IL-4) in B cells; this leads to epigenetic alteration of the *Bcl6* locus[9]. It is therefore possible that TFAM is required for an initial burst of mitochondrial biogenesis to facilitate GC program entry.

Deletion of TFAM substantially altered the balance of ETC protein expression, with loss of mitochondrially encoded subunits and compensatory upregulation of nuclear-encoded proteins. This was reflected in metabolic disturbance, with impaired oxidative phosphorylation after activation, upregulation of glycolysis and, importantly, mtROS generation, which was seen even in unstimulated cells. Temporal control of *Tfam* deletion either very early in B cell development using *Cd79a*-Cre, or after activation with Aicda-Cre or in vitro with TAT-Cre did not lead to unexpected or major phenotypic differences but it is possible that adaptations might occur in a dynamic manner.

An increase in F-actin was consistent across genetic or pharmacological interference with mitochondrial transcription and translation.

**Fig. 8 | *Tfam* deletion in B cells prevents lymphoma. a**, Schematic depicting the adoptive transfer strategy of lymphoma cells from Eµ-Myc mice. After the development of lymphoma, cells were transferred into WT congenic (CD45.1⁺) recipients. Representative flow cytometry plots showing the identification of CD19⁺CD45.2⁺ donor lymphoma cells and CD19⁺CD45.1⁺ WT B cells from the inguinal lymph nodes of recipient mice after 3 weeks. **b**, Flow cytometry histogram plot depicting COXI expression in CD45.2⁺ lymphoma cells and CD45.2⁻ WT B cells (*n* = 5 mice) as described in **a**. FMO, fluorescence minus one. Quantification of COXI gMFI. Data are representative of two independent experiments. **c**, Flow cytometry histogram depicting TFAM expression in CD45.2⁺ lymphoma cells and CD45.1⁺ WT B cells (*n* = 5 mice) as described in **a**. Quantification of TFAM gMFI. Data are representative of two independent experiments. **d**, Kaplan–Meier graph depicting the survival curve of Eµ-Myc × Tfam homozygous (*Cd79a*^Cre/+Tfam^loxP/loxP × Eµ-Myc, *n* = 9), Eµ-Myc Tfam heterozygous (*Cd79a*^Cre/+Tfam^loxP/+ × Eµ-Myc, *n* = 13) and Eµ-Myc (*Cd79a*^+/+Tfam^loxP/loxP × Eµ-Myc or Cd79a^+/+Tfam^loxP/+ × Eµ-Myc, *n* = 18) mice followed for up to 6 months of age. **e**, Representative flow cytometry histogram depicting COXI expression in the Daudi cell line treated with the IMT1 at various concentrations (0.1 µM, 1 µM, 10 µM) or vehicle (DMSO) for 120 h. Representative of three independent experiments. **f**, Quantification of COXI levels by flow cytometry, normalized to SDHB in IMT1- or vehicle-treated Daudi cells at different concentrations and time points (0, 24, 48, 72, 96, 120 h), representative of two independent experiments. **g**, Airyscan ICC confocal images of COXI and TOMM20 in Daudi cells treated with DMSO or IMT1 (1 µM) for 120 h. Scale bar,

3 µm. Representative of two independent experiments. **h**, Quantification of live Daudi cell concentrations over a 120-h incubation period in the presence of IMT1 or vehicle (DMSO), representative of two independent experiments with three to five technical replicates each. **i**, Flow cytometry-based quantification of mtROS in the Daudi cell line treated with IMT1 (1 µM) or vehicle (DMSO) for 120 h. Each point represents technical replicates (*n* = 3 per group). Representative of two independent experiments. **j**, Airyscan confocal images of Daudi cells treated with IMT1 (1 µM) or DMSO and stained for F-actin phalloidin. Scale bar, 5 µm. Representative of two independent experiments. **k**, Quantification of F-actin phalloidin fluorescence by flow cytometry (*n* = 5 technical replicates). Data are representative of two independent experiments. **l**, Quantification of COXI levels in Daudi cells by flow cytometry after treatment with vehicle (ethanol) or CHL at 10 µg ml⁻¹ or 25 µg ml⁻¹ for 120 h (*n* = 8 technical replicates). Data are representative of two independent experiments. **m**, Quantification of F-actin levels in Daudi cells by flow cytometry after treatment with vehicle (ethanol) or CHL at 10 µg ml⁻¹ or 25 µg ml⁻¹ for 120 h (*n* = 8 technical replicates). Data are representative of two independent experiments. **n**, Quantification of live Daudi cell concentrations over a 120-h incubation period in the presence of CHL at 10 µg ml⁻¹ or 25 µg ml⁻¹ or vehicle (ethanol) as in **m**. Representative of two independent experiments with four technical replicates (*n* = 4). Statistical significance was calculated using an unpaired two-tailed *t*-test (**b,c,i**), a Mantel–Cox log-rank test (**d**) or an ordinary one-way ANOVA with Tukey's (**h,n**) or Dunnett's multiple comparisons test (**k–m**). Data are presented as the mean ± s.e.m.

Dynamic actin cytoskeletal modification is essential for normal cell movement and its probable disruption through ROS accumulation, reversible with the addition of a ROS scavenger, probably contributes to the positioning and motility defects we observed[50]. This adds weight to the idea that mitochondria are intimately linked to cytoskeletal function and that this role may operate independently of ATP generation[51].

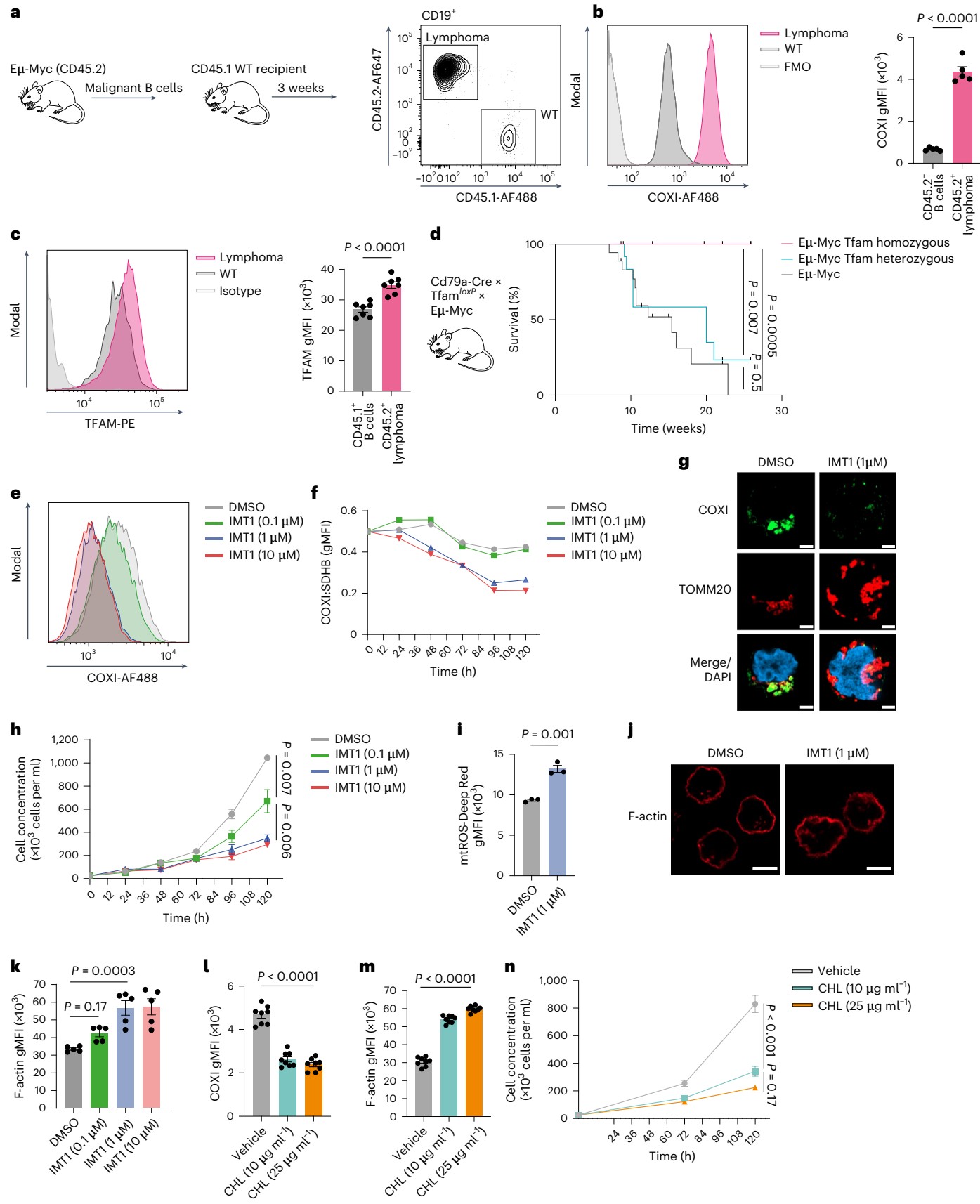

These defects collectively have the potential to compromise cellular interaction, in particular that with $T_{FH}$ cells required to enter the GC program, and for normal dynamics once within. We observed that Tfam$^{-/-}$ B cells failed to differentiate normally into GC B cells in an ex vivo $T_{FH}$ coculture system and yet were unaffected when CD40 ligation was artificially provided in the iGB culture system. The capacity of Tfam$^{-/-}$ iGB cells to present antigen to cognate T cells was normal, however, as were $T_{FH}$ cell numbers within GCs after immunization. How *Tfam* deletion in B cells may affect T cell interaction is therefore deserving of future study. We have used the iGB system developed by Nojima et al.[38] to precisely control deletion of *Tfam* before adoptive transfer; while this is an important and widely used tool to generate GC B cell precursors in vitro, which can then participate in the GC reaction in vivo, uncertainties remain about their fidelity to true APs and this would also benefit from future study.

Although we have directly established the importance of TFAM as a regulator of B cell development and activation, the factors driving the counterintuitive switch to oxidative phosphorylation in GC B cells is to be uncovered, as do the signaling mechanisms controlling the differences in mitochondria we observed between GC microenvironments. Disruption of mitochondrial integrity also induces a phenotype associated with immune aging; to what extent the mechanisms we describe might hold true in the diminished humoral immune response seen with age is another area deserving of further exploration[20].

We show that deletion of *Tfam* is sufficient to completely prevent the development of Myc-driven lymphoma. Although loss of *Tfam* at an early developmental stage leads to B cell lymphopenia, and thus the pool of B cells that may become malignant is reduced, the high penetrance of the model contrasted with the complete protection against lymphoma suggests that this is insufficient to explain the phenotype we observed. How TFAM acts to support lymphomagenesis requires further study but may be associated with its promotion of mitochondrial translation[52]. Our observation that inhibition of mitochondrial transcription and translation reduces growth of lymphoma cells suggest that this should be prioritized as a therapeutic target.

## Online content

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

## Methods

### Mice

B6.Cg-*Tfam*^tm1.1Ncdl^/J (strain 026123), B6.C(Cg)-*Cd79a*^tm1(Cre)Reth^/EhobJ (strain 020505), B6.129P2-*Aicda*^tm1(Cre)Mnz^/J (strain 007770), B6;129S6-*Gt(ROSA)26Sor*^tm9(CAG-tdTomato)Hze^/J (strain 007905), B6.Cg-Tg(TcraTcrb)425Cbn/J (strain 004194) and B6.Cg-Tg(IghMyc)22Bri/J (strain 002728) were purchased from The Jackson Laboratory. Tg(Prdm1-Venus)1^Sait^ [Blimp1-mVenus] (strain 3805969) was a kind gift from M. Saitou (Kyoto University). Gt(ROSA)26Sor^tm1(CAG-mCherry/GFP)Ganl^ (mito-QC) was a kind gift from I. Ganley (University of Dundee). B6.SJL.CD45.1 mice were provided by the central breeding facility of the University of Oxford. Male and female mice between the ages of 6 and 15 weeks were used. Mice were bred and maintained under specific pathogen-free conditions at the Kennedy Institute of Rheumatology, University of Oxford. Mice underwent regular checks to ensure they did not have any pathogenic microorganisms. They were housed in cages that had individual ventilation and were provided with items to stimulate their environment. The temperature was kept between 20 and 24 °C, with a humidity level of 45–65%. They were exposed to a 12-h cycle of light and darkness (7:00 to 19:00), with a 30-min period of dawn and dusk. All procedures and experiments were performed in accordance with the UK Scientific Procedures Act (1986) under a project license authorized by the UK Home Office (PPL no. PP1971784).

### Immunization

For SRBC immunization, 1 ml sterile SRBCs (catalog no. 12977755, Thermo Fisher Scientific) were washed twice with 10 ml ice-cold PBS and reconstituted in 3 ml PBS and administered as 0.2-ml injections intraperitoneally. In some experiments, an enhanced SRBC immunization method was followed to maximize GC B cell yield by immunizing mice with 0.1 ml SRBC on day 0 followed by a 0.2-ml second injection on day 5 (ref. [53]). For protein antigen immunizations, 50 μg NP$_{(30-39)}$-CGG (catalog no. N-5055D-5, Biosearch Tech) was mixed with alum (Thermo Fisher Scientific) or PBS for boost immunization at a 1:1 ratio and rotated at 20 °C (room temperature) for 30 min before intraperitoneal injection. For NP-CGG- and SRBC-based immunizations, day 14 and day 12 were used as read-out time points, respectively, unless specified otherwise.

### Flow cytometry and imaging flow cytometry

Flow cytometry was performed as described previously[54]. Briefly, collected spleens were injected with ice-cold PBS and mashed through a 70-μm strainer (Falcon) or crushed between the rough ends of microscope slides. For Peyer's patch and lymph node dissociation, a 40-μm strainer and 70-μm strainer (VWR) were used, respectively. RBCs were depleted by incubating splenocytes with ACK Lysis Buffer (Gibco) for 3–4 min at 20 °C. Single-cell suspensions were incubated with Fixable Viability Dye eFluor 780 (eBioscience) in PBS (30 min on ice), followed by Fc Block (5 min) and surface antibodies (30 min on ice) in FACS buffer (PBS supplemented with 0.5% BSA and 2 mM EDTA). For intracellular staining, cells were fixed at 20 °C with 4% freshly prepared paraformaldehyde (PFA) (Cell Signaling Technology) for 15 min and permeabilized with 90% freezer-cold methanol (10 min on ice with frequent vortexing) unless specified otherwise. Phalloidin-based F-actin (catalog no. A30107, Thermo Fisher Scientific) staining was performed using the BD Perm/Wash reagent (catalog no. 554723) after 4% PFA fixation. For in vivo cell cycle analysis, 5-ethynyl-2′-deoxyuridine (EdU) (1 mg, catalog no. A10044, Thermo Fisher Scientific) was injected intraperitoneally and mice were sacrificed after 2.5 h. Cells were stained for surface markers, fixed and permeabilized then labeled using Click chemistry according to the manufacturer's instructions (Click-iT Plus EdU Flow cytometry kit, catalog no. C10634, Thermo Fisher Scientific). FxCycle Violet (catalog no. F10347, Thermo Fisher Scientific) reagent was used for cell cycle characterization. For mitochondrial superoxide Deep Red

(mtSOX, Dojindo) uptake indicating mtROS, after viability dye and surface staining, cells were resuspended in warm complete Roswell Park Memorial Institute (RPMI) 1640 medium supplemented with mtSOX (10 μM) and incubated for 30 min at 37 °C. Cells were washed twice before flow cytometry acquisition. Flow cytometry was performed on a BD Fortessa X-20 or LSR II instrument or using a Cytek Aurora (5-laser) spectral flow cytometer. For imaging cytometry, single-cell suspensions were prepared from spleens of mito-QC mice, incubated with LIVE/DEAD dye, stained for surface markers and then fixed with 4% PFA. Washed cells were then resuspended in 50 μl FACS buffer and run on an Amnis ImageStream^X Mark II Imaging Flow Cytometer and analyzed with the Amnis IDEAS and FCS Express software (v.7.12). Flow cytometry data was analyzed using FlowJo (FlowJo LLC).

### Cell sorting

Naive B cells were isolated using the Pan B cell Isolation II Kit, anti-CD43, and/or anti-CD23 Microbeads (all Miltenyi Biotec) according to the manufacturer's instructions. Purity validated by flow cytometry was more than 90%. Isolation of the dark zone, light zone and gray zone subsets of GC B cells and T$_{FH}$ cells (CD19^−CD4^+CXCR5^+ICOS^+GITR^−) was performed via fluorescence-activated cell sorting (FACS).

For some experiments, untouched GC B cells were isolated using a magnetic bead-based protocol as described by Cato et al.[55]. Briefly, spleens were collected from SRBC-immunized mice and single-cell suspensions were prepared in ice-cold magnetic-activated cell sorting (MACS) isolation buffer (PBS with 0.5% BSA + 2 mM EDTA) followed by ACK RBC lysis (Gibco) for 4 min at 20 °C with occasional mixing every 30 s. After washing, cells were labeled with anti-CD43 microbeads (Miltenyi Biotec) and biotinylated antibodies against CD38 and CD11c (clones 90 and N418, respectively; both eBioscience), followed by incubation with anti-biotin microbeads, and were subsequently run through a MACS LS column (Miltenyi Biotec). Purity was confirmed by flow cytometry and immunocytochemistry (ICC) and exceeded 95%.

When sorting for single-cell RNA sequencing (scRNA-seq), spleens were crushed using the rough ends of microscope slides to maximize cell yield. Subsequently, non-B cells were depleted using the Pan B cell Isolation II Kit (Miltenyi Biotec). Enriched cells were then incubated with viability dye, anti-CD16/32 (Fc Block) and surface flow antibodies, including markers for an exclusion (dump) channel (anti-CD3, anti-Gr1 and anti-CD11c); live Dump^−tdTomato^+ cells were sorted by BD FACSAria.

### Detection of ETC and UPR^mt proteins with spectral flow cytometry

A complete list of antibodies is shown in the antibody table (see the spectral flow cytometry antibody panel section). All antibodies targeting intracellular mitochondrial proteins were directly conjugated and of mouse origin, with the exception of rabbit anti-Tfam monoclonal antibody (Abcam). Goat anti-rabbit AF405 Plus secondary antibody (Thermo Fisher Scientific) was used for detection. All antibodies used in the panel had been either validated for flow cytometry by the vendor or in the literature. Mouse anti-mitochondrial complex 1 antibody was conjugated in house using a Lightning-link PE-Cy7 Antibody Labeling kit (catalog no. 762-0005, Novus Biologicals). Cells were labeled with Zombie NIR viability dye (BioLegend) and Fc Block in PBS for 30 min on ice in 96-well V-bottom plates (Sarstedt). After washing, surface staining was performed in Brilliant stain buffer (catalog no. 563794, BD Biosciences) and FACS buffer (at a 1:1 ratio) for 30 min. Cells were then fixed in 4% freshly-made PFA at 20 °C for 15 min and permeabilized in freezer-cold methanol for 10 min with occasional vortexing. Anti-Tfam primary staining was performed in 50 μl FACS buffer supplemented with 2% goat serum for 45 min at 20 °C. After two washes, goat anti-rabbit secondary antibody and the remaining directly conjugated antibodies for ETC and UPR^mt proteins were added for 30 min at 20 °C. The cells were subsequently washed in FACS buffer and acquired on a

Cytek Aurora (5-laser) spectral flow cytometer. Exploratory pilot experiments were performed to determine the most suitable single-stained references (beads or cells) for individual marker–fluorochrome combinations. If cells were used as reference controls, they were obtained from matching organs (spleen or bone marrow). Reference controls were processed similarly to fully stained samples, with parallel fixation, permeabilization and washing steps. Acquired samples were unmixed using SpectroFlo and analyzed with FlowJo software. The 'Autofluorescence (AF) as a fluorescent tag' option was enabled during unmixing to minimize AF interference. In rare cases, minor adjustments were applied to unmixing on the SpectroFlo software. Geometric mean fluorescence intensity (gMFI) values for ETC and UPR$^{mt}$ proteins were calculated using FlowJo.

### Single-mitochondrion translation assay

The technique was performed as described previously but with modifications[56]. Total B cells were isolated from the spleens of Aicda-WT and Aicda-Tfam mice immunized with NP-CGG using magnetic beads (Pan B cell Isolation Kit II). Experiments were performed in batches so that one Aicda-WT and one Aicda-Tfam mouse spleen were processed in each replicate. Cells were pulsed with OPP (20 μM, catalog no. NU-931-5, Jena Bioscience) in complete RPMI 1640 supplemented with the cytoplasmic translation inhibitor harringtonine (1 μg ml$^{-1}$, catalog no. ab141941-5mg, Abcam) for 45 min at 37 °C. After a wash with complete medium, cells were resuspended in ice-cold mitochondria isolation buffer (1×, 320 mM sucrose, 2 mM EGTA, 10 mM Tris-HCl, at pH 7.2 in water, prepared as 2× stock and stored in aliquots at −20 °C). Cells were then homogenized with a Dounce homogenizer with a 2-ml reservoir capacity (catalog no. ab110171, Abcam), using ten strokes with a type B pretzel. The homogenizer was rinsed with distilled water before each sample was processed to avoid cross-contamination. Homogenates underwent differential centrifugation at 4 °C, with intact cells and isolated nuclei first pelleted at 1,000$g$ for 8 min. The supernatant containing the mitochondria was then transferred into new microcentrifuge tubes and centrifuged at 17,000$g$ for 15 min. Enriched mitochondria, which appeared as brown-colored pellets, were fixed in 1% PFA in 0.5 ml PBS on ice for 15 min, followed by a wash with PBS. Permeabilization and subsequent antibody staining was performed in 1× BD Perm/Wash buffer (diluted in MilliQ-filtered water) at 20 °C. Preadsorbed primary rabbit anti-RFP (1:250, catalog no. 600-401-379, Rockland) antibody labeling was performed at 20 °C for 30 min. After a single wash, the Click reaction was performed using the Click-iT Plus OPP AF647 Protein Synthesis Assay Kit (catalog no. C10458, Thermo Fisher Scientific) at 20 °C for 30 min. After a wash, mitochondria were resuspended in 1× BD Perm/Wash buffer containing AF488-conjugated anti-COXIV (1:100, catalog no. 66110-1-Ig, Proteintech) and PE-conjugated donkey anti-rabbit IgG (1:200, catalog no. 406421, BioLegend). After washing with Perm/Wash buffer, mitochondria were resuspended in 250 μl filtered (0.2 μm) PBS and acquired using a BD Fortessa X-20 flow cytometer. The threshold for SSC-A (log-scale) was set to the minimum value (20,000) to allow acquisition of subcellular particles. Submicron Particle Size Reference Beads (catalog no. F13839, Thermo Fisher Scientific) were also used to identify mitochondria. For analysis, mitochondria were identified based on the COXIV and SSC-A properties; RFP$^+$ (mitochondria with AP-GC origin) and RFP$^-$ (mitochondria from naive B cells) were gated based on the anti-RFP antibody signal. The gMFI for the OPP-AF647 fluorescence was calculated for the RFP$^+$ and RFP$^-$ mitochondrial subsets and their ratio was used as a translation index and pooled after batch correction. Mitochondria from reporter-free cells, and those untreated with OPP, served as negative controls. In the validation experiments, 300 μg ml$^{-1}$ CHL or ethanol vehicle was used to inhibit mitochondrial translation along with the cytoplasmic translation inhibitor harringtonine; mitochondria were detected by COXIV-AF488 (1:100, catalog no. 66110-1-Ig, Proteintech) and SDHA-AF594 antibodies (1:250, catalog no. ab170172, Abcam) as described above.

### Immunohistochemistry (IHC)

Collected spleens were immediately transferred to Antigenfix (Diapath) solution in 1.5 ml and fixed overnight at 4 °C. The next day, spleens were washed in PBS followed by overnight incubation in 30% sucrose (in PBS) at 4 °C for cryoprotection. On the following day, spleens were snap-frozen in 100% methanol on dry ice and stored at −80 °C until cryosectioning at 8–12-μm thickness. Slides were then rehydrated in PBS at 20 °C, then permeabilized and blocked in PBS containing 0.1% Tween-20, 10% goat serum and 10% rat serum at 20 °C for 2 h. For panels requiring intracellular detection, Tween-20 was replaced by 0.5% Triton X-100. All primary antibody staining was performed overnight at 4 °C in PBS supplemented with 2% goat serum and 0.1% Triton X-100 (intracellular) or Tween-20 (surface); secondary staining was performed at 20 °C for 2 h the next day in PBS 0.05% Tween-20. TUNEL imaging was performed using the Click-iT Plus TUNEL In Situ Imaging Far Red kit according to the manufacturer's protocol (catalog no. C10619, Thermo Fisher Scientific). Slides were mounted with Fluoromount G (catalog no. 0100-01, Southern Biotech) and imaged with a ZEISS LSM 980 equipped with an Airyscan 2 module. See the relevant image analysis section in Supplementary Methods.

### ICC

Isolated cells were transferred onto 18-mm coverslips coated with 0.01% (weight/volume) poly-L-lysine (catalog no. P8920-100ML, 10× stock, Merck) and incubated for 10 min at 37 °C to ensure cell attachment. Cells were then fixed in prewarmed 1× PHEM buffer (60 mM PIPES, 25 mM HEPES, 10 mM EGTA and 4 mM MgSO$_4$·7H$_2$O, pH 6.8) with 4% PFA for 10 min at 37 °C, followed by permeabilization and blocking in 0.2% Triton X-100 with 10% goat serum for 30 min. Primary antibody labeling was performed overnight at 4 °C; secondary antibody staining was performed for 45 min at 20 °C (see antibody table). For the mitochondrial transcription assay based on 5-EU incorporation, isolated untouched naive B cells and GC B cells were resuspended in complete RPMI 1640 supplemented with 1 mM 5-EU (catalog no. C10330, Thermo Fisher Scientific) and transferred to 18-mm coverslips coated with poly-L-lysine. After incubation for 45 min, cells were briefly washed and fixed in warm 4% PFA diluted in PHEM buffer. After permeabilization and blocking for 30 min, incorporated 5-EU was detected by the Click-iT RNA AF594 Imaging Kit (catalog no. C10330, Thermo Fisher Scientific). Intracellular antibody labeling was performed after the Click reaction to minimize the interference of Click reagents with fluorochromes. For in vivo measurement of mitochondrial RNA synthesis, 2 mg 5-EU (Thermo Fisher Scientific) was injected intraperitoneally on day 12 after SRBC immunization and similar preparation and labeling steps described for the ex vivo 5-EU assay were followed. DAPI (catalog no. D8417-1MG, Sigma-Aldrich) staining was performed at 1 μM at 20 °C for 5 min, followed by a brief wash and mounting in Fluoromount G. For MitoTracker staining, cells were labeled with MitoTracker Red CMX ROS (150 nM, catalog no. M7512, Thermo Fisher Scientific). Slides were imaged with a ZEISS LSM 980 equipped with an Airyscan 2 module. See the relevant image analysis section in Supplementary Methods.

### iGB culture system

The iGB culture system was described previously by Nojima et al.[38]. Briefly, the 3T3 fibroblast cell line of BALB/c origin stably expressing CD40 ligand and B cell activating factor (BAFF) (40LB cell line), was cultured and maintained in high-glucose DMEM with GlutaMAX (catalog no. 31966021, Thermo Fisher Scientific) medium supplemented with 10% FCS and 50 U ml$^{-1}$ penicillin/streptomycin (catalog no. 15070063, Thermo Fisher Scientific) in T75 tissue culture flasks (catalog no. 658175, Sarstedt). Once cells were confluent, they were detached using trypsin/EDTA (catalog no. 25200056, Gibco) treatment, washed and collected in 15-ml tubes in 5 ml medium and irradiated (80 Gy). After irradiation, cells were washed, counted and seeded at $3 \times 10^6$ per dish (100 mm, catalog no. G664160, Greiner) or

0.5 × 10⁶ per well (6-well plate, catalog no. 353046, Falcon). Fibroblast attachment and stretching were allowed overnight at 37 °C and 5% $CO_2$. The next day, naive B cells were isolated using anti-CD43 microbeads and treated with TAT-Cre (approximately 1.5 μM or 66.7 U ml⁻¹, catalog no. SCR508, Merck) for 45 min as described in Supplementary Methods. After three washes, cells were counted and cultured on an irradiated 40LB layer at 5 × 10⁵ (100-mm dish) and 5 × 10⁴ (per well, 6-well plate) for 4–6 days.

### iGB adoptive transfer

iGB cells were generated as above. On day 4, fibroblasts and in vitro-differentiated plasmablasts (generally less than 10% frequency) were removed using biotinylated anti-H-2Kd (catalog no. 116604, BioLegend) and anti-CD138 (catalog no. 142511, BioLegend) followed by anti-biotin microbeads (Miltenyi Biotec); negative selection was performed using LS columns (Miltenyi Biotec). In some experiments, FACS was used to purify tdTomato⁺ iGB cells. For competitive experiments, purified WT iGBs (CD45.1/2⁺) were mixed 1:1 (ratio confirmed by flow cytometry before transfer) with CD45.2⁺ tdTomato⁺ Tfam⁻/⁻ iGBs and injected intravenously (6 × 10⁶ total cells in competitive setting or 3 × 10⁶ cells in noncompetitive setting) into CD45.1⁺ or CD45.2⁺ congenic hosts that were immunized with SRBC according to the enhanced protocol to maximize the recruitment of transferred iGB cells into the GC. On day 6 after iGB adoptive transfer, spleens were collected and analyzed by flow cytometry and confocal imaging to assess GC entry.

### Transwell migration assay

A total of 5 × 10⁵ enriched total B cells isolated from SRBC-immunized Aicda-Tfam and Aicda-WT mice were placed in a 6.5-mm transwell chamber with 5-μm pore size (catalog no. 3421, Corning) and incubated at 37 °C for 2 h in the presence of murine CXCL12 (200 ng ml⁻¹, BioLegend) or CXCL13 (1 μg ml⁻¹, BioLegend) in complete RPMI 1640. The relative chemotaxis/migration index was calculated as follows: percentage of GC B cells (CD38⁻GL-7⁺tdTomato⁺) in migrated live total cells divided by the percentage of GC B cells in total input cells . A total of 2 × 10⁵ purified total B cells from B-Tfam and B-WT mice were placed in a 6.5-mm transwell chamber with 5-μm pore size and incubated for 5 h in the presence or absence of murine CXCL12 (100 ng ml⁻¹, BioLegend) with or without mitoTempo (100 μM, Merck) and Ru265 (30 μM, Merck). After 5 h, cells were incubated with LIVE/DEAD and anti-B220 AF488 antibody and resuspended in 100 μl in 96-well V-bottom plates and acquired on a Cytek Aurora flow cytometer at high-flow setting with a stopping volume of 60 μl. Technical duplicates were also included. Specific migration (%) was calculated according to this formula: 100 × (number of B220⁺ cells migrated in response to CXCL12 − number of B220⁺ cells migrated in the absence of CXCL12)/number of input B cells.

### Lymphoma cell culture system

Daudi cells were cultured in RPMI 1640 medium (pH 7–7.4) supplemented with 10% FCS, 1× GlutaMAX (Gibco), 10 mM HEPES (Gibco) and 50 U ml⁻¹ penicillin/streptomycin and maintained at 37 °C in a humidified incubator with 5% $CO_2$. IMT1 (as a 1 mM stock solution in dimethylsulfoxide (DMSO), catalog no. HY-134539, MedChem Express) was used at 0.1 μM, 1 μM and 10 μM concentrations for a 0–120 h time window. CHL (catalog no. C0378-5G, Merck) was used at 10 μg ml⁻¹ and 25 μg ml⁻¹ concentrations (prepared in 100% ethanol fresh for each culture experiment) for a 0–120 h time window. Cell numbers were determined by manual counting using Trypan blue dye for dead cell exclusion at each time point.

### scRNA-seq analysis

Approximately 17,000 cells per sample were loaded onto the 10X Genomics Chromium Controller (Chip K). Gene expression and BCR sequencing libraries were prepared using the 10X Genomics Single Cell 5′ Reagent Kits v2 (Dual Index) according to the manufacturer's user guide (CG000330 Rev B). The final libraries were diluted to approximately 10 nM for storage. The 10-nM library was denatured and further diluted before loading on the NovaSeq 6000 sequencing platform (v.1.5 chemistry, 28/98 bp paired-end, Illumina) at the Oxford Genomics Centre.

Filtered output matrices from Cellranger v.6.0.1 were loaded in Seurat v.4.1.0. Cells with more than 5% mitochondrial reads and fewer than 200 genes were removed from the analysis. Data were normalized and transformed with SCTransform, with regression of cell cycle phase and mitochondrial reads, and integrated with the FindIntegrationAnchors and IntegrateData functions. The RunUMAP, FindNeighbors and FindClusters functions were used to cluster cells. Marker genes between clusters were identified using the FindAllMarkers function. Two small contaminant clusters (less than 1% of cells) were identified based on the expression of non-B cell genes and were removed from subsequent analyses. Clusters were identified by expression of canonical markers. Cluster proportions were calculated using DittoSeq. For visualization of uniform manifold approximation and projection (UMAP), equal number of cells from each experimental condition were displayed by random downsampling. Differential gene expression between conditions was calculated using the FindMarkers function with the 't-test' parameter. Pathway analysis was performed using the R package single-cell pathway analysis (SCPA). Pseudobulk differential gene expression between individual biological replicates was performed using EdgeR after count aggregation across cells using Scuttle.

Filtered contig outputs generated by Cellranger v.6.0.1 from cells processed in the Seurat workflow above were combined, filtered and visualized using scRepertoire v.1.4. For quantification of mutational load, the Immcantation pipeline was used. Germline segment assignment was performed with Change-O; the SHM count was calculated using SHazaM.

### Statistical analysis

The statistical tests used are indicated in the respective figure legends, with error bars indicating the mean ± s.e.m. $P \leq 0.05$ was deemed to indicate significance. Analyses were performed with Prism 9 (GraphPad Software) or R v.4.1. No statistical methods were used to predetermine sample sizes but our sample sizes are similar to those reported in previous publications[57]. The distribution of data was determined using normality testing to determine appropriate statistical methodology; otherwise, it was assumed to be normally distributed. For the in vivo experiments, we matched the sex and age of the mice in the experimental batches; however, other modes of randomization were not performed. Data collection and analysis were not performed blind to the conditions of the experiments in most of the experiments. Mice with complete absence of GCs and lacking alum spots after immunization were considered as failed intraperitoneal immunization and therefore excluded from the analysis. In selected experiments, batch correction was performed with the R package Batchma.

### Reporting summary

Further information on research design is available in the Nature Portfolio Reporting Summary linked to this article.

## Data availability

The scRNA-seq data have been deposited with the Gene Expression Omnibus under accession no. GSE208021. Source data are provided with this paper.

## Code availability

The code used to analyze the scRNA-seq data is available upon reasonable request and can be found at: https://github.com/alexclarke7/Yazicioglu_et_al.

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

## Acknowledgements

We thank J. Webber for flow sorting, T. Arnon and P. Magill (University of Oxford) for providing the mice, and L. Dustin (University of Oxford) for providing the Daudi cells. We thank D. Kitamura (Tokyo University of Science) for providing the 40LB cell line. We also thank K. Morten (University of Oxford) for helpful suggestions. We thank L. Uhl and G. Pirgova for their assistance and helpful guidance. We also thank the Kennedy Institute BSU staff for their support. Funding for this work was provided by the Wellcome Trust (no. 211072/Z/18/Z) and Cancer Research UK/Versus Arthritis (no. C70663/A29547) to A.J.C., the Kennedy Trust for Rheumatology Research to Y.F.Y. and M.L.D., and the US National Institutes of Health (no. HL118979) to M.L.D. S.J.D. is funded by an National Institute for Health and Care Research (NIHR) Global Research Professorship (no. NIHR300791). For the purpose of open access, the author has applied a CC BY public copyrightlicense to any Author Accepted Manuscript version arising from this submission. Flow cytometry and microscopy facilities were supported by the Kennedy Trust for Rheumatology Research through the Cell Dynamics Platform. We thank the Wolfson Imaging Centre Oxford for providing microscope facility support and the Don Mason flow cytometry facility and staff (R. Hedley and V. Tsioligka) of the Sir William Dunn School of Pathology, University of Oxford. The computational aspects of this research were supported by the Wellcome Trust Core Award grant no. 203141/Z/16/Z and the NIHR Oxford Biomedical Research Centre. The views expressed are those of the authors and not necessarily those of the NHS, NIHR or the Department of Health. The image of a laboratory mouse used was created by Gwilz and distributed under an CC BY-SA 4.0 license.

## Author contributions

Y.F.Y. and A.J.C. conceived and designed the study. Y.F.Y. performed most of the experiments. E.M., E.B.C., S.G., C.S., M. Ali, B.K. and M. Attar performed the experiments. Y.F.Y. and A.J.C. wrote the paper. Y.F.Y. performed the image analysis and A.J.C. and I.G.A.R. analyzed the single-cell data. M.L.D. and S.J.D. provided advice and guidance. A.J.C. supervised the study.

## Competing interests

The authors declare no competing interests.

## Additional information

**Extended data** is available for this paper at https://doi.org/10.1038/s41590-023-01484-3.

**Correspondence and requests for materials** should be addressed to Alexander J. Clarke.

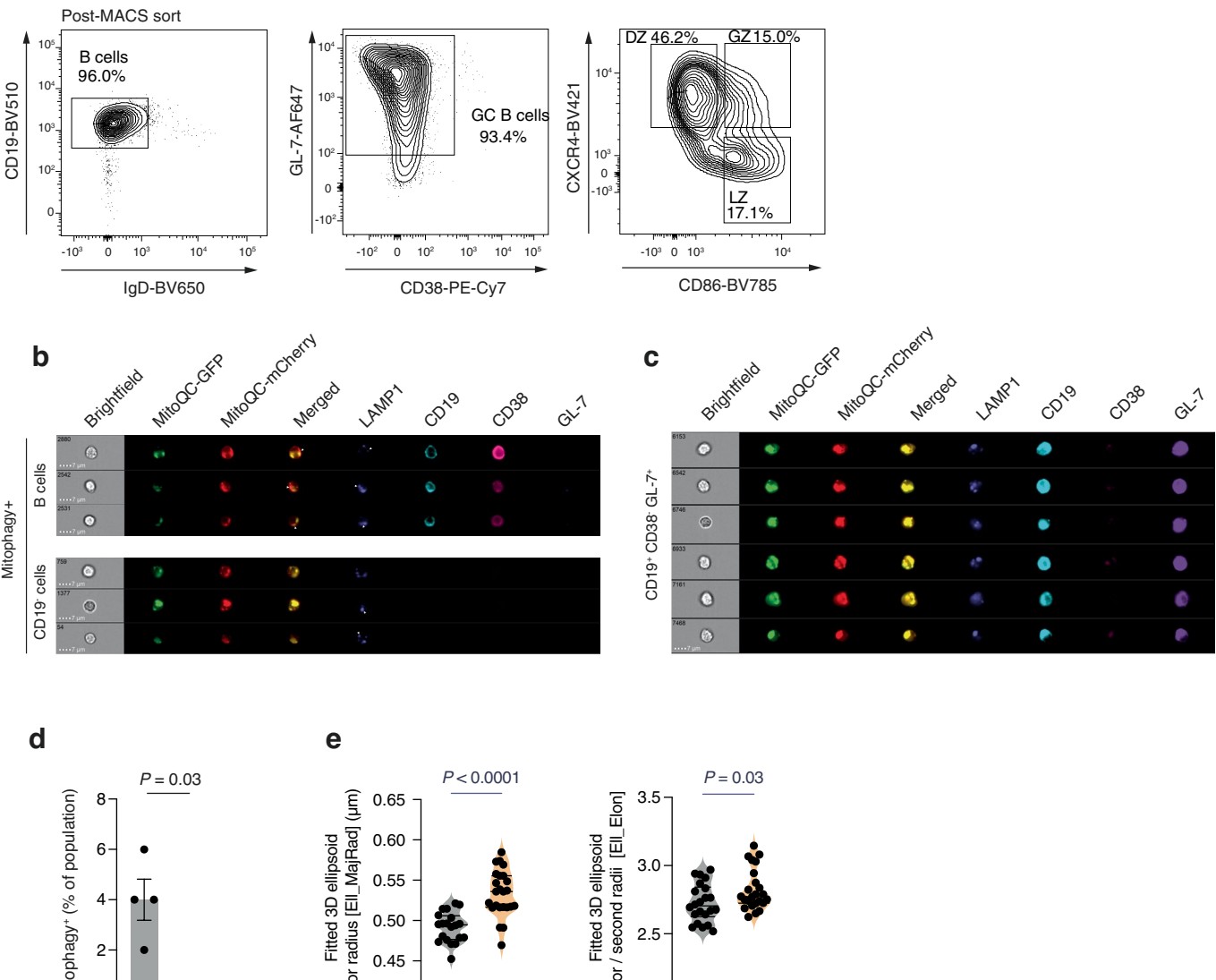

**Extended Data Fig. 1 | GC B cells undergo active mitochondrial remodeling.**
(**a**) Flow sorting strategy for DZ, LZ, and GZ from MACS-enriched GC B cells isolated from SRBC-immunized (enhanced protocol, day 12) Mito-QC mice. (**b**) Representative ImageStream image galleries of splenic CD19⁻ non-B cells and CD19⁺ B cells defined as undergoing mitophagy. Arrows indicate mitophagic foci of lysosomal-associated membrane protein 1 (LAMP1⁺) MitoQC-mCherry without MitoQC-GFP colocalization. (**c**) Representative ImageStream image galleries of splenic GC B cells (CD19⁺CD38⁻GL-7⁺). (**d**) Proportions of mitophagy⁺ population in CD38⁻GL-7⁺ GC B cells and non-GC B cells. Mitophagy was defined

as in **b**. Representative of two independent experiments with n = 4 mice. (**e**) Quantification of average major radius and aspect ratio (major radius/second radius) of mitochondrial nucleoids based on 3D fitted ellipsoid volume model in naïve (n = 20 cells for major radius and n = 22 cells for aspect ratio quantification) and GC B cells (n = 24 cells in both panels). Representative of two independent experiments. Statistical significance was calculated by two-tailed Mann Whitney U (**d**) and unpaired two-tailed t-test with Welch's correction (**e**). Data are presented as the mean ± s.e.m.

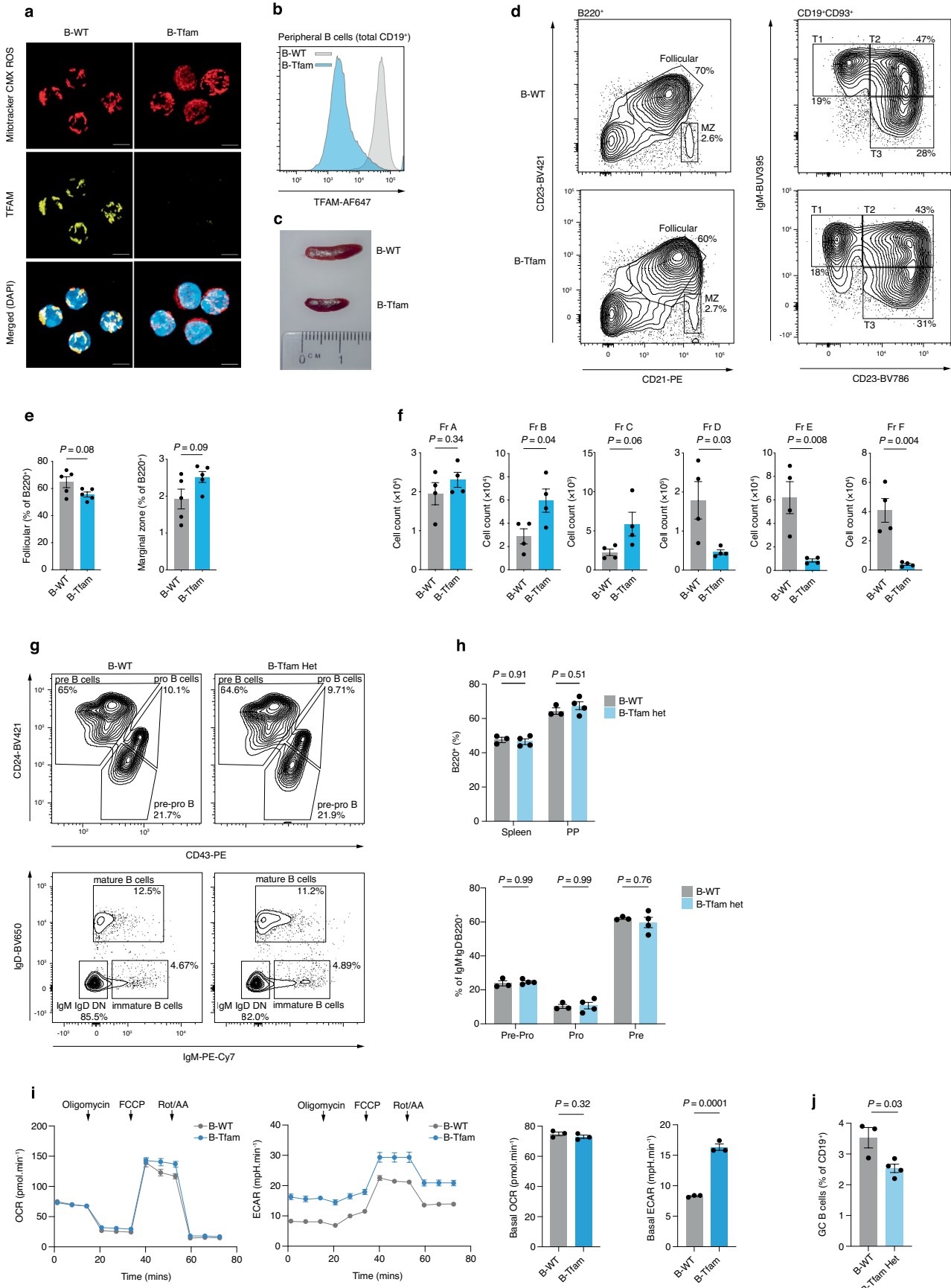

**Extended Data Fig. 2 | See next page for caption.**

**Extended Data Fig. 2 | *Tfam* is essential during B cell development and differentiation. (a)** 3D Airyscan confocal images of B cells from unimmunized B-WT and B-Tfam mice, stained for TFAM and with MitoTracker CMX ROS and DAPI. Scale bar, 3μm. Representative of three independent experiments. **(b)** Representative histogram of TFAM staining by intracellular flow cytometry in splenic CD19$^+$ B cells from unimmunized B-WT and B-Tfam mice. Representative of three independent experiments. **(c)** Representative images of spleens from B-WT and B-Tfam littermate mice. **(d)** Flow cytometry gating strategy for splenic follicular (B220$^+$CD23$^+$CD21$^{int}$) and marginal zone B cells (B220$^+$CD23$^-$CD21$^+$) and representative plots for CD19$^+$CD93$^+$ transitional B subsets (T1,T2, and T3) from B-WT and B-Tfam mice (quantified in Fig. 2a). **(e)** Proportional comparison of splenic follicular and marginal zone B cells from B-WT and B-Tfam mice (n = 5 per group). Representative of three independent experiments. **(f)** Cell counts of bone marrow B cell subsets from B-Tfam and B-WT mice (n = 4 per group) according to Hardy classification (Fr A-F). Representative of two independent experiments. **(g)** Representative flow cytometry plots of bone marrow B cell precursors in B-WT (n = 3) and B-Tfam heterozygous (*Cd79a*-Cre × Tfam$^{loxP/+}$) mice (n = 4). Representative of two independent experiments. **(h)** Proportional comparison of B220$^+$ B cells in spleen, Peyer's patches, and precursor bone marrow B cells from B-WT (n = 3) and B-Tfam heterozygous mice (n = 4). Representative of two independent experiments. **(i)** OCR and ECAR measurements of unstimulated naïve B cells from B-Tfam and B-WT mice and quantification of basal OCR and ECAR values (n = 3 mice per group), representative of two independent experiments. **(j)** Comparison of CD38$^-$GL-7$^+$ GC B cell proportions in spleens of SRBC-immunized B-Tfam Het (n = 4) and B-WT (n = 3) mice. Results representative of two independent experiments. Statistical significance was calculated by unpaired two-tailed t-test (**e,f, i,j**) or two-way ANOVA with Šidák's multiple comparison test (**h**). Data are presented as the mean ± s.e.m.

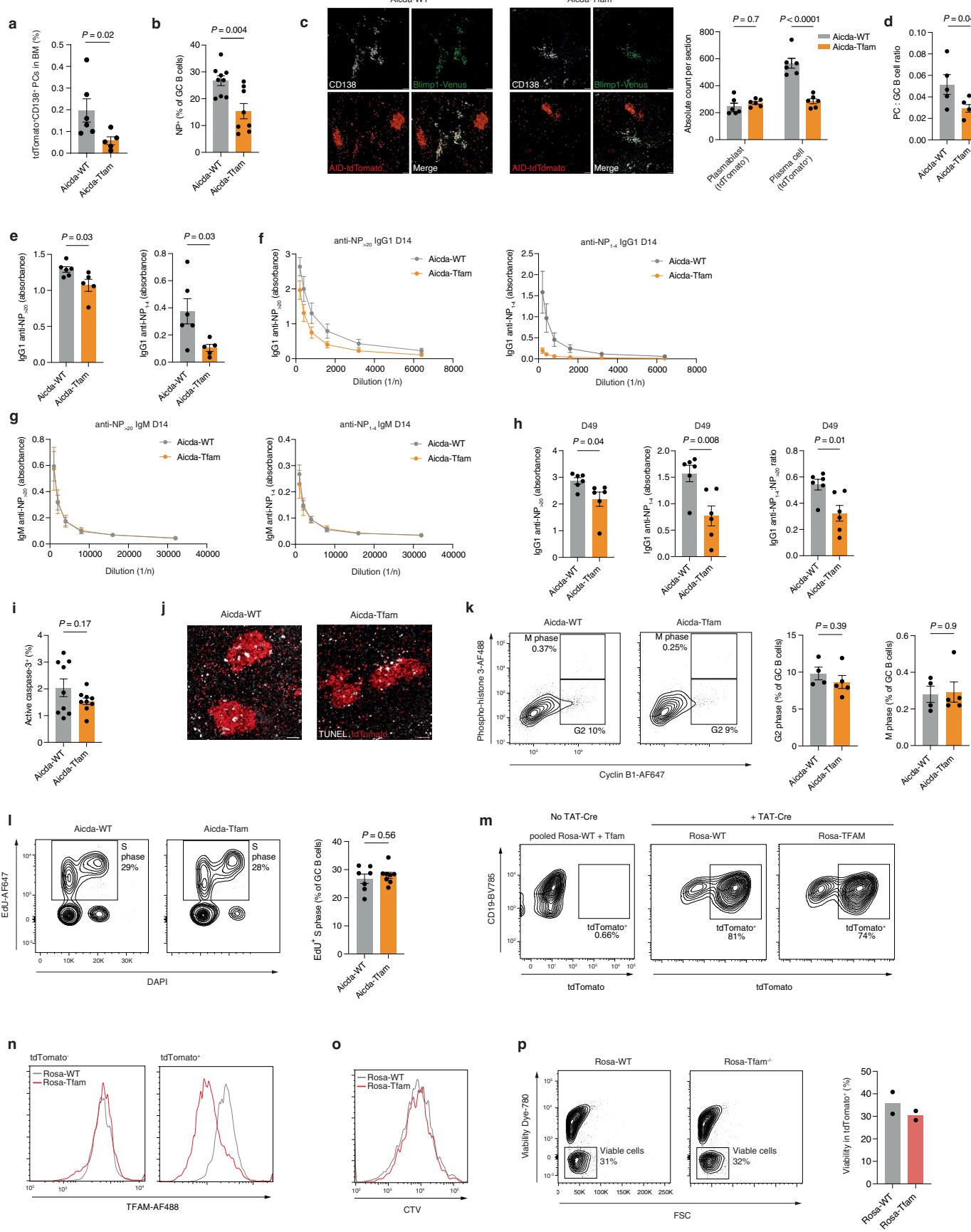

**Extended Data Fig. 3 | See next page for caption.**

**Extended Data Fig. 3 | GC B cells require TFAM. (a)** tdTomato$^+$CD138$^+$ plasma cell percentages within Dump$^-$ bone marrow cells from Aicda-WT (n = 6) and Aicda-Tfam (n = 5) mice at day 12 post SRBC-immunization. **(b)** Proportional comparison of NP-PE or NP-APC-binding GC B cells in NP-CGG-immunized Aicda-WT (n = 9) and Aicda-Tfam (n = 8) mice at day 14. **(c)** Plasma cell clusters in splenic red pulp following NP-CGG immunization. Scale bar, 50μm. Counts of tdTomato$^+$Blimp1-mVenus$^+$CD138$^+$ post-GC plasma cells and tdTomato$^-$Blimp1-mVenus$^+$CD138$^+$ plasmablasts. Results pooled from n = 3 non-serial sections per mouse (n = 2 mice per genotype). **(d)** Ratio of GC B cells (IRF4$^-$CD38$^-$ tdTomato$^+$) to post-GC plasma cells (IRF4$^+$tdTomato$^+$) in Aicda-Tfam (n = 6) and Aicda-WT mice (n = 5). **(e-h)**. ELISA quantifications and dilution curves of IgG1 or IgM anti-NP antibodies (NP$_{1-4}$-BSA and NP$_{>20}$-BSA respectively) in sera from Aicda-Tfam (n = 5) and Aicda-WT mice (n = 6) at day 14 **(e-g)** or day 49 **(h)** (n = 6 per genotype) following NP-CGG immunization. **(i)** Active caspase 3$^+$ apoptotic GC B cell percentages in Aicda-Tfam and Aicda-WT mice (n = 9 per group). **(j)** In situ TUNEL assay on Aicda-WT and Aicda-Tfam spleens following SRBC immunization. Scale bar, 50μm. **(k)** Representative flow cytometry plots and quantification of M and G2 cell cycle stages in GC B cells from Aicda-WT (n = 4) and Aicda-Tfam mice (n = 5). **(l)** Representative flow cytometry plots and quantification of EdU$^+$ GC B cells at S phase from Aicda-WT (n = 7) and Aicda-Tfam mice (n = 8). **(m-p)** Naïve B cells from Rosa26$^{STOP}$tdTomato-WT and Rosa26$^{STOP}$tdTomato-Tfam$^{loxP}$ mice (n = 2) were TAT-Cre treated and in vitro-stimulated with anti-IgM + anti-CD40 + IL-4 for four days. Representative flow cytometry plots of tdTomato **(m)**, TFAM **(n)**, and CTV **(o)**, and viability **(p)**. Statistical significance was calculated by unpaired two-tailed t-test **(b,d,e,h,i,k,l)**, two-tailed Mann Whitney U test **(a)**, two-way ANOVA with Šidák's multiple comparison test **(c)**. Data are presented as the mean ± s.e.m. Data representative of ≥2 independent experiments in all cases.

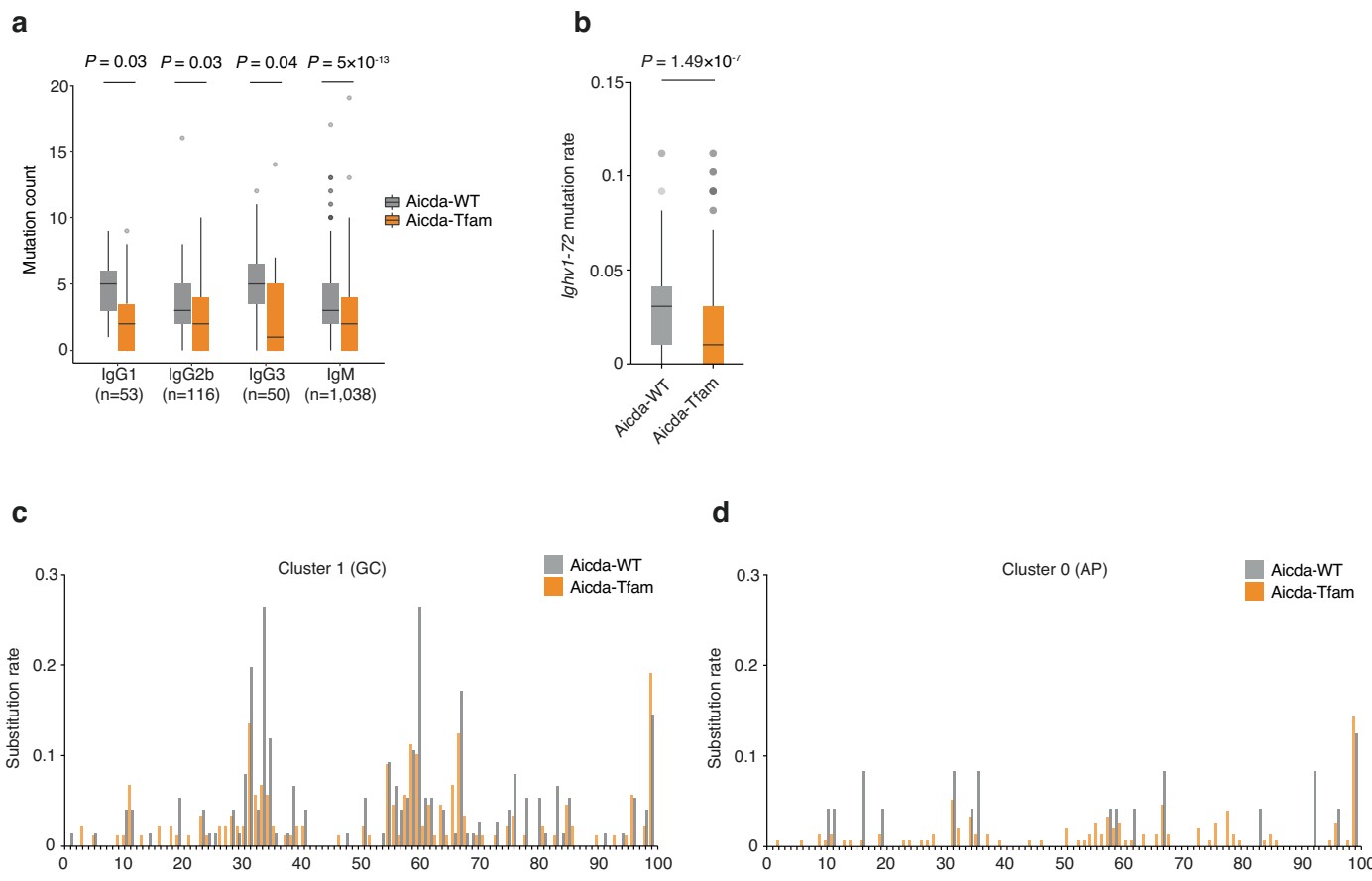

**Extended Data Fig. 4 | TFAM regulates B cell clonality. (a)** Quantification of somatic hypermutation by *Igh* mutation count for indicated immunoglobulin isotype across all sequenced B cells in which isotype call could be made. Data are as described in Fig. 4a (IgG1 = 53 cells, IgG2b = 116 cells, IgG3 = 50 cells, IgM = 1038 cells, pooled from n = 3 Aicda-WT and n = 3 Aicda-Tfam mice). **(b)** Quantification of overall mutation rate for *Ighv1-72* gene segment (n = 76 cells in Aicda-WT, n = 89 in Aicda-Tfam, pooled from n = 3 Aicda-WT and n = 3 Aicda-Tfam mice). **(c)** Amino acid substitution rate across *Ighv1-72* in GC B cell cluster for Aicda-WT and Aicda-Tfam mice (n = 76 cells in Aicda-WT, n = 89 in Aicda-Tfam, pooled

from n = 3 Aicda-WT and n = 3 Aicda-Tfam mice). **(d)** Amino acid substitution rate across *Ighv1-72* in AP B cell cluster for Aicda-WT and Aicda-Tfam mice (n = 24 cells in Aicda-WT, n = 154 in Aicda-Tfam, pooled from n = 3 Aicda-WT and n = 3 Aicda-Tfam mice). Statistical significance was calculated by two-tailed t-test with correction for multiple comparison by the Benjamini-Hochberg method(**a**), or two-tailed unpaired t-test (**b**). In **a**,**b** the box and whisker plots depicts the minimum and maximum values no greater than ±1.5 × the IQR, the lower and upper quartiles, and the median.

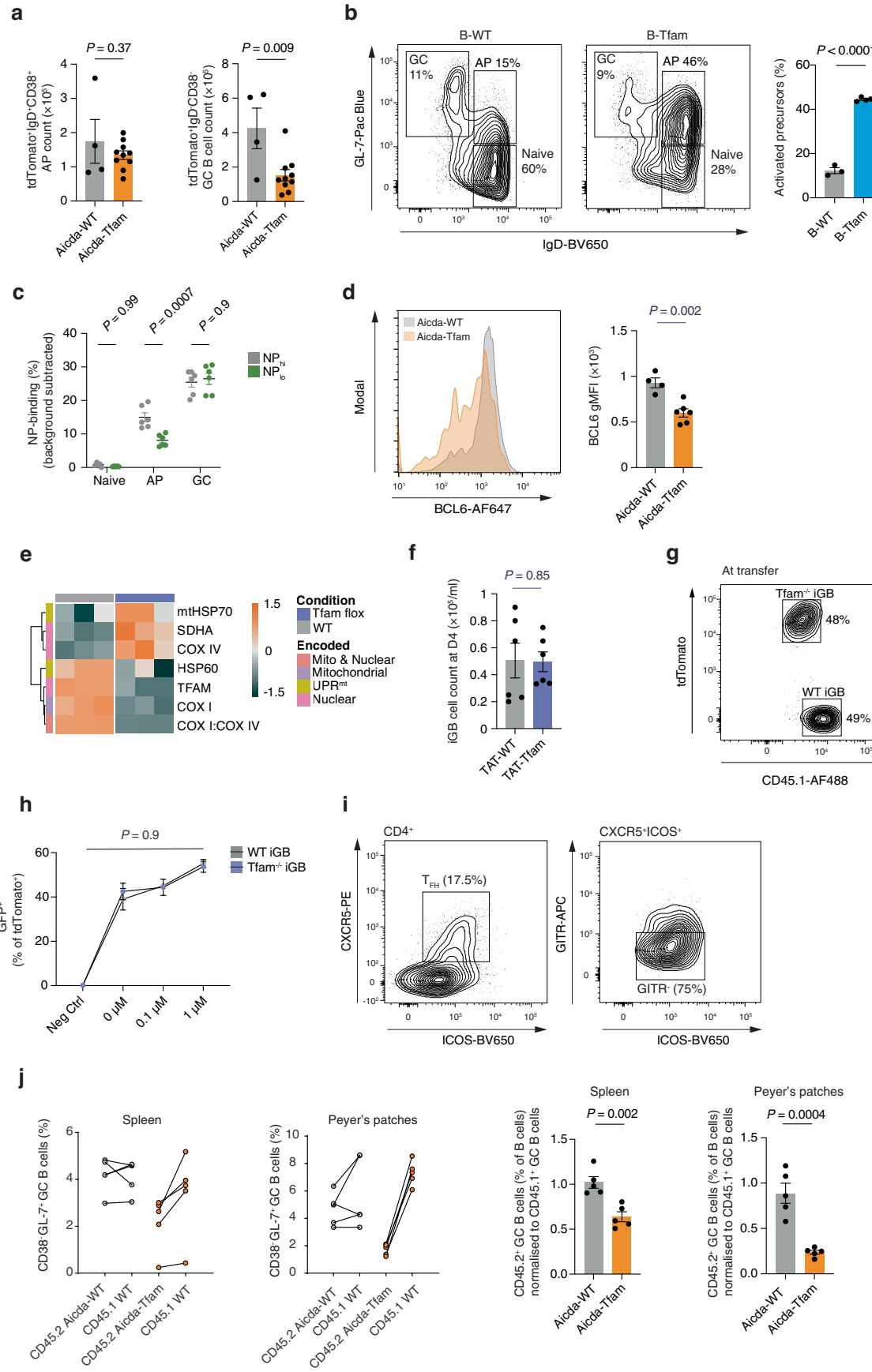

**Extended Data Fig. 5 | See next page for caption.**

**Extended Data Fig. 5 | TFAM is required for GC B cell commitment. (a)** Counts of AP and GC B cells from NP-CGG-immunized Aicda-WT (n = 4) and Aicda-Tfam mice (n = 10). (**b**) Flow cytometry plot and quantification of AP and GC B cell subsets in B-WT (n = 3) and B-Tfam (n = 4) mice immunized with SRBC (enhanced protocol). (**c**) NP-binding rates of naïve B cells, APs, and GC B cells from Aicda-WT (n = 6) to high (NP$_{hi}$) and low (NP$_{lo}$) NP-APC conjugates. (**d**) Quantification of BCL6 expression (gMFI) in GC B cells from Aicda-WT (n = 4) and Aicda-Tfam mice (n = 6). (**e**) Heatmap of row Z-scores for gMFI of indicated mitochondrial proteins, measured by flow cytometry in TAT-Cre treated Tfam$^{-/-}$ or WT iGB cells (n = 3 mice per group). (**f**) Live cell counts of WT and Tfam$^{-/-}$ iGB cells at day 4. Results representative of two independent experiments with n = 6 technical replicates from three mice. (**g**) Pre-transfer tdTomato$^+$Tfam$^{-/-}$ and tdTomato$^-$CD45.1/2$^+$ WT iGB cell ratio in competitive iGB transfer experiment.

(**h**) GFP$^+$ activated OTII-Tg CD4$^+$ T cells were mixed with tdTomato$^+$ WT or Tfam$^{-/-}$ iGBs pulsed with OVA 323-339 peptide. Percentage of GFP$^+$ tdTomato$^+$ doublets indicating T-B conjugates was quantified. Three technical replicates of pooled n = 2 mice shown. Representative of two independent experiments with n = 3 mice per group in total. (**i**) Gating and flow sort strategy for MACS-enriched CD4$^+$ICOS$^+$CXCR5$^+$GITR$^-$ T$_{FH}$ cells. (**j**) Quantification of CD45.2$^+$ GC B cells from spleens and Peyer's patches of Aicda-WT and Aicda-Tfam (n = 5) 50:50 competitive bone marrow chimeras at day 7 following SRBC immunization, normalized to CD45.1 WT GC B cell proportions. Statistical significance was calculated by unpaired two-tailed t-test (**a,b,d,j**), two-tailed Mann Whitney U test (**f**) or two-way ANOVA with Šidák's multiple comparison test (**c,h**). Data are presented as the mean ± s.e.m. Data representative of ≥2 independent experiments in all cases.

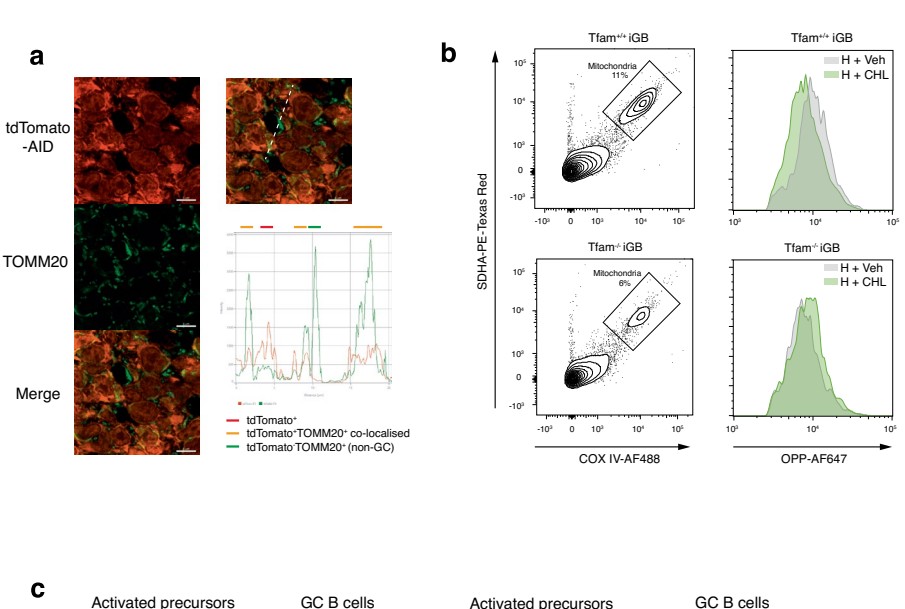

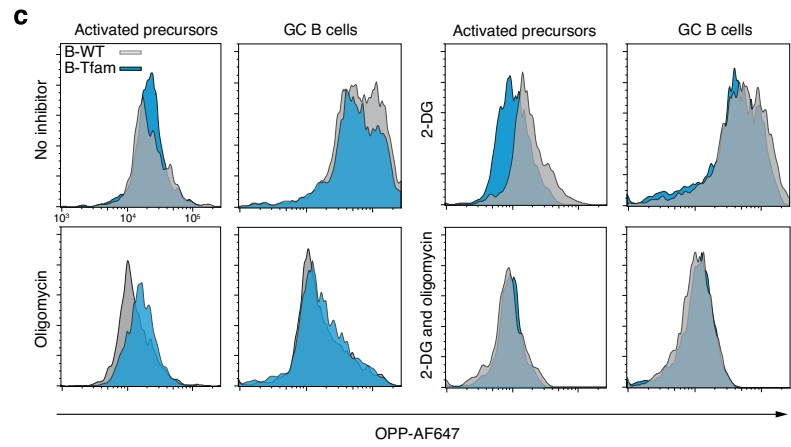

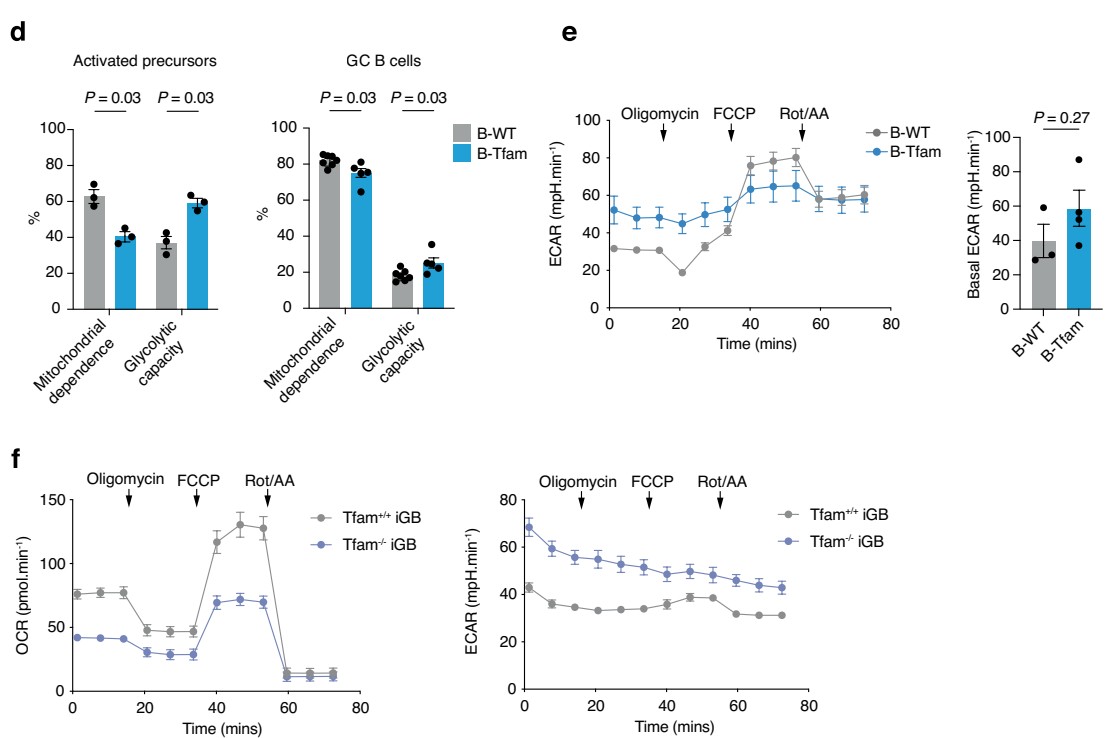

**Extended Data Fig. 6 | See next page for caption.**

**Extended Data Fig. 6 | TFAM regulates mitochondrial translation in activated B cells.** (**a**) Airyscan in situ confocal image and signal intensity chart of GC B cells expressing tdTomato depicting the diffusion of RFP into TOMM20⁺ mitochondria. Scale bar, 5μm. Representative of four independent experiments. (**b**) Mitochondrial OPP incorporation assay performed on WT and Tfam⁻/⁻ iGB cells at day 6. Flow cytometry gating strategy of mitochondria as COX IV⁺ SDHA⁺ particles. OPP-AF647 signal with harringtonine alone (baseline)(H, 1μg/ml) and chloramphenicol (CHL, 300μg/ml) or vehicle (ethanol) treatments depicted in flow cytometry histogram plots. Representative of two independent experiments. (**c**) Flow cytometry histogram plots depicting OPP incorporation in splenic IgD⁺GL-7int AP and IgD⁻CD38⁻GL-7⁺ GC B cells from B-WT and B-Tfam mice in response to metabolic inhibitors (oligomycin and/or 2-DG), shifts in OPP-AF647 signal indicates metabolic properties. Representative of three independent experiments. (**d**) Quantification of mitochondrial dependence and glycolytic capacity of cells based on OPP incorporation in splenic IgD⁺GL-7int AP (n = 3 mice per group) and IgD⁻GL-7⁺CD38⁻ GC B cells from B-WT (n = 7 mice) and B-Tfam mice (n = 5), treated ex vivo with metabolic inhibitors (oligomycin and/or 2-DG). Data pooled from two independent experiments. (**e**) ECAR measurements (MitoStress test) of B-Tfam (n = 4) and B-WT (n = 3 mice) B cells stimulated overnight with anti-CD40 + IL-4. Data pooled from two independent experiments. (**f**) OCR and ECAR measurements (MitoStress test) of 2×10⁵ iGB cells (day 5, after overnight rest in IL-4) from TAT-Cre treated WT (Tfam⁺/⁺) and Tfam^loxp (Tfam⁻/⁻) B cells. Results representative of two independent experiments. Statistical significance was calculated by two-way ANOVA with Šidák's multiple comparison test (**d**) or unpaired two-tailed t-test (**e**). Data are presented as the mean ± s.e.m.

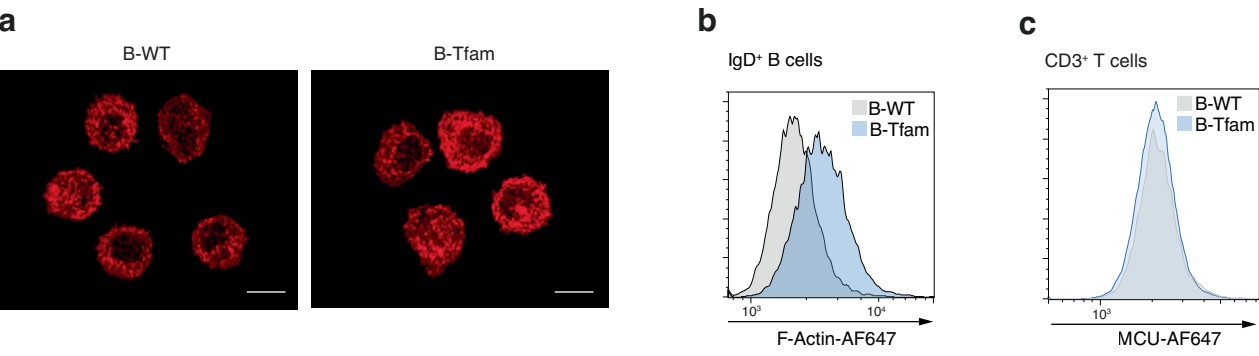

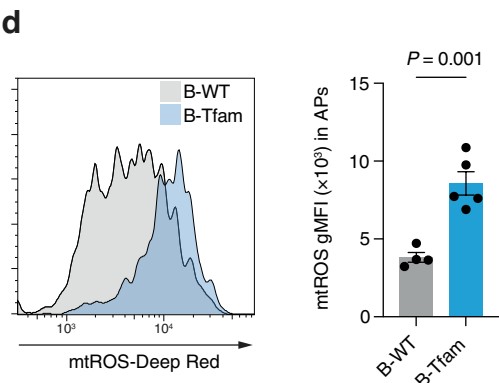

**Extended Data Fig. 7 | *Tfam* deletion disrupts B cell mobility. (a)** 3D Airyscan confocal images of F-actin phalloidin-stained total B cells from unimmunized B-WT and B-Tfam mice. Scale bar, 3 µm. Representative of two independent experiments. (**b**) Representative flow cytometry histogram of F-actin phalloidin fluorescence of IgD[+] B cells from unimmunized B-WT and B-Tfam mice. Representative of two independent experiments. (**c**) Representative flow cytometry histogram of MCU fluorescence of CD3[+] T cells from unimmunized B-WT and B-Tfam mice. Representative of three independent experiments. (**d**) Representative flow cytometry histogram and quantification of mtROS Deep Red fluorescence in IgD[+] GL-7[int] AP cells from immunized B-WT (n = 4) and B-Tfam mice (n = 5). Data representative of two independent experiments. Statistical significance was calculated by unpaired two-tailed t-test (**d**). Data are presented as the mean ± s.e.m.

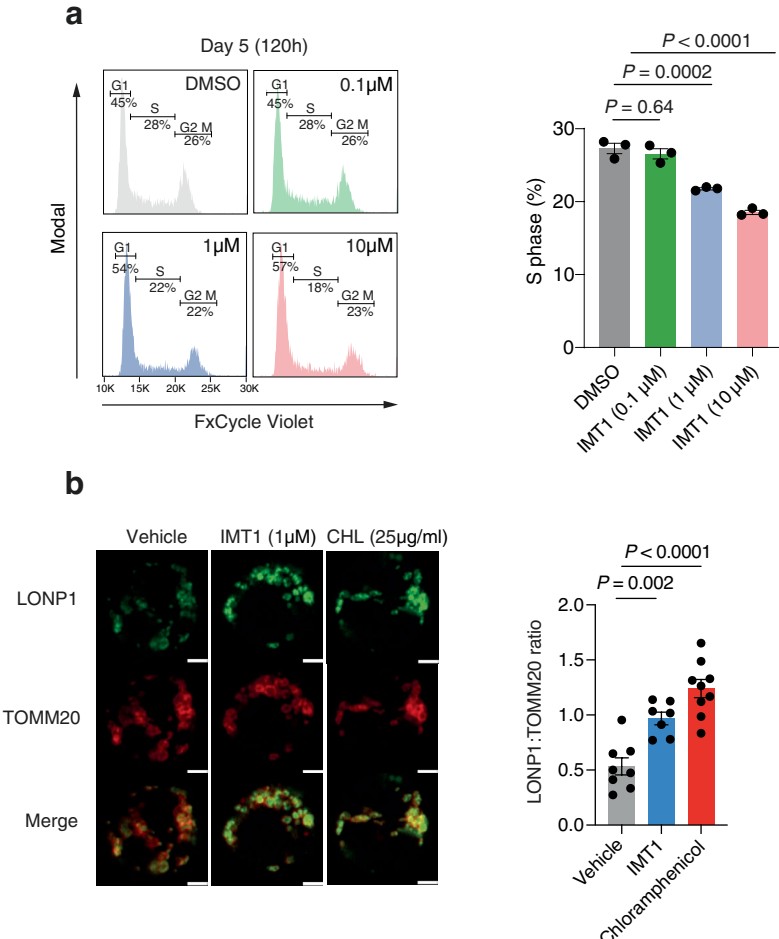

**Extended Data Fig. 8 | *Tfam* deletion in B cells prevents lymphoma. (a)** Flow cytometry-based cell cycle stage characterization (G1, S, G2-M) in Daudi cells at 120h following IMT1 treatment. Quantification of Daudi cells in S phase, representative of two independent experiments with n = 3 technical replicates. (**b**) Representative confocal images of Daudi cells treated with IMT1 (1μM) and CHL (25μg/ml) for 5 days. Quantification of UPR[mt] associated protease LONP1 normalized to mitochondrial mass (TOMM20 signal). Scale bar, 2 μm. Each symbol represents a cell. Vehicle (n = 8 cells), IMT1 (n = 7 cells) and CHL (n = 9 cells). Statistical significance was calculated by ordinary one-way ANOVA with Dunnet's multiple comparisons test (**a,b**). Data are presented as the mean ± s.e.m.

# Reporting Summary

## Statistics

For all statistical analyses, confirm that the following items are present in the figure legend, table legend, main text, or Methods section.

| n/a | Confirmed | |
|---|---|---|
| ☐ | ☒ | The exact sample size (*n*) for each experimental group/condition, given as a discrete number and unit of measurement |
| ☐ | ☒ | A statement on whether measurements were taken from distinct samples or whether the same sample was measured repeatedly |
| ☐ | ☒ | The statistical test(s) used AND whether they are one- or two-sided<br>*Only common tests should be described solely by name; describe more complex techniques in the Methods section.* |
| ☐ | ☒ | A description of all covariates tested |
| ☐ | ☒ | A description of any assumptions or corrections, such as tests of normality and adjustment for multiple comparisons |
| ☐ | ☒ | A full description of the statistical parameters including central tendency (e.g. means) or other basic estimates (e.g. regression coefficient) AND variation (e.g. standard deviation) or associated estimates of uncertainty (e.g. confidence intervals) |
| ☐ | ☒ | For null hypothesis testing, the test statistic (e.g. *F*, *t*, *r*) with confidence intervals, effect sizes, degrees of freedom and *P* value noted<br>*Give P values as exact values whenever suitable.* |
| ☒ | ☐ | For Bayesian analysis, information on the choice of priors and Markov chain Monte Carlo settings |
| ☒ | ☐ | For hierarchical and complex designs, identification of the appropriate level for tests and full reporting of outcomes |
| ☒ | ☐ | Estimates of effect sizes (e.g. Cohen's *d*, Pearson's *r*), indicating how they were calculated |

*Our web collection on statistics for biologists contains articles on many of the points above.*

## Software and code

Policy information about availability of computer code

| Data collection | Zeiss ZEN (blue edition) v3.4 - Zeiss LSM 980 with Airyscan 2 module<br>BD FACS Diva v8.01 - BD LSR II and Fortessa X20 (4-laser) flow cytometer<br>SpectroFlo v3.0 - Cytek Aurora 5-laser flow cytometer<br>INSPIRE v200.1.620.0 - ImageStream<br>SeaHorse Wave Analyzer Software v2.6 - Seahorse. XFe96 Analyzer<br>10x Genomics Cellranger 6.0.1 for single cell analysis |
|---|---|
| Data analysis | Zeiss ZEN (blue edition) v3.4 - Image analysis<br>Flow Jo v10 - Flow cytometry<br>SeaHorse Wave Desktop Software v2.6 - SeaHorse analysis<br>ImageJ  v1.53t - Image analysis<br>Huygens (for deconvolution) v22.04 - Image analysis<br>IDEAS v6.2 and FCS Express v7 - for ImageStream<br>Prism v9.4 for statistical analysis<br>Single cell RNA sequencing data were analyzed using R 4.1.2, with the following packages: Seurat (v4.0.1), DittoSeq (v1.6.0), SCPA (v0.0.0.9), Scuttle (v1.4), EdgeR (v3.36), scRepertoire (1.4), and the Immcantation suite (doi:10.1093/bioinformatics/btu138, doi:10.1093/bioinformatics/btv359) Code used in analyses is available upon reasonable request. |

For manuscripts utilizing custom algorithms or software that are central to the research but not yet described in published literature, software must be made available to editors and reviewers. We strongly encourage code deposition in a community repository (e.g. GitHub). See the Nature Portfolio guidelines for submitting code & software for further information.

## Data

Policy information about availability of data

All manuscripts must include a data availability statement. This statement should provide the following information, where applicable:
- Accession codes, unique identifiers, or web links for publicly available datasets
- A description of any restrictions on data availability
- For clinical datasets or third party data, please ensure that the statement adheres to our policy

Single cell RNA sequencing (gene expression and VDJ) data have been deposited on GEO under accession number GSE208021.

## Human research participants

Policy information about studies involving human research participants and Sex and Gender in Research.

| | |
|---|---|
| Reporting on sex and gender | N/A |
| Population characteristics | N/A |
| Recruitment | N/A |
| Ethics oversight | N/A |

Note that full information on the approval of the study protocol must also be provided in the manuscript.

# Field-specific reporting

Please select the one below that is the best fit for your research. If you are not sure, read the appropriate sections before making your selection.

☒ Life sciences          ☐ Behavioural & social sciences          ☐ Ecological, evolutionary & environmental sciences

For a reference copy of the document with all sections, see nature.com/documents/nr-reporting-summary-flat.pdf

# Life sciences study design

All studies must disclose on these points even when the disclosure is negative.

| | |
|---|---|
| Sample size | No statistical methods were used to pre-determine sample sizes, but our sample sizes are similar to those reported in previous publications. (PMID: 29326381). Effect size for some experiments has been estimated via pilot experiments. The distribution of data was determined using normality testing to determine appropriate statistical methodology, or otherwise assumed to be normally distributed. |
| Data exclusions | Mice with complete absence of germinal centers as well as lack of Alum spots after immunisation were considered as failed intraperitoneal immunisation and therefore excluded from analysis. |
| Replication | All experiments included biological and technical replicates. Each experiment reflects at least two independent replicates. All experiments included in the manuscript were reproducible and representative. |
| Randomization | For in vivo experiments we matched the sex and age of the mice in experimental batches. Care was extra taken to minimise cage effect and litter effect by co-housing control and experimental mice and using littermate controls where it was possible. Randomization measures performed for in vivo experiments were further carried over into in vitro and ex vivo experiments, which were set up with cells isolated from wild type or knock out mice. During sample acquisitions experimental and control samples were run consecutively in alternating fashion. Other modes of randomization was not performed as they were not relevant for our study. |
| Blinding | For some experiments, researchers were blinded, for example where mice were genotyped after the experiment was completed. For monitoring disease development in lymphoma-prone mice, animal facility technicians played a role as blinded observers. Data collection and analysis were not performed blind to the conditions of the experiments in most of the experiments because these experiments were performed by the same researchers and the mouse genotype information needed to know to ensure employment of both wild type and knock out mice in the study. |

# Reporting for specific materials, systems and methods

We require information from authors about some types of materials, experimental systems and methods used in many studies. Here, indicate whether each material, system or method listed is relevant to your study. If you are not sure if a list item applies to your research, read the appropriate section before selecting a response.

## Materials & experimental systems

| n/a | Involved in the study |
|---|---|
| ☐ | ☒ Antibodies |
| ☐ | ☒ Eukaryotic cell lines |
| ☒ | ☐ Palaeontology and archaeology |
| ☐ | ☒ Animals and other organisms |
| ☒ | ☐ Clinical data |
| ☒ | ☐ Dual use research of concern |

## Methods

| n/a | Involved in the study |
|---|---|
| ☒ | ☐ ChIP-seq |
| ☐ | ☒ Flow cytometry |
| ☒ | ☐ MRI-based neuroimaging |

## Antibodies

Antibodies used

Flow cytometry antibodies

Antibody   Clone   Manufacturer   Cat no:
Anti-CD45.2  104  BioLegend  109841
Anti-CD45.1  A20  BioLegend   110718
Anti-CXCR4  REA107  Miltenyi Biotec  130-123-274
Anti-CD19  6D5  Biolegend  115545
Anti-B220  RA3-6B2  Biolegend  103247
Anti-CD38  90  Biolegend  102717
Anti-CD138  281-2  Biolegend  142507
Anti-CD138  281-2  Biolegend  142525
Anti-CD86  GL-1  Biolegend  105031
Anti-CD86  GL-1  Biolegend  105043
Anti-CXCR4  2B11  Thermo Fisher  48-9991-82
Anti-IgD  11-26c.2a  Biolegend  405723
Anti-IgD  11-26c.2a  Biolegend  405721
Anti-IgM  RMM-1  Biolegend  406513
Anti-CD21/35  7E9  Biolegend  123409
Anti-BP-1  6C3  Miltenyi Biotec  130-102-181
Anti-CD23  B3B4  Biolegend  101621
Anti-CD43  S11  Biolegend  143205
Anti-CD24  M1/69  Biolegend  101819
Anti-GL-7  GL7  Biolegend  144605
Anti-GL-7  GL7  Biolegend  144611
Anti-GL-7  GL7  Biolegend  144613
Anti-Gr1  RB6-8C5  Biolegend  108423
Anti-CD3  17A2  Biolegend  100221
Anti-CD4  RM4-5  Biolegend  100553
Anti-CD19  6D5  Biolegend  115537
Anti-CXCR5  L138D7  Biolegend  145503
Anti-IgG1  RMG1-1  Biolegend  406629
Anti-ICOS  C398.4A  Biolegend  313549
Anti-GITR  DTA-1  Biolegend  126311
Anti-cyclin B1  V152  Biolegend  647905
Anti-cleaved caspase 3 (Asp175)  9661  Cell signaling  9661T
Anti-phospho histone 3 (Ser10)  D2C8  Cell signaling  3465S
Anti-SDHB  21A11AE7  Abcam  ab197722
Anti-COX I  1D6E1A8  Abcam  ab154477
Anti-TOMM20  EPR15581-54  Abcam  ab186735
Anti-TFAM  EPR23548-120  Abcam  ab252432
Anti-LAMP1  EPR21026  Abcam  ab237307
Anti-MCU  D2Z3B  Cell Signaling  14997S
Anti-CD16/32 (BD mouse Fc Block)  2.4G2  BD  553142
NP-PE    Biosearch  N-5070-1
NP-APC    in-house  in-house
Goat Anti-Rabbit IgG (H&L) SuperClonal antibody    ThermoFisher  A27034
Goat Anti-Rabbit IgG (H&L) SuperClonal antibody    ThermoFisher  A55055
Donkey anti-rabbit IgG (minimal x-reactivity) Antibody BioLegend  406421

Spectral flow cytometry antibody panel

Antibody   Clone   Manufacturer   Cat no:

Anti-CD19  6D5  BioLegend  115545
Anti-CD21/35  7E9  Biolegend  123421
Anti-CD93  AA4.1  BD Bioscience  741989
Anti-IgD  11-26c.2a  Biolegend  405721
Anti-CD4  RM4-5  Biolegend  100547

Anti-IgM   II/41   BD Bioscience   743329
Anti-CD23   B3B4   BD Bioscience   563988
Anti-B220   RA3-6B2   Biolegend   103247
Anti-CD19   6D5   Biolegend   115543
Anti-CD24   M1/69   BD Bioscience   563450
Anti-BP-1 (CD249)   6C3   BD Bioscience   741743
Anti-CD43   S7   BD Bioscience   747726
Anti-CXCR4   2B11   BD Bioscience   741783
Anti-CD86   GL-1   Biolegend   105043
Anti-CD38   90/CD38 (Ab90)   BD Bioscience   741955
Anti-CD138   281-2   Biolegend   142519
Anti-CD4   RM4-5   Biolegend   100547
Anti-GL-7   GL7   Biolegend   144609
Anti-TFAM   EPR23548-120   Abcam   ab252432
Complex I Monoclonal Antibody   18G12BC2   Invitrogen   43-8800
Anti-Cytochrome C   7H8.2C12   Novus Bio   NB100-56503AF700
Anti-COX I (Complex IV)   1D6E1A8   Abcam   ab198600
Anti-COX IV   3C7D2   Proteintech   CL488-60251
Anti-SDHA   2E3GC12FB2AE2   Abcam   ab170172
Anti-ATP5A1   1B10H3   Proteintech   CL555-66037
Anti-HSP60   2F10E7   Proteintech   CL594-66041
Anti-Grp75/mtHSP70   OTI9F8   Novus Bio   NBP1-47801
Zombie fixable viability dye    Biolegend   423105
Anti-CD16/32 (FcBlock)   2.4G2   BD   553142
Goat anti-Rabbit IgG (H+L) Highly Cross-Adsorbed Secondary Antibody    ThermoFisher   A48254

IHC & ICC

Antibody   Clone    Manufacturer   Cat no:
Anti-CD38   90   Biolegend   102716
Anti-CD21/35   7E9   Biolegend   123423
Anti-CD21/35   7E9   Biolegend   123407
Anti-CD138   281-2   Biolegend   142525
Anti-IgD   11-26c.2a   Biolegend   405707
Anti-GL-7   GL-7   Biolegend   144605
Anti-GL-7   GL-7   Biolegend   144611
Anti-GL-7   GL-7   Biolegend   144613
CD3e Monoclonal Antibody   500A2   eBioscience   14-0033-82
Anti-SDHB   21A11AE7   Abcam   ab197722
Anti-COX I   1D6E1A8   Abcam   ab198600
Anti-TOMM20   EPR15581-54   Abcam   ab186735
Anti-Lonp1   Polyclonal   Atlas   HPA002192
Anti-TFAM   EPR23548-120   Abcam   ab252432
Goat Anti-Hamster IgG (H+L)-0.5 mL    Thermofisher   A-21451
Goat Anti-Rabbit IgG (H&L) SuperClonal antibody     ThermoFisher   A27034
Goat Anti-Rabbit IgG (H&L) SuperClonal antibody     ThermoFisher   A55055

Validation

All antibodies were ordered from commercial vendors, which validated antibodies (Biolegend, BD Bioscience, eBioscience, Miltenyi, Abcam, Cell signalling). When applicable, we performed further validation using knock-out mice or relevant isotype controls. Please refer to the manufacturer's website using the catalog numbers listed above should the validation information of a particular antibody need to be viewed. Further information can be obtained from the vendor's websites:

https://www.miltenyibiotec.com/GB-en/products/macs-antibodies/antibody-validation.html?countryRedirected=1
https://www.bdbiosciences.com/en-gb/products/reagents/flow-cytometry-reagents/research-reagents/quality-and-reproducibility
https://www.biolegend.com/en-us/quality/quality-control
https://www.abcam.com/primary-antibodies/a-guide-to-antibody-validation
https://www.ptglab.com/
https://www.cellsignal.com/learn-and-support/videos-and-webinars/cst-antibody-validation-documentary?gclid=CjwKCAiAmJGgBhAZEiwA1JZolqXP7c01OXJ8shE-lMEmfmQIzVesF22qol99dmdWL4DEEgZACFKj5hoCyLsQAvD_BwE&gclsrc=aw.ds

Below we provided the full antibody list with the reference or validation details from vendor or our lab.

Antibody   Clone   Manufacturer   Validation/Reference
Anti-CD45.2   104   75   BioLegend   PMID: 32103173
Anti-CD45.2   104   100   BioLegend    PMID: 33765443
Anti-CD45.1   A20   100   BioLegend    PMID: 33765443
Anti-CD45.1   A20   100   BioLegend   PMID: 33046889
Anti-CXCR4    REA107   Miltenyi   PMID: 25344471
Anti-CD19   6D5    Biolegend    PMID: 30257198
Anti-B220   RA3-6B2    Biolegend    PMID: 31412246
Anti-CD38   90   Biolegend    PMID: 31810882
Anti-CD138   281-2   Biolegend    PMID: 30538335
Anti-CD138    281-2   Biolegend    PMID: 31618654
Anti-CD86   GL-1   Biolegend    PMID: 34157302

Anti-CD86  GL-1  Biolegend   PMID: 34343496
Anti-CXCR4  2B11  Thermofisher   PMID: 33432228
Anti-IgD  11-26c.2a  Biolegend   PMID: 33010224
Anti-IgD  11-26c.2a  Biolegend   PMID: 29752062
Anti-IgM  RMM-1  Biolegend   PMID: 33278339
Anti-CD21/35  7E9  Biolegend   PMID: 28841417
Anti-BP-1  6C3  Miltenyi Biotec   PMID: 2809203
Anti-CD23  B3B4  Biolegend   PMID: 34214192
Anti-CD43  S11  Biolegend   PMID: 30270123
Anti-CD24  M1/69  Biolegend   PMID: 20720183
Anti-GL-7  GL7  Biolegend   PMID: 29752062
Anti-GL-7  GL7  Biolegend   PMID: 31563464
Anti-GL-7  GL7  Biolegend  PMID: 29221730
Anti-Gr1  RB6-8C5  Biolegend   PMID: 30538335
Anti-CD3  17A2  Biolegend   PMID: 27641500
Anti-CD4  RM4-5  Biolegend   PMID: 30737144
Anti-CD19  6D5  Biolegend   PMID: 30770250
Anti-CXCR5  L138D7  Biolegend   PMID: 34860581
Anti-IgG1  RMG1-1  Biolegend   Application FC - Quality tested (Biolegend)
Anti-ICOS  C398.4A  Biolegend   PMID: 35202565
Anti-GITR  DTA-1  Biolegend   PMID: 30611611
Anti-cyclin B1  V152  Biolegend   PMID: 36189922
Anti-cleaved caspase 3 (Asp175)   9661  Cell signaling   PubMed ID: 36658493
Anti-phospho histone 3 (Ser10)  D2C8  Cell signaling   PMID: 31131319

Anti-SDHB  21A11AE7  Abcam   Knockout validated by Abcam
Anti-COX I  1D6E1A8  Abcam   PMID: 33238133 / further validated in Tfam-KO mice and chloramphenicol treated cells
Anti-TOMM20  EPR15581-54  Abcam   PMID: 33882315
Anti-TFAM  EPR23548-120  Abcam   "Suitable for: IP, Flow Cyt (Intra), WB, ICC/IF (abcam)" / further validated in Tfam KO mice using
ICC, IHC-Fr and flow cytometry / showed strong colocalisation with other mitochondrial markers in WT cells
Anti-LAMP1  EPR21026  Abcam   "Suitable for: ICC/IF" by Abcam (used in imaging flow cytometry experiments)
Anti-MCU  D2Z3B  Cell Signaling   Quality tested for IF by the vendor, further validated in Tfam-KO B cells which shows upregulated
MCU and T cells (WT) from same mice being the negative control

Anti-CD16/32 (BD mouse Fc Block)  2.4G2  BD   PMID: 11709085
NP-PE   Biosearch   Validated on unimmunised tissue/mice
NP-APC   in-house   Validated on unimmunised tissue/mice
Goat Anti-Rabbit IgG (H&L) SuperClonal antibody    ThermoFisher   N/A
Goat Anti-Rabbit IgG (H&L) SuperClonal antibody    ThermoFisher   N/A

Spectral flow cytometry antibody panel

Antibody   Clone   Manufacturer    Validation/Reference

Anti-CD19  6D5  BioLegend   PMID: 30257198
Anti-CD21/35  7E9  Biolegend   PMID: 33296685
Anti-CD93  AA4.1  BD Bioscience   PMID: 11739500
Anti-IgD  11-26c.2a  Biolegend   PMID: 29752062
Anti-CD4  RM4-5  Biolegend   PMID: 29150240
Anti-IgM  II/41  BD Bioscience   PMCID: PMC22519
Anti-CD23  B3B4  BD Bioscience   PMID: 10508270
Anti-B220  RA3-6B2  Biolegend   PMID: 29942093
Anti-CD19  6D5  Biolegend   PMID: 28636954
Anti-CD24  M1/69  BD Bioscience   PMID: 9075925
Anti-BP-1 (CD249)  6C3  BD Bioscience   PMID: 12374812
Anti-CD43  S7  BD Bioscience   PMID: 1827140
Anti-CXCR4  2B11  BD Bioscience   PMID: 9570576
Anti-CD86  GL-1  Biolegend   PMID: 34157302
Anti-CD38  90/CD38 (Ab90)  BD Bioscience   PMID: 9694721
Anti-CD138  281-2  Biolegend   PMID: 32213346
Anti-CD4  RM4-5  Biolegend   PMID: 29150240
Anti-GL-7  GL7  Biolegend   PMID: 34914544
Anti-TFAM  EPR23548-120  Abcam   "Suitable for: IP, Flow Cyt (Intra), WB, ICC/IF (abcam)" / further validated in Tfam KO mice using
ICC, IHC-Fr and flow cytometry / showed strong colocalisation with other mitochondrial markers in WT cells
Complex I Monoclonal Antibody  18G12BC2  Invitrogen   Validated for flow cytometry by the vendor
Anti-Cytochrome C  7H8.2C12  Novus Bio   "Applications WB, Flow, ICC/IF, IHC, IHC-P " from vendor
Anti-COX I (Complex IV)  1D6E1A8  Abcam   Validated in Tfam-KO mice and chloramphenicol treated B cells
Anti-COX IV  3C7D2  Proteintech   PMID: 33607155
Anti-SDHA  2E3GC12FB2AE2  Abcam   "Suitable for: ICC/IF" from abcam / validated using Mito-flow approach in-house
Anti-ATP5A1  1B10H3  Proteintech   Validated by vendor
Anti-HSP60  2F10E7  Proteintech   Validated by vendor
Anti-Grp75/mtHSP70  OTI9F8  Novus Bio   "Applications: WB, Flow, ICC/IF, IHC, IHC-P" from vendor's website
Zombie fixable viability dye   Biolegend
Anti-CD16/32 (FcBlock)  2.4G2  BD   PMID: 11709085
Goat anti-Rabbit IgG (H+L) Highly Cross-Adsorbed Secondary Antibody    ThermoFisher

IHC & ICC

| Antibody | Clone | Manufacturer | Validation/Reference |
|---|---|---|---|
| Anti-CD38 | 90 | Biolegend | Validated by vendor |
| Anti-CD21/35 | 7E9 | Biolegend | Validated by vendor |
| Anti-CD21/35 | 7E9 | Biolegend | Validated in house via confocal imaging of FDC |
| Anti-CD138 | 281-2 | Biolegend | Validated in house via confocal imaging using Blimp-1 mVenus reporter mouse |
| Anti-IgD | 11-26c.2a | Biolegend | Validated by vendor |
| Anti-GL-7 | GL-7 | Biolegend | Validated by vendor |
| Anti-GL-7 | GL-7 | Biolegend | Validated by vendor |
| Anti-GL-7 | GL-7 | Biolegend | Validated in house via confocal imaging using Aicda-tdTom reporter mouse |
| CD3e Monoclonal Antibody | 500A2 | eBioscience | PMID: 31968240 |
| Anti-SDHB | 21A11AE7 | Abcam | Knockout validated by Abcam |
| Anti-COX I | 1D6E1A8 | Abcam | Validated in Tfam-KO mice and chloramphenicol treated B cells |
| Anti-TOMM20 | EPR15581-54 | Abcam | PMID: 33882315 / further validated by the vendor |
| Anti-Lonp1 | Polyclonal | Atlas | Validated in house via TOM20 costaining |
| Anti-TFAM | EPR23548-120 | Abcam | "Suitable for: IP, Flow Cyt (Intra), WB, ICC/IF (abcam)" / further validated in Tfam KO mice using ICC, IHC-Fr and flow cytometry / showed strong colocalisation with other mitochondrial markers in WT cells |

| | | |
|---|---|---|
| Goat Anti-Hamster IgG (H+L)-0.5 mL | | Thermofisher |
| Goat Anti-Rabbit IgG (H&L) SuperClonal antibody | | ThermoFisher |
| Goat Anti-Rabbit IgG (H&L) SuperClonal antibody | | ThermoFisher |

## Eukaryotic cell lines

Policy information about cell lines and Sex and Gender in Research

| | |
|---|---|
| Cell line source(s) | Daudi Cell line (kindly gifted by Lynn Dustin, commercially available at https://www.atcc.org/products/ccl-213), 40LB cell line (A fibroblast cell line, BALB/3T3 A31, stably transfected with expression vectors encoding mouse CD40-ligand (CD40L) and BAFF, kindly gifted by Daisuke Kitamura, commercially available at https://cellbank.brc.riken.jp/cell_bank/CellInfo/?cellNo=RCB5304&lang=En) |
| Authentication | Morphology was assessed by microscopy and presence of B-cell lineage markers (Daudi) and CD40L expression (40LB) by flow cytometry |
| Mycoplasma contamination | The cell lines were not tested for mycoplasma in our lab. |
| Commonly misidentified lines (See ICLAC register) | No commonly misidentified cell lines used in this study |

## Animals and other research organisms

Policy information about studies involving animals; ARRIVE guidelines recommended for reporting animal research, and Sex and Gender in Research

| | |
|---|---|
| Laboratory animals | B6.Cg-Tfamtm1.1Ncdl/J (JAX:026123), B6.C(Cg)-Cd79atm1(cre)Reth/EhobJ (JAX: 020505), B6.129P2-Aicdatm1(cre)Mnz/J (JAX:007770), B6;129S6-Gt(ROSA)26Sortm9(CAG-tdTomato)Hze/J (Ai9, JAX:007905), B6.Cg-Tg(TcraTcrb)425Cbn/J (JAX: 004194) and B6.Cg-Tg(IghMyc)22Bri/J [Eμ-Myc] (JAX: 002728) were purchased from Jackson Laboratories. Tg(Prdm1-Venus)1Sait [Blimp1-mVenus] (MGI:3805969) was a kind gift from Mitinori Saitou (Kyoto University). Gt(ROSA)26Sortm1(CAG-mCherry/GFP)Ganl (MitoQC) was a kind gift from Ian Ganley (University of Dundee). B6.SJL.CD45.1 mice were provided by central breeding facility of the University of Oxford. Male and female mice between the ages of 6-15 weeks were used. Mice were bred and maintained under specific pathogen-free conditions at the Kennedy Institute of Rheumatology, University of Oxford. All procedures and experiments were performed in accordance with the UK Scientific Procedures Act (1986) under a project license authorized by the UK Home Office (PPL number: PP1971784). The mice underwent regular checks to ensure they did not have any pathogenic microorganisms. They were housed in cages that had individual ventilation and were provided with things to stimulate their environment. The temperature was kept between 20-24°C, with a humidity level of 45-65%. They were exposed to a 12-hour cycle of light and darkness (7 am to 7 pm), with a thirty-minute period of dawn and dusk. |
| Wild animals | No wild animals employed in this study. |
| Reporting on sex | Our findings are applicable to both male and female mice and we did not observe gender-effect throughout our study. Therefore our study design did not restrict the usage of particular sex. Please see the randomisation section above for our randomisation strategy. |
| Field-collected samples | The study did not involve any field-collected samples |
| Ethics oversight | Mice were bred and maintained under specific pathogen-free conditions at the Kennedy Institute of Rheumatology, University of Oxford. All procedures and experiments were performed in accordance with the UK Scientific Procedures Act (1986) under a project license authorized by the UK Home Office (PPL number: PP1971784) |

Note that full information on the approval of the study protocol must also be provided in the manuscript.

# Flow Cytometry

## Plots

Confirm that:

☒ The axis labels state the marker and fluorochrome used (e.g. CD4-FITC).

☒ The axis scales are clearly visible. Include numbers along axes only for bottom left plot of group (a 'group' is an analysis of identical markers).

☒ All plots are contour plots with outliers or pseudocolor plots.

☒ A numerical value for number of cells or percentage (with statistics) is provided.

## Methodology

**Sample preparation**

Flow cytometry and imaging flow cytometry

Briefly, harvested spleens were injected with ice cold PBS and mashed through a 70μm strainer (Falcon) or crushed between the frosted ends of microscope slides. For Peyer's patch dissociation, a 40μm strainer (VWR) was used. RBCs were depleted by incubating splenocytes with ACK Lysis Buffer (Gibco) for 3-4 mins at 20°C. Single cell suspensions were incubated with Fixable Viability Dye eFluor™ 780 (eBioscience) in PBS, followed by FcBlock (5 mins) and surface antibodies (30 mins on ice) in FACS buffer (PBS supplemented with 0.5% BSA and 2mM EDTA). For intracellular staining, cells were fixed at 20°C with 4% freshly prepared PFA (Cell Signaling) for 15 mins and permeabilized with methanol (90% ice cold for 10 mins on ice with frequent vortexing) unless specified otherwise. Phalloidin-based F-Actin (Thermofisher, cat no: A30107) staining was performed using BD Perm/Wash reagent (BD, Cat. No. 554723) following 4% PFA fixation. For in vivo cell cycle analysis, 5-ethynyl-2'-deoxyuridine (EdU) (1 mg, Thermofisher, cat no: A10044) was injected intraperitoneally and mice were sacrificed after 2.5 h. Cells were stained for surface markers, fixed and permeabilized then labelled using Click chemistry according to manufacturer's instructions (Click iT Plus EdU Flow cytometry kit, Thermo Fisher, cat: C10634). FxCycle Violet (Thermo Fisher, cat: F10347) reagent was used for cell cycle characterization. For mitochondrial superoxide deep red (mtSOX, Dojindo) uptake, following viability dye and surface staining, cells were resuspended in warm complete RPMI supplemented with mtSOX (10μM) and incubated for 30 mins at 37°C. Cell were washed twice before flow cytometry acquisition. Flow cytometry was performed on BD Fortessa X-20 or LSR II instruments (both BD), or using a Cytek Aurora (5 laser) spectral flow cytometer. For imaging cytometry, single cell suspensions were prepared from spleens of MitoQC mice, incubated with live-dead dye, stained for surface markers, and then fixed with 4% PFA. Washed cells were then resuspended in 50μl FACS Buffer and run on an Amnis ImageStreamX Mark II Imaging Flow Cytometer and analyzed with IDEAS (EMD Millipore) and FCS Express software (v7.12). Flow cytometry data was analyzed using FlowJo (BD).

Detection of ETC/UPRmt proteins with spectral flow cytometry

A complete list of antibodies is shown in the antibody table. All antibodies targeting intracellular mitochondrial proteins were directly conjugated and of mouse origin, with the exception of rabbit anti-Tfam monoclonal antibody (Abcam). Goat anti-mouse AF405 Plus secondary antibody (ThermoFisher) was used for detection. All antibodies used in the panel had been either validated for flow cytometry by the vendor or in the literature. Mouse anti-mitochondrial complex 1 antibody was conjugated in house using a PE-Cy7 Lightning-link kit (Novus Bio, cat: 762-0005). Cells were labelled with Zombie NIR viability dye (Biolegend) and Fcblock in PBS for 30 min on ice in 96-well V-bottom plates. Following washing, surface staining was performed in Brilliant staining buffer (BD, cat: 563794) for 30 min. Cells were then fixed in 4% freshly-made PFA at 20°C for 15 mins and permeabilized in freezer-cold methanol for 10 mins with occasional vortexing. Anti-Tfam primary staining was performed in 50μl FACS Buffer supplemented with 2% goat serum for 45 mins at 20°C. Following two washes, goat anti-rabbit secondary antibody and the remaining directly conjugated antibodies for ETC/UPRmt proteins were added for 30 mins at 20°C. The cells were subsequently washed in FACS buffer and acquired on a Cytek Aurora (5 laser) spectral flow cytometer. Exploratory pilot experiments were performed to determine the most suitable single-stained references (beads or cells) for individual marker-fluorochrome combinations. If cells were used as reference controls, they were obtained from matching organs (spleen or bone marrow). Reference controls were processed similarly to fully stained samples, with parallel fixation, permeabilization, and washing steps. Acquired samples were unmixed using SpectroFlow and analyzed with FlowJo. The 'Autofluorescence (AF) as a fluorescent tag' option was enabled during unmixing to minimize AF interference. In some rare cases minor adjustments were applied to unmixing on the SpectroFlow software. gMFI values for ETC/UPRmt proteins were calculated using FlowJo.

**Instrument**

BD Fortessa X20, BD LSR II, BD Aria, Cytek Aurora (5 laser)

**Software**

FlowJo v10, Spectroflo v3

**Cell population abundance**

Cell sorting

Naïve B cells were isolated using the Pan B cell Isolation II Kit, anti-CD43, and/or anti-CD23 Microbeads (all Miltenyi) according to the manufacturer's instructions. Purity validated by flow cytometry was >90%. Isolation of DZ-LZ-GZ subsets of GC B cells, and TFH cells (CD19- CD4+ CXCR5+ ICOS+ GITR-) was performed via fluorescence-activated cell sorting (FACS).

For some experiments, untouched GC B cells were isolated using a magnetic bead-based protocol as described.55. Briefly, spleens were harvested from SRBC-immunized mice, and single cell suspensions were prepared in ice cold MACS isolation buffer (PBS with 0.5% BSA + 2mM EDTA) followed by ACK RBC lysis (Gibco) for 4 mins at 20°C with occasional mixing every

30s. After washing, cells were labelled with anti-CD43 microbeads (Miltenyi) and biotinylated antibodies against CD38 and CD11c (both eBioscience, clones 90 and N418 respectively), followed by incubation with anti-biotin microbeads, and subsequently run through a MACS LS column (Miltenyi). Purity was confirmed by flow cytometry and immunocytochemistry (ICC) and exceeded 95%.

When sorting for single cell RNA sequencing, spleens were crushed using the rough ends of microscope slides to maximize cell yield. Subsequently, non-B cells were depleted using the Pan B cell Kit II (Miltenyi). Enriched cells were then incubated with viability dye, anti-CD16/32 (FcBlock) and surface flow antibodies, including markers for an exclusion (dump) channel (anti-CD3, anti-Gr1 and anti-CD11c), and live Dump- tdTomato+ cells were sorted by BD Aria FACS. When flow sorting was performed, the purity was carefully monitored and each case it was above 90%.

Gating strategy | ymphocytes were initially gated based on FSC/SSC properties followed by exclusion of doublets. Then, viable cells were gated by excluding Live/Dead+ cells. Further downstream gating strategy was indicated in main and supplementary figures/legends

☒ Tick this box to confirm that a figure exemplifying the gating strategy is provided in the Supplementary Information.

