## [Peer Review File · Nature Immunology]

Peer Review Information

Journal: Nature Immunology

Manuscript Title: Dynamic mitochondrial transcription and translation in B cells control germinal centre entry and lymphomagenesis

Corresponding author name(s): Dr Alexander Clarke

Reviewer Comments & Decisions:

Decision Letter, initial version:
--

16th Aug 2022

Dear Alex,

Thank you for providing a point-by-point response to the referees' comments on your manuscript entitled "Dynamic mitochondrial transcription and translation in B cells control germinal centre entry and lymphomagenesis". As noted previously, while they find your work of considerable potential interest, they have raised a number of substantial concerns that must be addressed. In light of these comments, we cannot accept the current manuscript for publication, but would be very interested in considering a revised version that addresses along the lines proposed in your response.

We invite you to submit a substantially revised manuscript, however please bear in mind that we will be reluctant to approach the referees again in the absence of major revisions.

Specifically, the revision should include new experiments to address:

- (1) Quantitate ETC component proteins in AP, GC, MZ, pro and pre-B cell subsets, and correlate this with TFAM levels.
- (2) Perform Seahorse analyses on B-Tfam follicular B cells and Aicda-Tfam GC B cells.
- (3) Perform in vitro analysis of B cells treated with the mitochondrial transcription inhibitor IMT1 and translation inhibitor chloramphenicol and measure the mitochondrial unfolded protein response-related protease LONP1.
- (4) Measure LONP1/HSP60 protein levels in Tfam-deficient AP and GC B cells.
- (5) Perform ELISA to detect high and low affinity NP-specific IgM and IgG2a antibodies, as well as BCR analysis of W33L and K59 mutations.
- (6) Provide PC quantification following NP-CGG immunization and measure B cell memory after prime-boost immunization.
- (7) Perform adoptive transfer of dye-labeled MD4 B cells treated with inhibitors of either mitochondrial transcription (IMT1) or translation (chloramphenicol) into wild-type recipient mice, and measure the fraction of labeled MD4 cells in the GL-7+ GC compartment.

- (8) Measure mitochondrial translation in Tfam-deficient GC B cells by performing in vivo labelling of nascent protein synthesis using the puromycin analogue OPP and quantifying OPP incorporation.
(9) Examine whether modulation of mitochondrial calcium uniporter activity or mtROS scavengers can rescue the defective cytoskeletal and migratory phenotype.

Additionally, I agree with the referee #2 and think that a plausible additional mechanism might be defective antigen presentation to Tfh cells - might test T cell priming by Tfam-deficient B cells vs WT B cells. Could be in vitro experiments and shown as supplementary data.

Please include the additional textual clarifications as indicated in your response letter.

When you revise your manuscript, please take into account all reviewer and editor comments, please highlight all changes in the manuscript text file in Microsoft Word format.

- * Include a "Response to referees" document detailing, point-by-point, how you addressed each referee comment. If no action was taken to address a point, you must provide a compelling argument. This response will be sent back to the referees along with the revised manuscript.
- * If you have not done so already please begin to revise your manuscript so that it conforms to our Article format instructions at <http://www.nature.com/ni/authors/index.html>. Refer also to any guidelines provided in this letter.
- * Include a revised version of any required reporting checklist. It will be available to referees (and, potentially, statisticians) to aid in their evaluation if the manuscript goes back for peer review. A revised checklist is essential for re-review of the paper.

The Reporting Summary can be found here:

When submitting the revised version of your manuscript, please pay close attention to our [href="https://www.nature.com/nature-portfolio/editorial-policies/image-integrity">Digital Image Integrity Guidelines.](https://www.nature.com/nature-portfolio/editorial-policies/image-integrity) and to the following points below:

Finally, please ensure that you retain unprocessed data and metadata files after publication, ideally archiving data in perpetuity, as these may be requested during the peer review and production

process or after publication if any issues arise.

[REDACTED]

If you wish to submit a suitably revised manuscript we would hope to receive it within 6 months. If you cannot send it within this time, please let us know. We will be happy to consider your revision so long as nothing similar has been accepted for publication at Nature Immunology or published elsewhere.

Nature Immunology is committed to improving transparency in authorship. As part of our efforts in this direction, we are now requesting that all authors identified as 'corresponding author' on published papers create and link their Open Researcher and Contributor Identifier (ORCID) with their account on the Manuscript Tracking System (MTS), prior to acceptance. ORCID helps the scientific community achieve unambiguous attribution of all scholarly contributions. You can create and link your ORCID from the home page of the MTS by clicking on 'Modify my Springer Nature account'. For more information please visit www.springernature.com/orcid.

Thank you for the opportunity to review your work.

Kind regards,

Laurie

Laurie A. Dempsey, Ph.D.
Senior Editor
Nature Immunology
l.dempsey@us.nature.com
ORCID: 0000-0002-3304-796X

Referee expertise:

Referee #1: Lymphocyte metabolism

Referee #2: Lymphocyte metabolism

Referee #3: GC B cells

Reviewers' Comments:

Reviewer #1:

Remarks to the Author:

In this study Yazicioglu et al. show that mitochondrial architecture changes during the GC response and that the mitochondrial transcription factor TFAM is needed for an efficient germinal center response as well as during B cell development. Mitochondria are not only important for energy production and the generation of biosynthetic precursor molecules, but also play various roles in signal transduction and cell fate decisions by controlling calcium homeostasis, ROS production and a variety of other processes. This study not only shows for the first time that TFAM plays crucial roles in B cell biology, but also significantly expands our knowledge on the role of mitochondria in B cell function. This study highlights the various defects that arise after TFAM deletion and could be thus of interest to a broad immunological audience.

The only two concerns in my opinion are:

First: A thorough analysis of when mitochondrial function is impaired after TFAM deletion is missing in the study. After deletion of the TFAM gene, not only does the mRNA and protein that are already present in the cells need to be depleted, but the cells also have to experience a need for producing new mitochondria/new mitochondrial proteins in order to manifest a deficit. It is unclear at what point TFAM-deficient B cells actually experience mitochondria-dependent defects. This information is crucial in order to be able to assess which B cell subsets need intact mitochondria for their survival and function. Providing information on whether ETC protein expression and oxygen consumption are matched with TFAM deficiency is important to understand the observed functional defects. It cannot be ruled out that B cells retain sufficient numbers of intact mitochondria for a certain period of time after TFAM deletion. Moreover, even if specific ETC protein levels are reduced, the cells might find a way to compensate and to maintain normal oxygen consumption. Exemplary for this scenario is the study of Milasta et al. in *Immunity* in 2016 which shows that a dramatic decrease in complex I protein levels does not significantly alter oxygen consumption in B cells and hence B cell function. To fully understand the role of TFAM in B cells it is important to directly measure these parameters and not just to infer from published studies.

Specifically, in figure 2 it would be important to show that marginal zone B cells, follicular B cells, pro and pre B cells in B-TFAM-deficient mice lack TFAM-dependent expression of ETC proteins and show reduced mitochondrial respiration. Similarly in Figure 3, ETC protein expression and mitochondrial respiration should be shown. The observed expansion of activated precursors could simply mean that the defects fully manifest only after a few rounds of proliferation.

The only functional metabolic characterization in this study is shown in figure 5F,G. This figure is difficult to understand as it is poorly explained in the legend. It is not clear to me what is meant by the percentage shown in G. Is this percent OOP staining in comparison to untreated cells? How is glycolytic capacity calculated? The OOP signal is lower in TFAM-deficient cells in 5F, why is the "percentage" in 5G higher? TFAM-deficient GCs still seem to be highly dependent on respiration. This is difficult for interpretation. Are these cells simply escapees? Did the cells somehow manage to not delete TFAM or retain functional mitochondria independently of TFAM? If the translation rate in GCs is dependent on mitochondrial ATP generation, why is it not dependent on TFAM? In any case, as mentioned, the expression of ETC proteins and mitochondrial respiration should be assessed in these cells.

Moreover, vague statements are made about COX1 expression and mitochondrial gene expression.

The authors write:

„We next examined the effect of Tfam deletion on mitochondrial translation in APs. We found significantly reduced COX1 expression in B-Tfam APs, with an increase in the nuclear- encoded enzyme succinate dehydrogenase B (SDHB), which forms part of the ETC complex II, suggesting an attempted compensatory mechanism for impaired translation of mtDNA (figure 5H-I).“

And on page 8.:

„In our transcriptional data, APs, in which Tfam deletion is likely to be recent, display generalised disruption of mitochondrial gene expression, with both up and downregulation. „

This is very confusing and experiments directly assessing the expression of ETC components in both APs and GCs should be performed.

In summary, in order to understand the role of TFAM in B cells it is important to match expression of ETC proteins, mitochondrial respiration and B cell function.

Second: It is not clear, whether the absence of TFAM induces some sort of mitochondrial unfolded protein response or other type of stress. This is important to assess whether mitochondrial function itself is essential for germinal center B cells or rather, that GC B cells are particularly sensitive to mitochondria derived stress.

Minor comments:

- Previous knowledge about TFAM is not sufficiently discussed in the introduction or in the discussion. Similarly, previous studies focusing on B cells and mitochondria such as the recent study from Urbanczyk et al. should be discussed.

- In FigS1.B quantification needed as well as a positive control showing that the mouse model and settings used are actually suited at detecting mitophagy

- In Fig.2A- example stainings for marginal zone and follicular B cells should be shown

- In Fig.2B- total cell numbers for B cell subsets in the bone marrow should be shown

- Fig.2D- Information on how often the experiment was performed is missing

- Fig.3I- The overall apoptosis rate in the GC seems to be quite low. This could be either due to apoptotic cells being quickly removed by macrophages in vivo or due to technical issues. In any case this experiment appears to me misleading. I don't think it truly allows to assess whether TFAM-deficient B cells show normal levels of apoptosis.

- In Fig.S3E-F it is not clear how the cells were stimulated. From the main text it appears the cells were stimulated with anti-IgM, anti-CD40 and IL4 for 4 days. The measured viability of 90% seems to be too high with this type of stimulation. Example stainings should be shown. Since the survival rate is untypically high, this experiment does not appear to me to demonstrate that TFAM is not needed for B cell survival.

- In Fig.5 Total cell numbers should be included not just percentages

- In Fig.5E it is not clear whether these are heterozygous or homozygous mice

- Fig.5G it is unclear what these figures actually show- what is meant by the depicted percentage? Percentage taken from what?

- Figure Legend 5F: The authors write "Data representative of or pooled from two independent experiments." Is this a typing mistake? Is the word "pooled" included by accident?

- On page 8: The authors write: "interestingly, GC B cells themselves have relatively reduced expression of TFAM compared to AP B cells, despite high levels of mitochondrial mass and dilution. „ Im not sure what is meant by dilution.

Reviewer #2:

Remarks to the Author:

Referee report & manuscript assessment for Nature Immunology - 2022

Yazicioglu and colleagues,

Dynamic mitochondrial transcription and translation in B cells control germinal centre entry and lymphomagenesis,
from group of A J Clarke.

For authors and Editors

The germinal center (GC) reaction along with what regulates its formation and dynamics after its coalescence remain topics of hot interest as well as great importance in health and disease. One active thread of recent research, represented in this journals pages and elsewhere, is how pathways or organelles involved in intermediary metabolism affect the GC.

As noted by the authors of this excellent manuscript, prior papers have pointed to gene signatures or metabolic surrogates that suggest increased oxidative metabolism in GC B cells [for instance, PMID: 27501247, 34031613, 32066950], and have even presented some limited evidence that this increase matters in terms of the homeostasis, function, or outputs of GC B cells. The authors sensibly do not to put it so bluntly, but one substantial limitation of prior work has been that a definite distinction between impacts that occur prior to establishment of GC identity and residence versus those that actually arise in GC B cells is lacking. Another, sadly, is that few of the more prominent papers critically analyze how a particular experimental perturbation affects the functional outputs (though one cannot state this categorically or as an absolute as a limitation of high-profile papers on GC dynamics or regulation).

This manuscript makes use of a long-established conditional allele for inactivation of the gene encoding Tfam, a mitochondrial protein akin to the architectural transcription factors in the HMG-box family. Elegant approaches with super-resolution microscopy, reporter and tracking alleles, and several sorts of Cre transgene – most pertinently Aicda-Cre - are used to analyze the impact on GC of having caused changes in mtDNA compaction that in turn are associated with altered mitochondrial morphology and function. A key finding is that GC are impaired – as a population and in terms supporting somatic hypermutation and affinity maturation (Figs 3H, 4K-M). Based on the evidence they present, the authors conclude that the principal basis for this is a defect of activated B cells serving as potential precursors (termed 'AP' in the MS) to the GC population. In vitro evidence is presented to the effect that the Trans-Well migration to CXCL12 and 13 (so, inferentially, the

dynamics driven by CXCR4 and 5) is reduced, potentially providing a basis for reduced GC via attenuated migration of activated precursors ('AP'). Two additional - if under-developed and somewhat unconnected - basic observations stemming from use of mb1-Cre (Cd79a-Cre) are reported in this same manuscript. In brief, lack of Tfam starting at the outset of B lineage specification in the marrow causes (i) a striking lack of mature B cells, probably from a developmental block during the pro to pre-B transition, and (ii) this absence of Tfam completely protects against E μ -Myc-driven lymphoma.

In general, the work is very well executed in what it does, well written and fairly interpreted, albeit with some caveats noted below. Several findings and features were of great interest to a reader, and seemed likely to prove important. The reductions in GC, SHM, and affinity maturation are clear. It has to be noted that Tfam is a system whose knockouts been studied on the order of 25 yr with a substantial body of work on the complex metabolic effects, involving ROS-mediated decreases in ETC complexes (e.g, complex I) with compensation from the MCU stabilization and increased mt[Ca⁺⁺]. Still, one strong positive feature was that the elegant dual-reporting system allowed definition of a dependency on Tfam in generation of what probably for the most part are post-GC plasmablasts - i.e., Prdm1+ cells to all appearances still CD19+, so not fully end-stage PC or LLPC-phenotype) derived from AID+ B cells (presumptively GC-derived). The aberrant calcium regulation is a second very interesting finding, one for which the change in MCU expression is a reasonable basis - though the extent to which the correlative finding might suffice is due at least in part to evidence from other papers and systems [e.g., most recently Nature Communications 13, 2769 (2022)] in which loss of Tfam is mechanistically linked to MCU stabilization]. The reduced in vitro chemotactic response to CXCL12 and -13 is a third key contribution to at least a potential or plausible mechanism, and arguably the most novel finding.

In considering the priority Editors might confer on this manuscript, there were a few - but relatively - few points on what was done so far where the data or their presentation needed to be better. The more nettlesome issue is, what degree of development of at least one relatively definitive mechanism to account for why Tfam was critical in the activated B cell precursors to GC ("AP" state) did this manuscript achieve in its present form?

Technical / data issues:

1. For a reader to have a full and proper sense of findings, not only percentages (as presented here exclusively in almost all cases) but also calculated number of cells are essential - even if one of the two is in the Supplemental Data (though ideally both would be in the main Figure sets). Also, in Fig. 3, D and F, it should be feasible with each mouse to calculate a ratio for GC (Fig. 3D) versus GC-derived plasmablast / PC (Fig 3F) and perform significance testing on the impression that the impact of Tfam loss is greater with respect to the GC to "post-GC PC" than it is for the impact on prevalence of GC B cells.
2. More should be developed or presented in terms of the relative levels of antigen-specific antibodies (that is, Tfam-B-KO vs control) of at least unswitched versus one or more switched isotypes.
 - (a) In focusing on GC-related processes, they mostly present the ratio of high-affinity to NP>20-binding IgG1, but in considering plasmablast/PC outputs from the GC, supplemental Fig. S3C (a marginal or non-difference in NP>20 at only one dilution value) creates a conundrum or disconnect in relation to Figs 3D and 3F. These latter figure panels indicate that outputs of PC are decreased, indeed down relative to the already reduced % of Tfam-null GC B cells.
 - (b) More broadly, for a manuscript set to be about the GC and its outputs, it is somewhat anomalous

or limiting to have only an analysis of d 14 after a primary immunization, rather than priming followed by a later boost, or other measures that analyzed a longer period of GC maintenance and function.

3. As noted, the in vitro findings with chemotaxis to CXCL12 & 13 appear novel, and are a reasonable basis for suggesting that maybe this is a substantial basis for the phenotype. But other GPCR that regulate B cell positioning might be affected, so rather than be left with a suggestion, what direct experimental evidence in vivo would enhance this work at the level associated with this journal. Not necessarily with intravital microscopy with 2-photon confocal imaging (a la Cyster, Nussenzweig, Victora, et alia), the authors should test transfer of labeled "AP" B cells (or in vitro avatars) into immunized recipients to assay if activated B cells with impaired Tfam are less prone to enter GC or are defective in LZ/DZ partitioning.

4. The conclusion put forward by the manuscript title, that a change in mitochondrial translation is responsible for the worse GC entry, if correct, would certainly be a big finding, but it is not adequately supported by the data put forward here. If indeed there were direct evidence that mitochondrial protein translation rates are altered or dynamic, generated using rigorous technical approaches, that would seem highly novel. However, neither the SCENITH approach, whatever its value, nor just flow-stained proteins at steady-state (SDHB; COX1), is strong enough evidence to support the conclusion in the manuscript title and body of text (e.g., end of section 5, bottom of p.6). A number of mitochondrial and ETC proteins or complexes are regulated in part by turnover, which can change according to factors such as mtROS. Certainly, a direct assay of mt translation as affected by Tfam – for instance, less incorporation of fMet? – would strengthen the work. While this reader gets concerned by blithe, over-simplistic acceptance of "whole mouse" interventions that are then over-interpreted at the level of a single cell type, those can be helpful (even at this journal). In particular, the authors' use of chloramphenicol (CAM) with the Daudi cells does point to what is thought to inhibit mitochondrial (and bacterial) translation but not the mammalian translation process. So testing CAM on both mice with Tfam deleted by Aicda-Cre versus WT controls could more adequately support the stated conclusion. At the highest level, the point as stated would call for evidence that reversing the defect experimentally increased GC population and function.

Subjective conceptual issues of priority

1. Until recently, there was essentially no strong direct evidence outside BioRxiv to establish that a specific perturbation of mitochondria in B cells altered Ab responses or for that matter GC physiology. However, the publication of Urbanczyk et al (Mielenz lab) in Cell Reports [June 2022, so before submission of this manuscript] clearly shows that the capacity to replicate mitochondria normally is essential for the GC and Ab responses and detracts from the new deliverable here. That manuscript should definitely be cited and discussed. The underlying mechanisms (for dnTwinkle vs lack of Tfam) may differ for Ab responses – they seem distinct in terms of marrow development. In that sense, the basic finding of this manuscript could still have meaningful impact - especially its identification of a pre-GC function (that is, of "AP" cells progressing to become GC B and then GC-derived plasmablasts).

2. The other subjective issue is the question, how deep do the results go in 'establishing' a new but well-supported mechanism relative to an estimate of the median \pm 1SD for papers in this journal? Although basically quite positive about the findings and the manuscript, the general sense of this reader was that something more is needed.

An example would be, even if stipulating that there were in vivo evidence that the altered chemokine responsiveness caused B cell blasts lacking Tfam to fail to localize to GC or partition normally therein,

how in biochemical terms did the altered mitochondrial function cause this? The defect of calcium flux - modest going on dubious (Fig. 6F) - does not seem sufficient to give a profound chemotactic defect, but if a knockdown / forced reduction of MCU in the Tfam-deficient cells led them to migrate normally into GC, that would be an example of a more robust body of evidence for a coherent mechanism. As it is, since endolysosomal defects may be downstream from loss of Tfam, one can equally posit that the problem really is one in which the pre-GC (and GC) B cells do not participate adequately in cognate helper interactions, i.e., fail to present peptides as efficiently to Tfh cells.

Reviewer #3:

Remarks to the Author:

In this study, Yazicioglu and colleagues showed that the mitochondrial mass and biogenesis was increased in GC B cells relative to naïve B cells. In addition, the genetic deletion of Tfam, an essential gene supporting mitochondrial biogenesis and function, impaired B cell development and GC formation. However, the apoptosis and mitosis of Tfam-deficient GC B cells appeared to be intact in their models. They then found that the frequencies of so-called activated precursors of GC B cells were increased in the Tfam-deficient mice. They also showed that the deletion of Tfam disrupted GC spatial organization, impaired cell mobility, as well as suppressed the development of lymphoma in a c-Myc transgenic model.

In summary, the authors successfully demonstrated a critical role of Tfam gene in B cell development and GC reactions. However, considering that GC B cells are known to be mainly fueled by oxidative phosphorylation (OxPhos) according to recent publications (Weisel et al. Nat Imm 2020; Chen et al. Nat Imm 2021), as well as that Tfam is a such fundamentally important gene for mitochondrial biogenesis, most findings related to the perturbation of Tfam in B cells are fully expected. Therefore, the overall novelty of this study is limited. The proposed mechanism of Tfam-mediated entry of activated GC-precursor B cells is interesting and the most novel aspect of this study, but which haven't been substantiated via direct evidence.

Specific critiques:

(1) Introduction p3: I think this misrepresents the state of the field: "Despite evidence of a central role for mitochondria, their dynamics, function, and regulation in GC B cell biology remains largely unknown." This statement here is in order for the authors to make the claim that the present study is valuable. However, I think the previous publications (Weisel et al. Nat Imm 2020; Chen et al. Nat Imm 2021) already provided quite an amount of evidence showing the dynamic and function of mitochondria in GC B cells. This manuscript should present a more accurate overview of the literature in positioning the present work.

(2) Regarding to the Tfam-mediated entry of activated GC-precursor B cells, the authors provided several evidence based on the imaging of actin cytoskeleton and GC organization and in vitro trans-well assay that are correlated with this argument. It's a challenge to distinguish recently generated GC B cells (GC-precursors) versus previously generated GC B cells by flow cytometry analysis during a polyclonal immune response in WT mice. The author used IgD+ as a marker is not very convincing. To directly demonstrated this claim, the author should employ adoptive cell co-transfer model, e.g. by co-transferring naïve WT and Tfam-ko BCR-transgenic B cells into WT mice followed by the monitoring of the dynamics at different time points, to demonstrate that Tfam deficiency B cells are stuck in the

activated or early GC stage compared with WT B cells. Then It would be ideal to further demonstrate that once GC B cells bypass the entry phase, the deletion of Tfam will not affect GC response by co-transferring mature WT and KO GC B cells, or using inducible Tfam KO system.

(3) It's not clear to me that whether the authors try to claim that the comprised actin cytoskeleton and chemotaxis (Fig. 6) contribute to the impaired GC B cell commitment (Fig. 5), which finally led to the reduced GC reactions (Fig. 3)? If so, in Fig. 6A-E, mature GC B cells, instead of activated GC-precursor B cells described in Fig. 5, manifested the defect in actin cytoskeleton and chemotaxis. If not, then which one is the major mechanism responsible for the compromised GC reactions? Thus, the logic and contents between Fig.5 (GC B cell commitment) and Fig. 6 (actin cytoskeleton and other cellular phenotypes) currently is incoherent. A related question is that does Tfam-deficient precursors manifest any defect in terms of cell cycling or viability?

(4) The authors found that the deficiency of Tfam reduced the numbers of early developing B cells, GC B cells and lymphoma, all of which are highly proliferative. They proposed that the impaired cell fate commitment (and/or cell migration) led to the compromised GC reactions. The key question is that does the similar mechanism, instead of the defect of cell proliferation, contribute to the deficiency of the early developing B cells and lymphoma? Also, whether the disrupted B cell development contribute to the suppression of lymphoma development needs to be addressed.

Dissecting and reconciling the mechanism by Tfam (and mitochondria) in these three biological contexts would add much value to the manuscript and would be of interest to the B cell community.

(5) The knockout efficiency of *Aicda*-cre needs to be carefully confirmed using antibody staining. Do the remaining CD38+ GL-7- cells (Fig. 3D, 3E, 3G) indeed not express TFAM in *Aicda*-Tfam mice? It's possible that the expression of Cre in newly generated GC B cells was not sufficient to efficiently delete flox alleles in the Tfam locus. Also, how about the proliferating state of GC B cells in B-Tfam mice (Fig. 1F)?

(6) The authors need to improve the consistency and quality of the data throughout the manuscript. For example, 1) the increased numbers of TFAM-nucleoid in GC B cells relative to naïve B cells as showed in Fig. 1H are not overserved in Fig. 1G. 2) The distribution and frequencies of pre-proB, pro-B, pre-B and immature B cells of B-WT mice illustrated in Fig. S2B are very different compared to that in Fig. 2B,2C. Therefore, the author claimed that B-Tfam het mice had normal B cell development is skeptical. 3) The frequencies of FO and MZ B cells in B220+ cells in B-Tfam mice were roughly 40% and 6%, respectively, as illustrated in Fig. 2A. What're the other ~50% of B220+ cells then? Representative flow plots showing the gating of MZ and FO B cells should help.

(7) Are B cell development normal in *Aicda*-Tfam mice? The spleen of *Aicda*-Tfam mice appear to be obviously smaller than the controls in Fig. 3B. There are also publications demonstrating AID expression during B cell development.

(8) Fig. S3E: Why not directly use B cells from B-Tfam or *Aicda*-Tfam mice to minimize the additional step of introducing TAT-cre? Also, the total numbers of B cells after 4 days culture should be presented.

(9) Page 4: "(figure S3F-G)" should be "(figure S3E-G)".

(10) Fig. 5C legend: remove "and Aicda-Tfam"

Author Rebuttal to Initial comments

See inserted pdf

Reviewer #1

(Remarks to the Author)

In this study Yazicioglu et al. show that mitochondrial architecture changes during the GC response and that the mitochondrial transcription factor TFAM is needed for an efficient germinal center response as well as during B cell development. Mitochondria are not only important for energy production and the generation of biosynthetic precursor molecules, but also play various roles in signal transduction and cell fate decisions by controlling calcium homeostasis, ROS production and a variety of other processes.

This study not only shows for the first time that TFAM plays crucial roles in B cell biology, but also significantly expands our knowledge on the role of mitochondria in B cell function. This study highlights the various defects that arise after TFAM deletion and could be thus of interest to a broad immunological audience.

The only two concerns in my opinion are:

First: A thorough analysis of when mitochondrial function is impaired after TFAM deletion is missing in the study. After deletion of the TFAM gene, not only does the mRNA and protein that are already present in the cells need to be depleted, but the cells also have to experience a need for producing new mitochondria/new mitochondrial proteins in order to manifest a deficit. It is unclear at what point TFAM-deficient B cells actually experience mitochondria-dependent defects. This information is crucial in order to be able to assess which B cell subsets need intact mitochondria for their survival and function. Providing information on whether ETC protein expression and oxygen consumption are matched with TFAM deficiency is important to understand the observed functional defects. It cannot be ruled out that B cells retain sufficient numbers of intact mitochondria for a certain period of time after TFAM deletion. Moreover, even if specific ETC protein levels are reduced, the cells

might find a way to compensate and to maintain normal oxygen consumption. Exemplary for this scenario is the study of Milasta et al. in *Immunity* in 2016 which shows that a dramatic decrease in complex I protein levels does not significantly alter oxygen consumption in B cells and hence B cell function. To fully understand the role of TFAM in B cells it is important to directly measure these parameters and not just to infer from published studies.

Specifically, in figure 2 it would be important to show that marginal zone B cells, follicular B cells, pro and pre B cells in B-TFAM-deficient mice lack TFAM-dependent expression of ETC proteins and show reduced mitochondrial respiration. Similarly in Figure 3, ETC protein expression and mitochondrial respiration should be shown. The observed expansion of activated precursors could simply mean that the defects fully manifest only after a few rounds of proliferation.

We thank the reviewer for their positive comments and interest in our manuscript. We agree that demonstrating both TFAM and ETC complex component expression across B cell developmental trajectories in the bone marrow, periphery, and GC in the context of *Tfam* deletion is important, and also something which has not been shown before in normal physiology. To do so, we have used high dimensional spectral flow cytometry to characterise expression of representative ETC proteins encoded either in the mitochondria, the nucleus, or complexes containing both. We have also included proteins associated with the mitochondrial unfolded protein response as requested. These data clearly demonstrate deletion of *Tfam*, and subsequently downregulation of mitochondrially-encoded proteins in several differentiation trajectories (bone marrow, unimmunised peripheral B cells, and activated precursor

and GC B cells), occurring ontologically close to Cre expression. We have also shown robust TFAM deletion by day 4 of culture in the iGB system, with its functional consequences evident in downregulation of COX I and a compensatory increase in nuclear-encoded ETC proteins SDHA and COX IV.

These new results are included in revised Figure 2, revised Extended Data Figure 5, and a new Figure 6.

We agree with the reviewer that direct measurement of respiratory capacity and lactate production in Tfam-deficient B cells is important. We have therefore used Seahorse extracellular flux analysis to characterise naïve and stimulated B-Tfam B cells, and iGB B cells in which specific temporal control of Tfam deletion was established using TAT-Cre. We found that in naïve peripheral B cells, loss of Tfam did not affect oxygen consumption rate (OCR), which from our new ETC expression data shown above, may be due to compensation from upregulation of other nuclear encoded ETC proteins. However, the extracellular acidification rate (ECAR) was significantly raised following Tfam deletion. The data is shown below and has been added to Extended Data Fig. 2.

We have also activated B-Tfam and B-WT B cells with anti-CD40 and IL-4 and performed Seahorse analysis. This shows that the ability of Tfam-deficient B cells to compensate for defective respiration cannot be sustained following stimulation. The data is shown below and has been added to Fig. 6 and Extended Data Fig. 6.

Although we would ideally use Seahorse for all the additional cell types mentioned by the reviewer, owing to the focus of the manuscript, and more importantly technical limitations on minimum cell numbers, we were unable to do so given the very marked phenotype seen with *Tfam* deletion. Whilst there are examples in the literature of Seahorse being performed on GC B cells, they are either wild type cells with a transgenic BCR supported with CD40L (Wiesel et al.)(1), or from large simultaneous cohorts (8-10) of treated wild type mice (Biram et al.)(2). Due to the extensive validation of SCENITH in comparison to Seahorse(3), and the clear correlation between these modalities in our own data following *Tfam* deletion, we make the case that in this particular setting the results from these techniques can be considered functionally equivalent.

The only functional metabolic characterization in this study is shown in figure 5F,G. This figure is difficult to understand as it is poorly explained in the legend. It is not clear to me what is meant by the percentage shown in G. Is this percent OOP staining in comparison to untreated cells? How is glycolytic capacity calculated? The OOP signal is lower in TFAM-deficient cells in 5F, why is the “percentage” in 5G higher? TFAM—deficient GCs still seem to be highly dependent on respiration. This is difficult for interpretation. Are these cells simply escapees? Did the cells somehow manage to not delete TFAM or retain functional mitochondria independently of TFAM? If the translation rate in GCs is dependent on mitochondrial ATP generation, why is it not dependent on TFAM? In any case, as mentioned, the expression of ETC proteins and mitochondrial respiration should be assessed in these cells.

We acknowledge that the SCENITH system has not been as well explained in the manuscript as it should be and we have now provided a note in the legend of the revised version to refer to the methods section where detailed information on calculations is provided.

To briefly summarise, the percentages are derived from the ratios between the gMFIs of OPP-Click AF647 of cells treated with vehicle, 2-deoxyglucose (2-DG), oligomycin, or 2-DG and oligomycin. Below is an excerpt from Arguello et al(3), describing example calculations used in their paper.

We would highlight that there is a clear and statistically-significant difference in metabolism in activated precursor and GC B cells as determined by SCENITH, using applicable calculations. SCENITH is an increasingly established technique, with over 110 citations at present e.g. Lopes et al, Nature Immunology 2021(4).

We have adjusted the panel in the new main Figure 6 to now include glucose dependence and fatty acid oxidation which we feel are more representative of the underlying metabolic alterations. Moreover we have also modified the overall message in the work to better reflect the metabolic differences observed in these new experiments, or interpretations of existing data.

Moreover, vague statements are made about COX1 expression and mitochondrial gene expression. The authors write:

„We next examined the effect of Tfam deletion on mitochondrial translation in APs. We found significantly reduced COX1 expression in B-Tfam APs, with an increase in the nuclear-encoded enzyme succinate dehydrogenase B (SDHB), which forms part of the ETC complex II, suggesting an attempted compensatory mechanism for impaired translation of mtDNA (figure 5H-I).“

And on page 8.:

„In our transcriptional data, APs, in which Tfam deletion is likely to be recent, display generalised disruption of mitochondrial gene expression, with both up and downregulation. „ This is very confusing and experiments directly assessing the expression of ETC components in both APs and GCs should be performed.

We apologise for the confusion, and we have either removed or reworded these sentences where the quantification of ETC proteins does not provide additional clarity.

In summary, in order to understand the role of TFAM in B cells it is important to match expression of ETC proteins, mitochondrial respiration and B cell function.

These data have now been provided as described above.

Second: It is not clear, whether the absence of TFAM induces some sort of mitochondrial unfolded protein response or other type of stress. This is important to assess whether mitochondrial function itself is essential for germinal center B cells or rather, that GC B cells are particularly sensitive to mitochondria derived stress.

This is an interesting question and we have included measurement of the UPR^{mt} proteins HSP60 and mtHSP70 in our spectral cytometry panel across B cell subsets (Figure 2 and new Figure 6). We see that they are most upregulated during normal bone marrow B cell development, but also most affected by TFAM loss in the periphery. Following activation and GC commitment, HSP60 and mtHSP70 peak in the GC, but are reduced following TFAM deletion. Why this should occur is interesting, and it may be that loss of Tfam sufficiently reduces the rate of mitochondrial protein translation to alleviate the physiologic UPR^{mt}.

We were unfortunately unable to obtain satisfactory data using anti-LONP1 in normal B cells by flow cytometry, due to the lack of suitable antibodies reactive to mouse tissue and cells.

The situation in Daudi cells (which are of human origin) is much clearer however – inhibition of mitochondrial transcription and translation with IMT1 or chloramphenicol leads to upregulation of LONP1. This data has been added to Extended Data Figure 8.

Minor comments:

- Previous knowledge about TFAM is not sufficiently discussed in the introduction or in the discussion. Similarly, previous studies focusing on B cells and mitochondria such as the recent study from Urbanczyk et al. should be discussed.

We have more fully introduced TFAM and discussed Urbanczyk et al. in the revised version.

- In FigS1.B quantification needed as well as a positive control showing that the mouse model and settings used are actually suited at detecting mitophagy

We have used CCCP treatment to induce mitophagy during experimental optimisation as a positive control. We have also quantified S1B. The data are shown below.

We have improved our image gallery examples of mitophagy, and provided quantification. The data have been added to Extended Data Fig. 1 and are shown below:

- In Fig.2A- example stainings for marginal zone and follicular B cells should be shown

We have now provided representative flow cytometry gating strategies for these populations in Extended Data Fig. 2D.

- In Fig.2B- total cell numbers for B cell subsets in the bone marrow should be shown

We have provided absolute cell counts for B cell subsets in the bone marrow in Extended Data Fig. 2F.

- Fig.2D- Information on how often the experiment was performed is missing

This has been indicated in the revised version.

- Fig.3I- The overall apoptosis rate in the GC seems to be quite low. This could be either due to apoptotic cells being quickly removed by macrophages in vivo or due to technical issues. In any case this experiment appears to me misleading. I don't think it truly allows to assess whether TFAM-deficient B cells show normal levels of apoptosis.

We agree that demonstration of apoptosis rates in this setting can be challenging, and that rapid clearance by tingible body macrophages may be confounding. The rate of apoptosis we detect by active caspase-3 staining is in line with previous reports (e.g.(5,6)). TUNEL staining allows the detection of recently phagocytosed apoptotic cells, and we did not detect differences in Aicda-Tfam mice. We have included a section in the discussion considering the technical challenges in these experiments.

- In Fig.S3E-F it is not clear how the cells were stimulated. From the main text it appears the cells were stimulated with anti-IgM, anti-CD40 and IL4 for 4 days. The measured viability of 90% seems to be too high with this type of stimulation. Example stainings should be shown. Since the survival rate is untypically high, this experiment does not appear to me to demonstrate that TFAM is not needed for B cell survival.

The reviewer's point drew to our attention a labelling error in S3F. The quantification presented is in fact the percentage of tdTomato⁺ cells in the viable population (the opposite). We apologise and this has been corrected. The actual viability is around 30% after 4 days, and was not significantly different between genotypes. We have also included a representative flow cytometry gating strategy in Extended Data Fig. 3P. Please see below.

- In Fig.5 Total cell numbers should be included not just percentages

We have now included absolute cell counts wherever possible. When not provided, this is due to the use of correlative imaging to provide spatial information on splenic tissue from the same biological replicate.

- In Fig.5E it is not clear whether these are heterozygous or homozygous mice

They are homozygous – we have clarified this in the revised version.

- Fig.5G it is unclear what these figures actually show- what is meant by the depicted percentage? Percentage taken from what?

We kindly direct the reviewer to the more detailed description of SCENITH presented above, and in the methods section.

- Figure Legend 5F: The authors write “Data representative of or pooled from two independent experiments.” Is this a typing mistake? Is the word “pooled” included by accident?

This is a typo which has been corrected. Apologies and we thank the reviewer for drawing it to our attention.

- On page 8: The authors write: “interestingly, GC B cells themselves have relatively reduced expression of TFAM compared to AP B cells, despite high levels of mitochondrial mass and dilution. „ Im not sure what is meant by dilution.

By this, we mean the distribution of mitochondrial content between daughter cells following the very rapid division observed in GC B cells.

Reviewer #2

(Remarks to the Author)

Referee report & manuscript assessment for Nature Immunology - 2022

Yazicioglu and colleagues,

Dynamic mitochondrial transcription and translation in B cells control germinal centre entry and lymphomagenesis,
from group of A J Clarke.

For authors and Editors

The germinal center (GC) reaction along with what regulates its formation and dynamics after its coalescence remain topics of hot interest as well as great importance in health and disease. One active thread of recent research, represented in this journal's pages and elsewhere, is how pathways or organelles involved in intermediary metabolism affect the GC.

As noted by the authors of this excellent manuscript, prior papers have pointed to gene signatures or metabolic surrogates that suggest increased oxidative metabolism in GC B cells [for instance, PMID: 27501247, 34031613, 32066950], and have even presented some limited evidence that this increase matters in terms of the homeostasis, function, or outputs of GC B cells. The authors sensibly do not put it so bluntly, but one substantial limitation of prior work has been that a definite distinction between impacts that occur prior to establishment of GC identity and residence versus those that actually arise in GC B cells is lacking. Another, sadly, is that few of the more prominent papers critically analyze how a particular experimental perturbation affects the functional outputs (though one cannot state this categorically or as an absolute as a limitation of high-profile papers on GC dynamics or regulation).

This manuscript makes use of a long-established conditional allele for inactivation of the gene encoding Tfam, a mitochondrial protein akin to the architectural transcription factors in the HMG-box family. Elegant approaches with super-resolution microscopy, reporter and tracking alleles, and several sorts of Cre transgene – most pertinently Aicda-Cre – are used to analyze the impact on GC of having caused changes in mtDNA compaction that in turn are associated with altered mitochondrial morphology and function. A key finding is that GC are impaired – as a population and in terms supporting somatic hypermutation and affinity maturation (Figs 3H, 4K-M). Based on the evidence they present, the authors conclude that the principal basis for this is a defect of activated B cells serving as potential precursors (termed 'AP' in the MS) to the GC population. In vitro evidence is presented to the effect that the Trans-Well migration to CXCL12 and 13 (so, inferentially, the dynamics driven by CXCR4 and 5) is reduced, potentially providing a basis for reduced GC via attenuated migration of activated precursors ('AP'). Two additional – if under-developed and somewhat unconnected – basic observations stemming from use of mb1-Cre (Cd79a-Cre) are reported in this same manuscript. In brief, lack of Tfam starting at the outset of B lineage specification in the marrow causes (i) a striking lack of mature B cells, probably from a developmental block during the pro to pre-B transition, and (ii) this absence of Tfam completely protects against E μ -Myc-driven lymphoma.

In general, the work is very well executed in what it does, well written and fairly interpreted, albeit with some caveats noted below. Several findings and features were of great interest to a reader, and seemed likely to prove important. The reductions in GC, SHM, and affinity maturation are clear. It has to be noted that Tfam is a system whose knockouts have been studied on the order of 25 yr with a substantial body of work on the complex metabolic effects, involving ROS-mediated decreases in ETC complexes (e.g., complex I) with compensation from the MCU stabilization and increased mt[Ca⁺⁺]. Still, one strong positive feature was that the elegant dual-reporting system allowed definition of a dependency on Tfam in generation of what probably for the most part are post-GC plasmablasts – i.e., Prdm1⁺ cells to all appearances still CD19⁺, so not fully end-stage PC or LLPC-phenotype) derived from AID⁺ B cells (presumptively GC-derived). The aberrant calcium regulation is a second very interesting finding, one for which the change in MCU expression is a reasonable basis – though the extent to which the correlative finding might suffice is due at least in part to evidence from other papers and systems [e.g., most recently Nature Communications 13,

2769 (2022)] in which loss of Tfam is mechanistically linked to MCU stabilization]. The reduced in vitro chemotactic response to CXCL12 and -13 is a third key contribution to at least a potential or plausible mechanism, and arguably the most novel finding.

In considering the priority Editors might confer on this manuscript, there were a few – but relatively - few points on what was done so far where the data or their presentation needed to be better. The more nettlesome issue is, what degree of development of at least one relatively definitive mechanism to account for why Tfam was critical in the activated B cell precursors to GC (“AP” state) did this manuscript achieve in its present form?

Technical / data issues:

1. For a reader to have a full and proper sense of findings, not only percentages (as presented here exclusively in almost all cases) but also calculated number of cells are essential – even if one of the two is in the Supplemental Data (though ideally both would be in the main Figure sets). Also, in Fig. 3, D and F, it should be feasible with each mouse to calculate a ratio for GC (Fig. 3D) versus GC-derived plasmablast / PC (Fig 3F) and perform significance testing on the impression that the impact of Tfam loss is greater with respect to the GC to “post-GC PC” than it is for the impact on prevalence of GC B cells.

We thank the reviewer for their positive comments and interest in our manuscript. We have now included absolute cell counts wherever possible. When not provided, this is due to the use of correlative imaging to provide spatial information on splenic tissue from the same biological replicate. We have provided the calculation suggested above in Extended Data Fig 3D. The new data below demonstrates PC:GC B cell ratios and absolute counts of plasmablasts and post-GC plasma cells following NP-CGG immunisation and has been added to Extended Data Fig. 3.

2. More should be developed or presented in terms of the relative levels of antigen-specific antibodies (that is, Tfam-B-KO vs control) of at least unswitched versus one or more switched isotypes.

(a) In focusing on GC-related processes, they mostly present the ratio of high-affinity to NP>20-binding IgG1, but in considering plasmablast/PC outputs from the GC, supplemental Fig. S3C (a marginal or non-difference in NP>20 at only one dilution value) creates a conundrum or disconnect in relation to Figs 3D and 3F. These latter figure panels indicate that outputs of PC are decreased, indeed down relative to the already reduced % of Tfam-null GC B cells.

(b) More broadly, for a manuscript set to be about the GC and its outputs, it is somewhat anomalous or limiting to have only an analysis of d 14 after a primary immunization, rather than priming followed by a later boost, or other measures that analyzed a longer period of

GC maintenance and function.

We have now characterised the antigen-specific immune response in significantly more detail. We have generated dilution curves for anti-IgG1/IgM antibodies against NP-BSA with high and low conjugation ratios. This shows that anti-NP_{>20} antibodies are indeed significantly reduced in *Aicda*-*Tfam* mice, and confirm the previously demonstrated decrease in anti-NP₁₋₄ antibodies. We have also included BCR repertoire data illustrating the frequency of W33L and K59 mutations in the IgHV1-72 segment in GC B cells. The data is shown below and has been added to Fig. 3, Extended Data Fig. 3, and to Fig. 4.

There were no differences in IgM anti-NP antibodies at either high or low conjugation ratios, in keeping with their probable extrafollicular plasmablast origin (*Aicda* not expressed and *Tfam* not deleted). We have also quantified splenic plasma cells (tdTomato⁺Blimp1-Venus⁺CD138⁺) and plasmablasts (tdTomato⁻Blimp1-Venus⁺CD138⁺) following NP-CGG immunisation, which supports this conclusion as shown above (data added to Extended Data Fig. 3).

The reviewer is quite right that assessment of longer term memory responses is important, and we have now included this in Fig. 3. The new data are shown below.

3. As noted, the *in vitro* findings with chemotaxis to CXCL12 & 13 appear novel, and are a reasonable basis for suggesting that maybe this is a substantial basis for the phenotype. But other GPCR that regulate B cell positioning might be affected, so rather than be left with a suggestion, what direct experimental evidence *in vivo* would enhance this work at the level associated with this journal. Not necessarily with intravital microscopy with 2-photon confocal imaging (a la Cyster, Nussenzweig, Victora, et alia), the authors should test transfer of labeled “AP” B cells (or *in vitro* avatars) into immunized recipients to assay if activated B cells with impaired *Tfam* are less prone to enter GC or are defective in LZ/DZ partitioning.

We agree with the reviewer that direct demonstration of failure of activated B cells to enter the GC is an important experiment, and thank them for their suggested methodology. Since AP B cells are too rare to sort and transfer, we have used the iGB culture system developed by Kitamura(7) to generate activated precursor-like B cells and then adoptively transferred them into a primed host(8,9). We were unfortunately unable to assess DZ/LZ partitioning due to the rarity of the transferred cells. The following data has been added to revised Fig. 5:

4. The conclusion put forward by the manuscript title, that a change in mitochondrial translation is responsible for the worse GC entry, if correct, would certainly be a big finding, but it is not adequately supported by the data put forward here. If indeed there were direct evidence that mitochondrial protein translation rates are altered or dynamic, generated using rigorous technical approaches, that would seem highly novel. However, neither the SCENITH approach, whatever its value, nor just flow-stained proteins at steady-state (SDHB; COX1), is strong enough evidence to support the conclusion in the manuscript title and body of text (e.g., end of section 5, bottom of p.6). A number of mitochondrial and ETC proteins or complexes are regulated in part by turnover, which can change according to factors such as mtROS. Certainly, a direct assay of mt translation as affected by Tfam – for instance, less incorporation of fMet? – would strengthen the work.

We agree that directly demonstrating dynamic mitochondrial translation in GC B cells, and its impairment following Tfam deletion, would be an important and novel finding. We have therefore developed and optimised an experimental technique to do so at the single mitochondrion level. In brief, we isolated B cells from Aicda-Tfam and Aicda-WT immunised mice and use OPP incorporation in the presence of the cytoplasmic ribosome protein synthesis inhibitor harringtonine to selectively label mitochondrial proteins. Following cell homogenisation, we then used flow cytometry to identify mitochondria based on size and COX IV expression. Since tdTomato enters mitochondria by diffusion, we were able to identify mitochondria which have expressed Aicda and therefore deleted Tfam. This novel approach allowed us to directly demonstrate increased mitochondrial translation in tdTomato⁺ cells, and show that loss of Tfam leads to its reduction.

The following data has been added to the revised manuscript and related revised (new) Fig. 6:

We agree with the reviewer that COX I and SDHB might have distinct turnover and degradation mechanisms due to their assembly in separate ETC complexes (complexes IV and II, respectively).

The ratio between mitochondrial DNA (mtDNA)-encoded subunits to nuclear DNA-encoded subunits of the same ETC complex reflects the efficiency of mitochondrial translation (10,11). Therefore we have included measurement of the two adjacent subunits of complex IV (mtDNA-encoded subunit COX I and nuclear-encoded subunit COX IV) in our spectral cytometry panel across B cell subsets. The ratio of COX I:COX IV in normal physiology had a dynamic trajectory across B cell subsets in bone marrow and spleen, and peaked at the GC stage. We observed a profound mismatch in this ratio in B cells from B-Tfam mice, in keeping with their reduced mitochondrial OPP incorporation. The following data is presented in Fig. 2 and (new) Fig. 6 in heatmap format.

While this reader gets concerned by blithe, over-simplistic acceptance of “whole mouse” interventions that are then over-interpreted at the level of a single cell type, those can be helpful (even at this journal). In particular, the authors’ use of chloramphenicol (CAM) with the Daudi cells does point to what is thought to inhibit mitochondrial (and bacterial) translation but not the mammalian translation process. So testing CAM on both mice with *Tfam* deleted by *Aicda-Cre* versus WT controls could more adequately support the stated conclusion. At the highest level, the point as stated would call for evidence that reversing the defect experimentally increased GC population and function.

We agree that directly treating mice with chloramphenicol could be supportive of our results, but we would however refer to historical literature (e.g. Weisberger & Daniel, 1969)(12), which demonstrates chloramphenicol or its analogues profoundly suppresses the humoral immune response when administered early after primary challenge. Contemporary work has shown that T cells are highly sensitive to inhibition of mitochondrial translation, and that chloramphenicol is genotoxic due to reactive oxygen species generation (13,14). We would suggest that owing to the extensive off-target effects of chloramphenicol, both on other immune cells and its lack of pharmacological specificity, this experiment would, as the reviewer highlights, not provide sufficient refinement to draw meaningful mechanistic conclusions about *Tfam* deletion in vivo.

However, by incubating WT and *Tfam*^{-/-} iGBs ex vivo with chloramphenicol and using mitochondrial OPP incorporation as a readout, we found that chloramphenicol efficiently reduced translation in WT mitochondria, an effect not seen in those from *Tfam*^{-/-} cells. We have added this data to (new) Extended Figure 6.

Since we achieved mitochondrial translation inhibition through ex vivo chloramphenicol treatment, we have attempted to adoptively transfer in vitro-activated MD4 (HY10 BCR Tg) B cells treated with chloramphenicol and/or IMT1 to congenic hosts challenged with SRBC-HEL, in an experimental approach similar to that described in Fig. 5C. We employed the MD4 BCR-Tg system to achieve more

rapid and efficient entry of transferred cells into the GC. Whilst COX I levels were substantially decreased in treated B cells before adoptive transfer, due to the reversibility of both of these reagents, the sustained suppression of mitochondrial translation could not be achieved in vivo within the time window that would allow reliable measurement of GC entry.

Subjective conceptual issues of priority

1. Until recently, there was essentially no strong direct evidence outside BioRxiv to establish that a specific perturbation of mitochondria in B cells altered Ab responses or for that matter GC physiology. However, the publication of Urbanczyk et al (Mielenz lab) in Cell Reports [June 2022, so before submission of this manuscript] clearly shows that the capacity to replicate mitochondria normally is essential for the GC and Ab responses and detracts from the new deliverable here. That manuscript should definitely be cited and discussed. The underlying mechanisms (for dnTwinkle vs lack of Tfam) may differ for Ab responses – they seem distinct in terms of marrow development. In that sense, the basic finding of this manuscript could still have meaningful impact - especially its identification of a pre-GC function (that is, of “AP” cells progressing to become GC B and then GC-derived plasmablasts).

As also requested by reviewer 1, we have cited and discussed Urbanczyk et al in the revised version. We appreciate that the reviewer highlighted our manuscript as distinct. Our work describes the dynamic regulation of mitochondria and their structural configuration in GC B cells, and in detail mechanistically characterises the effects of interference of this process through deletion of *Tfam*. We observe defective GC formation due to failure of activated precursor B cells to developmentally progress to become GC B cells. The great majority of our experimental work is directly ex vivo, studying bone fide GC B cells and their precursors. Importantly, we demonstrate the essential role of mitochondrial translation in lymphomagenesis, and reveal it to be a potentially novel therapeutic target in the treatment of lymphoma.

2. The other subjective issue is the question, how deep do the results go in ‘establishing’ a new but well-supported mechanism relative to an estimate of the median \pm 1SD for papers in this journal? Although basically quite positive about the findings and the manuscript, the general sense of this reader was that something more is needed.

An example would be, even if stipulating that there were in vivo evidence that the altered chemokine responsiveness caused B cell blasts lacking Tfam to fail to localize to GC or partition normally therein, how in biochemical terms did the altered mitochondrial function cause this? The defect of calcium flux - modest going on dubious (Fig. 6F) - does not seem sufficient to give a profound chemotactic defect, but if a knockdown / forced reduction of MCU in the Tfam-deficient cells led them to migrate normally into GC, that would be an example of a more robust body of evidence for a coherent mechanism. As it is, since endolysosomal defects may be downstream from loss of Tfam, one can equally posit that the problem really is one in which the pre-GC (and GC) B cells do not participate adequately in cognate helper interactions, i.e., fail to present peptides as efficiently to Tfh cells.

We argue that our work clearly exceeds the reviewer’s threshold for importance, even before the revision experiments we have performed. We provide detailed and novel mechanistic insight into mitochondrial function in GC B cells, and clearly build on recent work by e.g. Weisel et al. 2020, Chen et al. 2021, and Luo et al. 2022 (1,15,16) published in Nature Immunology, addressing previous limitations in approach and mechanism. In brief, Weisel demonstrated the reliance of GC B cells on OxPhos, Chen

validated this via an orthogonal but non-mechanistic approach, and we provide a detailed account of the dynamics of mitochondria and introduce mitochondrial translation as a gatekeeper for GC-derived lymphoma. Luo et al recently show that SREBP signalling is important to regulate fatty acid metabolism in GC B cells, a process intimately associated with OxPhos in mitochondria. We use many highly innovative experimental approaches which we believe more generally advance the field of immunometabolism.

The reviewer's point about T_{FH} cell interaction is very valid and we have more extensively addressed this question in new experimental work presented in revised Fig. 5 and Extended Data Fig 5. We first co-cultured sorted ex vivo T_{FH} cells with TAT-Cre-treated Rosa26^{STOP}tdTomato × *Tfam* flox or wild type control B cells(17). We noted a reduction in tdTomato⁺GL-7⁺IgG1⁺ induced GC B cells following *Tfam* deletion. However, following immunisation, we did not detect a numerical defect in the T_{FH} compartment, and the capacity of iGB cells to process antigen and form conjugates with T cells was intact. We have also used our competitive mixed bone marrow chimera data (in which T_{FH} generation would be normal) to argue that the magnitude of effect was equivalent to that seen in a non-competitive setting, and therefore whilst defective T cell interaction has been demonstrated, the overall phenotype is principally cell-intrinsic. The new data is shown below:

The reviewer's suggestion to attempt rescue of the migration phenotype is an excellent one. We have examined migration over 5h in response to CXCL12 in ex vivo B cells from B-*Tfam* and B-WT mice, in the presence of the mitochondrial ROS scavenger mitoTEMPO, or the selective MCU inhibitor Ru265. Strikingly, mitoTEMPO but not Ru265 rescued the migration defect seen in B-*Tfam* B cells. The data is shown below and has been added to revised (new) Fig. 7.

Reviewer #3

(Remarks to the Author)

In this study, Yazicioglu and colleagues showed that the mitochondrial mass and biogenesis was increased in GC B cells relative to naïve B cells. In addition, the genetic deletion of Tfam, an essential gene supporting mitochondrial biogenesis and function, impaired B cell development and GC formation. However, the apoptosis and mitosis of Tfam-deficient GC B cells appeared to be intact in their models. They then found that the frequencies of so-called activated precursors of GC B cells were increased in the Tfam-deficient mice. They also showed that the deletion of Tfam disrupted GC spatial organization, impaired cell mobility, as well as suppressed the development of lymphoma in a c-Myc transgenic model.

In summary, the authors successfully demonstrated a critical role of Tfam gene in B cell development and GC reactions. However, considering that GC B cells are known to be mainly fueled by oxidative phosphorylation (OxPhos) according to recent publications (Weisel et al. Nat Imm 2020; Chen et al. Nat Imm 2021), as well as that Tfam is a such fundamentally important gene for mitochondrial biogenesis, most findings related to the perturbation of Tfam in B cells are fully expected. Therefore, the overall novelty of this study is limited. The proposed mechanism of Tfam-mediated entry of activated GC-precursor B cells is interesting and the most novel aspect of this study, but which haven't been substantiated via direct evidence.

We thank the reviewer for their positive comments. We would respectfully direct reviewer 3 to our previous comments on these recent publications, and to the opinion of reviewer 2 on their limitations and the subsequent novelty of our work in this context.

Specific critiques:

(1) Introduction p3: I think this misrepresents the state of the field: "Despite evidence of a central role for mitochondria, their dynamics, function, and regulation in GC B cell biology remains largely unknown." This statement here is in order for the authors to make the claim that the present study is valuable. However, I think the previous publications (Weisel et al. Nat Imm 2020; Chen et al. Nat Imm 2021) already provided quite an amount of evidence showing the dynamic and function of mitochondria in GC B cells. This manuscript should present a more accurate overview of the literature in positioning the present work.

We have expanded our introduction section by providing more discussion of the highlighted papers including Urbanczyk et al 2022 as requested by the other

reviewers. However, we still respectfully disagree with the reviewer on this sentence “I think the previous publications (Weisel et al. Nat Imm 2020; Chen et al. Nat Imm 2021) already provided quite an amount of evidence showing the dynamic and function of mitochondria in GC B cells.”, as outlined above.

(2) Regarding to the Tfam-mediated entry of activated GC-precursor B cells, the authors provided several evidence based on the imaging of actin cytoskeleton and GC organization and in vitro trans-well assay that are correlated with this argument. It's a challenge to distinguish recently generated GC B cells (GC-precursors) versus previously generated GC B cells by flow cytometry analysis during a polyclonal immune response in WT mice. The author used IgD⁺ as a marker is not very convincing. To directly demonstrated this claim, the author should employ adoptive cell co-transfer model, e.g. by co-transferring naïve WT and Tfam-ko BCR-transgenic B cells into WT mice followed by the monitoring of the dynamics at different time points, to demonstrate that Tfam deficiency B cells are stuck in the activated or early GC stage compared with WT B cells. Then It would be ideal to further demonstrate that once GC B cells bypass the entry phase, the deletion of Tfam will not affect GC response by co-transferring mature WT and KO GC B cells, or using inducible Tfam KO system.

Our discrimination of activated GC precursors is based on the use of a tdTomato reporter (Rosa26^{STOP}tdTomato Ai9 allele) activated by expression of Aicda (using Aicda-Cre). This reporter system is widely used in GC biology for lineage tracking and is expressed at an early stage of B cell activation and GC commitment. The expansion of an activated precursor phenotype in Aicda-Tfam mice was also demonstrated using single cell RNAseq. We did not use IgD alone as this is insufficient as the reviewer suggests, but in non-labelled B-Tfam mice also used GL-7 binding. To further confirm that the AP cell compartment was induced by our immunisation challenge, we demonstrated the precursor state and antigen specificity of IgD⁺ tdTomato⁺ AP cells in Extended Data Fig. 5C, by showing their NP-binding capacity.

We are reluctant to widely employ a BCR transgenic system for the reasons thoroughly outlined by Boothby et al(18) in response to the paper by Weisel et al 2020. Moreover we do not consider that its use will provide additional insight in the manner suggested by the reviewer.

Whilst the transfer of mature GC B cells or the use of an inducible-Tfam KO system are interesting suggestions, we are not aware that transfer of mature GC B cells is a viable strategy, and we are also unsure as to how an inducible system would allow us to readily address the question of failure of APs to enter the GC reaction.

(3) It's not clear to me that whether the authors try to claim that the comprised actin cytoskeleton and chemotaxis (Fig. 6) contribute to the impaired GC B cell commitment (Fig. 5), which finally led to the reduced GC reactions (Fig. 3)? If so, in Fig. 6A-E, mature GC B cells, instead of activated GC-precursor B cells described in Fig. 5, manifested the defect in actin cytoskeleton and chemotaxis. If not, then which one is the major mechanism responsible for the compromised GC reactions? Thus, the logic and contents between Fig.5 (GC B cell commitment) and Fig. 6 (actin cytoskeleton and other cellular phenotypes) currently is incoherent. A related question is that does Tfam-deficient precursors manifest any defect in terms of cell cycling or viability?

We would emphasise that given the multifunctional role of mitochondria, a single mechanism underlying the phenotype of Tfam deletion may not represent the

underlying biology, and our work is in keeping with this. GCs still do form in *Aicda-Tfam* mice, but their output and dynamics remain defective. We also highlight that our results demonstrate additional defects in *Tfam*^{-/-} B cells which are present even following GC entry. For example, the LZ/DZ ratio is skewed, there is a reduced GC:plasma cell ratio, and abnormal SHM even in the GC B cell cluster. Whether APs have defective cell cycling or higher rates of apoptosis is a good question. We examined AP B cells from immunised *Aicda*-WT and *Aicda-Tfam* mice (as per Extended Data Fig. 3L), and while there was a trend towards impaired EdU incorporation, it did not reach statistical significance (please see above). Moreover, the rate of EdU positivity was, as expected, considerably lower compared with GC B cells, which are highly proliferative. There was no difference in the rate of caspase-3 positivity. The data are shown below.

(4) The authors found that the deficiency of *Tfam* reduced the numbers of early developing B cells, GC B cells and lymphoma, all of which are highly proliferative. They proposed that the impaired cell fate commitment (and/or cell migration) led to the compromised GC reactions. The key question is that does the similar mechanism, instead of the defect of cell proliferation, contribute to the deficiency of the early developing B cells and lymphoma? Also, whether the disrupted B cell development contribute to the suppression of lymphoma development needs to be addressed. Dissecting and reconciling the mechanism by *Tfam* (and mitochondria) in these three biological contexts would add much value to the manuscript and would be of interest to the B cell community.

TFAM deficiency completely prevented the development of c-Myc-driven lymphoma, which limited our ability to draw comparisons with its role in normal GC physiology. We have therefore performed experimental work in the human Daudi GC-derived lymphoma line, using pharmacologic modulation of mitochondrial transcription and translation to reproduce our findings and demonstrate conserved mechanisms (e.g. defective actin cytoskeleton and mtROS).

The question of whether disrupted B cell development is responsible for protection from lymphoma in *Eμ-Myc-Tfam* mice is one we bring up in the discussion. Whilst as we discuss, the B-cell developmental defect likely does have a role, in a similar model in which the *Eμ-Myc* transgene is present in association with an allele which leads to severe B cell lymphopenia, the reduction in lymphoma development is much less than we observe(19).

However to address this question experimentally, we have deleted *Tfam* from established spontaneous lymphoma cells from Eμ-Myc-Tfam flox or WT mice (which are Cre negative) using TAT-Cre ex vivo (using an experimental design similar to that described in Extended Data Fig. 3M) and then adoptively transferred them into wildtype CD45.1 recipients. Mice which had received Tfam-deficient lymphoma cells had less lymphadenopathy at analysis after 3 weeks. The data is shown below.

Whilst we agree with the reviewer that the effect of *Tfam* deletion on B cell development is interesting, and we observe similar dynamic expression of ETC complex proteins which appear to be programmed by TFAM expression, we would regard it as largely outside the scope of our current manuscript which is focused on the GC. We have now added additional experimental work characterising the dynamics of ETC protein expression during B cell developmental trajectories (added to Fig. 2).

(5) The knockout efficiency of *Aicda*-cre needs to be carefully confirmed using antibody staining. Do the remaining CD38⁺ GL-7⁻ cells (Fig. 3D, 3E, 3G) indeed not express TFAM in *Aicda*-*Tfam* mice? It's possible that the expression of Cre in newly generated GC B cells was not sufficient to efficiently delete flox alleles in the *Tfam* locus. Also, how about the proliferating state of GC B cells in B-*Tfam* mice (Fig. 1F)?

The reviewer's point about TFAM is clearly important and we have now extensively characterised TFAM expression across B cell subsets, including AP B cells (new results are included in revised Fig. 2, revised Extended Data Fig. 5, and a new Fig. 6). Fig. 3E-G do not include GL-7 staining, so we assume the reviewer refers to the tdTomato⁻ populations. These results confirm efficient and enduring loss of TFAM protein in B-*Tfam* AP and GC B cells. We have also confirmed lack of expression of TFAM in *Aicda*-*Tfam* AP and GC B cells. The data is shown below. We appreciate the reviewer's question about the proliferative state of B-*Tfam* GC B cells. We anticipate that this will mirror that observed in *Aicda*-*Tfam* GC B cells, as we have been unable to identify significant phenotypic divergence.

(6) The authors need to improve the consistency and quality of the data throughout the manuscript. For example, 1) the increased numbers of TFAM-nucleoid in GC B cells relative to naïve B cells as showed in Fig. 1H are not overserved in Fig. 1G.

We respectfully disagree here. If examined closely, Tfam nucleoid numbers depicted in this single 2D Z plane of 3D images were indeed higher in GC B cells compared to naïve B cells (9 nucleoids vs 12 nucleoids). However, mitochondrial nucleoids are 3D structures, and so we quantified nucleoid counts based on 3D segmented images and included 3D presentation of the segmented mitochondrial nucleoid network in Fig 1H.

2) The distribution and frequencies of pre-proB, pro-B, pre-B and immature B cells of B-WT mice illustrated in Fig. S2B are very different compared to that in Fig. 2B,2C. Therefore, the author claimed that B-Tfam het mice had normal B cell development is skeptical. 3) The frequencies of FO and MZ B cells in B220+ cells in B-Tfam mice were roughly 40% and 6%, respectively, as illustrated in Fig. 2A. What're the other ~50% of B220+ cells then? Representative flow plots showing the gating of MZ and FO B cells should help.

We thank the reviewer for highlighting this. It came to our attention that in Fig S2B pro-B cell percentages were accidentally duplicated for pre-B cells, creating a discrepancy between the quantification and flow cytometry plots. We apologise for this mistake, and have corrected it accordingly in the revised version. We have provided flow cytometry plots and gating strategies as requested for this figure (Extended Data Fig. 2, and please see above). These original percentages were based on pre-gating for IgM or IgD expression, and we have simplified that gating in the revised manuscript. The percentages of Fo and MZ B cells are therefore now more typical.

(7) Are B cell development normal in Aicda-Tfam mice? The spleen of Aicda-Tfam mice appear to be obviously smaller than the controls in Fig. 3B. There are also publications demonstrating AID expression during B cell development.

The reviewer is correct to highlight the risk of leakage from Aicda-Cre. Since we include a tdTomato reporter allele, any cells which have expressed Aicda are labelled. We detected negligible numbers of tdTomato+ B cells in unimmunised mice. There is no defect in development of follicular B cells in Aicda-Tfam mice. The apparent difference in spleen size in 3B is reflective of a failure of GC formation in Aicda-Tfam mice, but there is no difference in B cell follicle size (unlike B-Tfam mice).

(8) Fig. S3E: Why not directly use B cells from B-Tfam or Aicda-Tfam mice to minimize the additional step of introducing TAT-cre? Also, the total numbers of B cells after 4 days culture should be presented.

We used TAT-Cre to allow precise temporal control of *Tfam* deletion, and to achieve more efficient in vitro deletion than we observed with endogenous expression of *Aicda-Cre* (as upregulation of AICDA is late and limited in vitro)(20). We have provided the total number of iGB cells at the end of the culture period in Extended Data Fig. 5, as this in vitro system is most extensively used. The data is shown below.

(9) Page 4: “(figure S3F-G)” should be “(figure S3E-G)”.

Apologies, we have corrected this in the revised version.

(10) Fig. 5C legend: remove “and Aicda-Tfam”

This has also been corrected in the revised version.

1. Weisel FJ, Mullett SJ, Elsner RA, Menk AV, Trivedi N, Luo W, et al. Germinal center B cells selectively oxidize fatty acids for energy while conducting minimal glycolysis. *Nat Immunol.* 2020 Mar;21(3):331–42.
2. Biram A, Liu J, Hezroni H, Davidzohn N, Schmiedel D, Khatib-Massalha E, et al. Bacterial infection disrupts established germinal center reactions through monocyte recruitment and impaired metabolic adaptation. *Immunity.* 2022 Feb;S1074761322000425.
3. Argüello RJ, Combes AJ, Char R, Gigan JP, Baaziz AI, Bousiquot E, et al. SCENITH: A Flow Cytometry-Based Method to Functionally Profile Energy Metabolism with Single-Cell Resolution. *Cell Metab.* 2020 Dec 1;32(6):1063-1075.e7.
4. Lopes N, McIntyre C, Martin S, Raverdeau M, Sumaria N, Kohlgruber AC, et al. Distinct metabolic programs established in the thymus control effector functions of $\gamma\delta$ T cell subsets in tumor microenvironments. *Nat Immunol.* 2021 Feb;22(2):179–92.
5. Zaheen A, Boulianne B, Parsa JY, Ramachandran S, Gommerman JL, Martin A. AID constrains germinal center size by rendering B cells susceptible to apoptosis. *Blood.* 2009 Jul 16;114(3):547–54.

6. Toboso-Navasa A, Gunawan A, Morlino G, Nakagawa R, Taddei A, Damry D, et al. Restriction of memory B cell differentiation at the germinal center B cell positive selection stage. *J Exp Med*. 2020 May 14;217(7):e20191933.
7. Nojima T, Haniuda K, Moutai T, Matsudaira M, Mizokawa S, Shiratori I, et al. In-vitro derived germinal centre B cells differentially generate memory B or plasma cells in vivo. *Nat Commun*. 2011 Aug 30;2:465–11.
8. Haniuda K, Fukao S, Kitamura D. Metabolic Reprogramming Induces Germinal Center B Cell Differentiation through Bcl6 Locus Remodeling. *Cell Rep*. 2020 Nov 3;33(5):108333.
9. Pirgova G, Chauveau A, MacLean AJ, Cyster JG, Arnon TI. Marginal zone SIGN-R1+ macrophages are essential for the maturation of germinal center B cells in the spleen. *Proc Natl Acad Sci*. 2020 Jun 2;117(22):12295–305.
10. Morén C, Garrabou G, Noguera-Julian A, Rovira N, Catalán M, Hernández S, et al. Study of oxidative, enzymatic mitochondrial respiratory chain function and apoptosis in perinatally HIV-infected pediatric patients. *Drug Chem Toxicol*. 2013 Oct 1;36(4):496–500.
11. Walker UA, Setzer B, Venhoff N. Increased long-term mitochondrial toxicity in combinations of nucleoside analogue reverse-transcriptase inhibitors. *AIDS*. 2002 Nov 8;16(16):2165.
12. Weisberger AS, Daniel TM. Suppression of Antibody Synthesis by Chloramphenicol Analogs. *Proc Soc Exp Biol Med*. 1969 Jun 1;131(2):570–5.
13. Ohnishi S, Murata M, Ida N, Oikawa S, Kawanishi S. Oxidative DNA damage induced by metabolites of chloramphenicol, an antibiotic drug. *Free Radic Res*. 2015 Sep 2;49(9):1165–72.
14. Almeida L, Dhillon-LaBrooy A, Castro CN, Adossa N, Carriche GM, Guderian M, et al. Ribosome-Targeting Antibiotics Impair T Cell Effector Function and Ameliorate Autoimmunity by Blocking Mitochondrial Protein Synthesis. *Immunity*. 2021 Jan;54(1):68-83.e6.
15. Luo W, Adamska JZ, Li C, Verma R, Liu Q, Hagan T, et al. SREBP signaling is essential for effective B cell responses. *Nat Immunol*. 2022 Dec 28;1–12.
16. Chen D, Wang Y, Manakkat Vijay GK, Fu S, Nash CW, Xu D, et al. Coupled analysis of transcriptome and BCR mutations reveals role of OXPHOS in affinity maturation. *Nat Immunol*. 2021 May 24;1–10.
17. Sage PT, Ron-Harel N, Juneja VR, Sen DR, Maleri S, Sungnak W, et al. Suppression by TFR cells leads to durable and selective inhibition of B cell effector function. *Nat Immunol*. 2016 Dec;17(12):1436–46.
18. Boothby MR, Raybuck A, Cho SH, Stengel KR, Haase VH, Hiebert S, et al. Over-Generalizing About GC (Hypoxia): Pitfalls of Limiting Breadth of Experimental Systems and Analyses in Framing Informatics Conclusions. *Front Immunol*. 2021 May 10;12:664249.

19. Nguyen HV, Vandenberg CJ, Ng AP, Robati MR, Anstee NS, Rimes J, et al. Development and survival of MYC-driven lymphomas require the MYC antagonist MNT to curb MYC-induced apoptosis. *Blood*. 2020 Mar 26;135(13):1019–31.
20. Dominguez-Sola D, Kung J, Holmes AB, Wells VA, Mo T, Basso K, et al. The FOXO1 Transcription Factor Instructs the Germinal Center Dark Zone Program. *Immunity*. 2015 Dec 15;43(6):1064–74.

Decision Letter, first revision:

21st Feb 2023

Dear Dr Clarke,

Your revised manuscript entitled, "Dynamic mitochondrial transcription and translation in B cells control germinal centre entry and lymphomagenesis" has now been seen by 3 referees. Both referees #1 and #2 are endorsing publication of the study, pending textual clarifications and improvements to the data presentation. We are willing to override the overall recommendation of the more negative referee #3 and their suggestion for additional experimental support to address their point #1 - this comment can be addressed by noting the limitations of the study in the Discussion section.

We therefore invite you to revise your manuscript taking into account all reviewer and editor comments. Please highlight all changes in the manuscript text file in Microsoft Word format.

* If you have not done so already please begin to revise your manuscript so that it conforms to our Article format instructions at <http://www.nature.com/ni/authors/index.html>. Refer also to any guidelines provided in this letter.

* Please include a revised version of any required reporting checklist. It will be available to referees to aid in their evaluation of the manuscript goes back for peer review. They are available here:

Reporting summary:

When submitting the revised version of your manuscript, please pay close attention to our [href="https://www.nature.com/nature-portfolio/editorial-policies/image-integrity">Digital Image Integrity Guidelines. and to the following points below:](https://www.nature.com/nature-portfolio/editorial-policies/image-integrity)

Finally, please ensure that you retain unprocessed data and metadata files after publication, ideally archiving data in perpetuity, as these may be requested during the peer review and production

process or after publication if any issues arise.

[REDACTED]

We hope to receive your revised manuscript within four weeks. If you cannot send it within this time, please let us know. We will be happy to consider your revision so long as nothing similar has been accepted for publication at Nature Immunology or published elsewhere.

Nature Immunology is committed to improving transparency in authorship. As part of our efforts in this direction, we are now requesting that all authors identified as 'corresponding author' on published papers create and link their Open Researcher and Contributor Identifier (ORCID) with their account on the Manuscript Tracking System (MTS), prior to acceptance. ORCID helps the scientific community achieve unambiguous attribution of all scholarly contributions. You can create and link your ORCID from the home page of the MTS by clicking on 'Modify my Springer Nature account'. For more information please visit www.springernature.com/orcid.

Kind regards,

Laurie

Laurie A. Dempsey, Ph.D.
Senior Editor
Nature Immunology
l.dempsey@us.nature.com
ORCID: 0000-0002-3304-796X

Reviewers' Comments:

Reviewer #1:

Remarks to the Author:

The authors have added a substantial amount of experimental results and answered my questions. I believe the study significantly increases our knowledge on B cell responses in the germinal center and highlights important roles mitochondria play beyond their role in bioenergetics.

My only request would be to remove the comment on B cell development when discussing the TWINKLE mice, since in that paper CD23cre was used to introduce the mutation, hence the effect of the TWINKLE mutation on B cell development cannot be assessed in the chosen model.

Reviewer #2:

Remarks to the Author:

The reasons for importance and interest value of the original manuscript (July 2022) remain in force. That version elicited relatively favorable reactions from two referees (#1 somewhat more positive even than this reader), and skepticism as to novelty and other issues from referee #3. The handling Editor had a laudable and very specific précis of what was needed. Here, the authors have submitted a revised and updated manuscript, along with an extensive Response to Reviews that included the referees' comments but did not include the editorial guidance. The revisions include substantial new data and even some rather creative ways to try to get at very difficult challenges. These changes tilt the balance in a way rather favorable to the value and impact of the work.

This reader's scorecard tallied new experiments and contributory data that dealt with 9 out of 10 of the Editor's guidance points. The recommendation of using inhibitors of mitochondrial transcription or translation on MD4 B cells prior to transfer and scoring for inclusion into GC did not generate a change in the manuscript, but was a long shot in light of the time scales in question. Overall, the responses to the reviews were comprehensive and reasonable. The time scale for revision (6 months) precludes using the genetic approach mooted by Reviewer #3 in point (2) of that referee critique.

Without limiting the matter to the revisions, three points that strike me as stand-out positives for the work are:

1. the basic finding that two major impacts of Tfam are during B cell development in the marrow, and at the pre-GC "activated precursor" stage (on this, see a comment below);
2. direct evidence that this perturbation of mitochondrial function reduced the rate of somatic mutation in the VH1-72 cluster among GC-phenotype B cells (Fig 4M);
3. the novel assay (Fig 6D-G) and the heroic finding – however scant the 'statistically significant difference' may be – of reduced mitochondrial OPP (translation) in Tfam-deficient "GC" B cells (i.e., those that expressed enough AID long enough to turn on the LSL RFP), i.e. Fig 6G.

There are other positives to the work. Although it ends up a sprawling hodge-podge with inclusion of the B lymphoma model on top of the B cell ontogeny and then a GC-centric theme, the new data on the lymphoma theme are contributory.

Although my view is mostly favorable, a few comments pertain as examples of where the work falls below the very best standards although (sad to say) having flaws commonly found in this journal and its ilk, or where, in future work, it is hoped that the authors do better.

1. Fig 6C – It is regrettable that these experiments used the B-Tfam (so that developmental history, and adaptation to survival as a naïve B cell affect the results) rather than Aicda-Cre-driven deletion and cytometric analyses. This is not a "deal-breaker" but it needs to be clear in the text as a limitation.

- 2a. The presentation or arrangement of various flow cytometry results that are overlays of single

parameter histograms is frustrating and likely to irk other readers or undermine confidence. Salient examples are Fig 6E, 7D, and Extended Data 6B. [there may be others with this same problem.] Specifically, the comparison on which an inference is made is that between Tfam-deficient experimental cells (say, GCB) and the wild-type counterparts (controls). For such comparisons, the log-scale of fluorescent signal on the x axis should be clearly visible AND the two squares displaying histograms should be stacked over one another, not next to each other. With proper alignment and display of the scale, then one can assess the actual relationship between signals for WT versus knockout (or whatever is the experimental manipulation). The preferred idea or style is to display things in a way where the reader does not have to take it on faith.

2b. As to Figure 6B, while papers Nature Immunology (and similar journals) provide ample precedent for a similarly frustrating insufficiency of showing all controls, and the data here are acceptable by that standard, it is unfortunate that that the OPP-AF647 signals for WT and Tfam-deficient iGB cells that were treated only with vehicle are not shown, only "H" or "H + CHL". It is understood what the core point the authors want to convey is, but key comparators simply make the work better.

2c. Another aspect of presentation that merits fixing is that the font sizes in many figure panels' labeling are impossibly small even when space suffices to give the reader a chance. [Once in print, they may be too pixilated when magnified, and otherwise illegible. One could ask, "Why bother?" but the standard is to have it viewable by the reader.

3. There are some textual points that, for suitable rigor or clarity, should be tightened up with minor edits.

a. Section 5, late in 4th paragraph – better as "Five days thereafter, however, Tfam $-/-$ iGB cells were at a substantial competitive disadvantage in GC participation", and, in the concluding sentence (just before section 6)

b. "Overall these data suggest that Tfam promotes the entry and / or maintenance of activated B cells into the GC, [..]"

c. Ideally, the authors should note the limitations of the iGB as avatars for GC B cells.

Because of the challenges working with actual GC B cells the field – for better and worse - accepts the so-called iGB cultures as avatars even though it is well recognized that there are many differences in how losses of function affect the in vitro-derived cells compared to GC B. Indeed, the data of Fig 5 and Extended Data Fig 5 underscore the disparity (i.e., production of iGB cells during the 4 d culture).

Reviewer #3:

Remarks to the Author:

The authors have carefully addressed the reviewers' concerns with new analyses (by employing several innovative experimental approaches), clarifications and text modifications. I have several comments remaining:

1. (Related to the original point #2) The identification of Tfam-regulated AP for GC entry is the most novel aspect of this work. However, the importance/contribution of AP to the defect of total GC reactions in *Aicda*-Tfam mice still has not been directly demonstrated. The authors tried to address this point by utilizing iGC B cell transfer model to show that those in vitro activated Tfam $-/-$ B cells manifested competitive disadvantage in GC participation following the adoptive transfer (Fig. 5C-F). It's not clear to what extent iGC B cells (IgD $-$ CD38 $-$ GL-7 $+$) could mimic AP (IgD $+$ CD38 $+$ tdTomato $+$ (and/or NP-binding) B cells as defined by the authors) for in vivo GC

commitment. Could the authors provide evidence to show iGC B cells are more similar to AP than mature in vivo GC B cells, e.g. at the transcriptional or metabolic level? What's more, I'm not aware of evidence that clearly demonstrate the fate of these AP. The expression of naïve B cell markers, together with less mutated BCRs and weak antigen-binding capacity may also indicate that these cells have been or will be out-competed by other GC B cells during the selection (meaning they're "losers" instead of precursors?).

2. (Related to the original point #3) The authors discovered that Tfam regulates mitochondrial translation and metabolic process (Fig. 6), as well as cell mobility (Fig. 7) of B cells. Would the authors further clarify/discuss whether these mechanisms execute in parallel, and equally contribute to the Tfam-mediated GC commitment?

3. statistical significance for each of proteins in Figs. 2D and 2E should be indicated; representative flow cytometry plots for Figs. 5J and 5J should be provided.

Author Rebuttal, first revision:

Reviewers' Comments:

Reviewer #1:

Remarks to the Author:

The authors have added a substantial amount of experimental results and answered my questions. I believe the study significantly increases our knowledge on B cell responses in the germinal center and highlights important roles mitochondria play beyond their role in bioenergetics.

We are delighted to hear the reviewer's positive comments on our revised manuscript and thank them for their helpful suggestions throughout the peer review.

My only request would be to remove the comment on B cell development when discussing the TWINKLE mice, since in that paper CD23cre was used to introduce the mutation, hence the effect of the TWINKLE mutation on B cell development cannot be assessed in the chosen model.

We agree with the reviewer and thank them for bringing this to our attention. We have removed the highlighted comment as requested.

Reviewer #2:

Remarks to the Author:

The reasons for importance and interest value of the original manuscript (July 2022) remain in force. That version elicited relatively favorable reactions from two referees (#1 somewhat more positive even than this reader), and skepticism as to novelty and other issues from referee #3. The handling Editor had a laudable and very specific précis of what was needed. Here, the authors have submitted a revised and updated manuscript, along with an extensive Response to Reviews that included the referees' comments but did not include the editorial guidance. The revisions include substantial new data and even some rather creative ways to try to get at very difficult challenges. These changes tilt the balance in a way rather favorable to the value and

impact of the work.

This reader's scorecard tallied new experiments and contributory data that dealt with 9 out of 10 of the Editor's guidance points. The recommendation of using inhibitors of mitochondrial transcription or translation on MD4 B cells prior to transfer and scoring for inclusion into GC did not generate a change in the manuscript, but was a long shot in light of the time scales in question. Overall, the responses to the reviews were comprehensive and reasonable. The time scale for revision (6 months) precludes using the genetic approach mooted by Reviewer #3 in point (2) of that referee critique.

We would like to express our gratitude to the reviewer for their positive comments on the comprehensiveness of the updated manuscript. We are also appreciative of their constructive input and suggestions, which have enabled us to enhance the scientific quality and significance of our work.

Without limiting the matter to the revisions, three points that strike me as stand-out positives for the work are:

1. the basic finding that two major impacts of Tfam are during B cell development in the marrow, and at the pre-GC "activated precursor" stage (on this, see a comment below);
2. direct evidence that this perturbation of mitochondrial function reduced the rate of somatic mutation in the VH1-72 cluster among GC-phenotype B cells (Fig 4M);
3. the novel assay (Fig 6D-G) and the heroic finding – however scant the 'statistically significant difference' may be – of reduced mitochondrial OPP (translation) in Tfam-deficient "GC" B cells (i.e., those that expressed enough AID long enough to turn on the LSL RFP), i.e. Fig 6G.

There are other positives to the work. Although it ends up a sprawling hodge-podge with inclusion of the B lymphoma model on top of the B cell ontogeny and then a GC-centric theme, the new data on the lymphoma theme are contributory.

Although my view is mostly favorable, a few comments pertain as examples of where the work falls below the very best standards although (sad to say) having flaws commonly found in this journal and its ilk, or where, in future work, it is hoped that the authors do better.

1. Fig 6C – It is regrettable that these experiments used the B-Tfam (so that developmental history, and adaptation to survival as a naïve B cell affect the results) rather than Aicda-Cre-driven deletion and cytometric analyses. This is not a "deal-breaker" but it needs to be clear in the text as a limitation.

We have added the following section to the discussion:

'Temporal control of Tfam deletion either very early in B cell development using Cd79a-Cre, or following activation with Aicda-Cre or in vitro with TAT-Cre did not lead to unexpected or major phenotypic differences, but it remains possible that adaptations might occur in a dynamic manner.'

2a. The presentation or arrangement of various flow cytometry results that are overlays of single parameter histograms is frustrating and likely to irk other readers or undermine confidence. Salient examples are Fig 6E, 7D, and Extended Data 6B. [there may be others with this same problem.] Specifically, the comparison on which an inference is made is that between Tfam-deficient experimental cells (say, GCB) and the wild-type counterparts (controls). For such comparisons, the log-scale of fluorescent signal on the x axis should be clearly visible AND the two squares displaying histograms should be stacked over one another, not next to each other. With proper alignment and display of the scale, then one can assess the actual relationship between signals for WT versus knockout (or whatever is the experimental manipulation). The preferred idea or style is to display things in a way where the reader does not have to take it on faith.

We have implemented the changes requested by the reviewer and rearranged the display of the relevant flow histogram plots accordingly in the manuscript. We have also made x-axis values available in all flow histogram overlay panels to facilitate reliable visual comparison between conditions.

2b. As to Figure 6B, while papers Nature Immunology (and similar journals) provide ample precedent for a similarly frustrating insufficiency of showing all controls, and the data here are acceptable by that standard, it is unfortunate that that the OPP-AF647 signals for WT and Tfam-deficient iGB cells that were treated only with vehicle are not shown, only “H” or “H + CHL”. It is understood what the core point the authors want to convey is, but key comparators simply make the work better.

We believe that the reviewer's comment relates to Extended Figure 6B, and we have discovered that the control vehicle conditions were inadvertently left out of the figure panels which has now been corrected and for which we apologise. The use of harringtonine is necessary to block cytoplasmic ribosomal protein synthesis in our assay, and so is therefore not an experimental condition but rather a baseline. We have clarified this in the figure legend which was previously misleading and for which we also apologise.

2c. Another aspect of presentation that merits fixing is that the font sizes in many figure panels' labeling are impossibly small even when space suffices to give the reader a chance. [Once in print, they may be too pixilated when magnified, and otherwise illegible. One could ask, “Why bother?” but the standard is to have it viewable by the reader.

We thank the reviewer for bringing this issue to our attention and are in agreement. This might relate to scaling issues following reviewer PDF conversion, and if not we would correct this at the production stage following editorial direction.

3. There are some textual points that, for suitable rigor or clarity, should be tightened up with minor edits.

- a. Section 5, late in 4th paragraph – better as “Five days thereafter, however, Tfam $-/-$ iGB cells were at a substantial competitive disadvantage in GC participation”, and, in the concluding sentence (just before section 6)
- b. “Overall these data suggest that Tfam promotes the entry and / or maintenance of activated B cells into the GC, [..]”

We thank the reviewer for these suggestions and we have updated these sentences accordingly.

c. Ideally, the authors should note the limitations of the iGB as avatars for GC B cells. Because of the challenges working with actual GC B cells the field – for better and worse - accepts the so-called iGB cultures as avatars even though it is well recognized that there are many differences in how losses of function affect the in vitro-derived cells compared to GC B. Indeed, the data of Fig 5 and Extended Data Fig 5 underscore the disparity (i.e., production of iGB cells during the 4 d culture).

We agree with the reviewer, and please see our response to similar comments by reviewer 3. We have added the following text to the discussion section to recognise the limitations of the iGB system:

‘We have used the iGB system developed by Nojima et al.¹ to precisely control deletion of Tfam prior to adoptive transfer, and whilst this is an important and widely used tool to generate GC B cell-precursors in vitro which can then participate in the GC reaction, uncertainties remain about their fidelity to true APs and this would also benefit from future study.’

Reviewer #3:

Remarks to the Author:

The authors have carefully addressed the reviewers' concerns with new analyses (by employing several innovative experimental approaches), clarifications and text modifications. I have several comments remaining:

We thank the reviewer for their positive comments and interest in our revised manuscript as well as helpful suggestions during peer review.

1. (Related to the original point #2) The identification of Tfam-regulated AP for GC entry is the most novel aspect of this work. However, the importance/contribution of AP to the defect of total GC reactions in Aicda-Tfam mice still has not been directly demonstrated. The authors tried to address this point by utilizing iGC B cell transfer model to show that those in vitro activated Tfam $-/-$ B cells manifested competitive disadvantage in GC participation following the adoptive transfer (Fig. 5C-F). It's not clear to what extent iGC B cells (IgD-CD38-GL-7+) could mimic AP (IgD+CD38+tdTomato+(and/or NP-binding) B cells as defined by the authors) for in vivo GC commitment. Could the authors provide evidence to show iGC B cells are more similar to AP than mature in vivo GC B cells, e.g. at the transcriptional or metabolic level?

We recognise the reviewer's point, and have added the following text to the discussion to act as a caveat:

*'We have used the iGB system developed by Nojima et al.¹ to precisely control deletion of *Tfam* prior to adoptive transfer, and whilst this is an important and widely used tool to generate GC B cell-precursors in vitro which can then participate in the GC reaction, uncertainties remain about their fidelity to true APs and this would also benefit from future study.'*

What's more, I'm not aware of evidence that clearly demonstrate the fate of these AP. The expression of naïve B cell markers, together with less mutated BCRs and weak antigen-binding capacity may also indicate that these cells have been or will be out-competed by other GC B cells during the selection (meaning they're "losers" instead of precursors?).

This is an interesting question but one which is inherently difficult to answer. We identified APs as transcriptionally and spatially distinct from established GC B cells, and relatively abundant in both wild type mice and following *Tfam* deletion. Their BCRs were low affinity and unmutated, and they bound NP less strongly than GC B cells. In our transfer experiments, *Tfam*^{-/-} B cells were clearly outcompeted. We did not observe a difference in apoptosis rate in AP or GC B cells from *Tfam*^{-/-} mice. A distinct IgD⁺ AP/pre-GC phenotype is well established in the literature^{2,3}, and we feel that further characterisation beyond the data we present would be outside the scope of our manuscript.

2. (Related to the original point #3) The authors discovered that *Tfam* regulates mitochondrial translation and metabolic process (Fig. 6), as well as cell mobility (Fig. 7) of B cells. Would the authors further clarify/discuss whether these mechanisms execute in parallel, and equally contribute to the *Tfam*-mediated GC commitment?

This is also an interesting question, but again our answer must be largely speculative. The fundamental role of TFAM is to enable mitochondrial transcription and translation, and so the phenotype we observe following its deletion must originate here. As TFAM only regulates mitochondrially-encoded ETC proteins, we observe an imbalance at the ETC complex component level, which we show leads to abnormal ROS generation, actin cytoskeletal abnormalities, and migration defects which we can rescue by ROS scavenging. We believe our data supports this mechanism, which is likely to occur in a progressive manner.

3. statistical significance for each of proteins in Figs. 2D and 2E should be indicated; representative flow cytometry plots for Figs. 5J and S5J should be provided.

We have now performed statistical testing for pairwise comparisons between B-WT and B-*Tfam* B cell subsets in Figure 2D and 2E as requested. Please see the representative gating for Figure 5J below. The gating for Extended Data Figure 5J (bone marrow chimera) indicating GC proportions is as shown in Figure 3D.

References

1. Nojima, T. *et al.* In-vitro derived germinal centre B cells differentially generate memory B or plasma cells in vivo. *Nat Commun* **2**, 465–11 (2011).
2. Schwickert, T. A. *et al.* A dynamic T cell–limited checkpoint regulates affinity-dependent B cell entry into the germinal center. *Journal of Experimental Medicine* **208**, 1243–1252 (2011).
3. Hägglöf, T. *et al.* Continuous germinal center invasion contributes to the diversity of the immune response. *Cell* **186**, 147-161.e15 (2023).

Decision Letter, second revision:

25th Feb 2023

Dear Dr. Clarke,

Thank you for submitting your revised manuscript "Dynamic mitochondrial transcription and translation in B cells control germinal centre entry and lymphomagenesis" (NI-A34325B). I have looked over your revisions and now note that we'll be happy in principle to publish it in Nature Immunology, pending minor revisions to comply with our editorial and formatting guidelines.

We will now perform detailed checks on your paper and will send you a checklist detailing our editorial and formatting requirements in about a week. Please do not upload the final materials and make any revisions until you receive this additional information from us.

If you had not uploaded a Word file for the current version of the manuscript, we will need one before beginning the editing process; please email that to immunology@us.nature.com at your earliest convenience.

Thank you again for your interest in Nature Immunology Please do not hesitate to contact me if you have any questions.

Kind regards,

Laurie

Laurie A. Dempsey, Ph.D.
Senior Editor
Nature Immunology
l.dempsey@us.nature.com
ORCID: 0000-0002-3304-796X

Final Decision Letter:

In reply please quote: NI-A34325C

Dear Dr. Clarke,

I am delighted to accept your manuscript entitled "Dynamic mitochondrial transcription and translation in B cells control germinal centre entry and lymphomagenesis" for publication in an upcoming issue of Nature Immunology.

Over the next few weeks, your paper will be copyedited to ensure that it conforms to Nature Immunology style. Once your paper is typeset, you will receive an email with a link to choose the appropriate publishing options for your paper and our Author Services team will be in touch regarding any additional information that may be required.

Please note that Nature Immunology is a Transformative Journal (TJ). Authors may publish their research with us through the traditional subscription access route or make their paper immediately open access through payment of an article-processing charge (APC). Authors will not be required to make a final decision about access to their article until it has been accepted. Find out more about Transformative Journals.

Your paper will be published online soon after we receive your corrections and will appear in print in the next available issue. Content is published online weekly on Mondays and Thursdays, and the embargo is set at 16:00 London time (GMT)/11:00 am US Eastern time (EST) on the day of publication. Now is the time to inform your Public Relations or Press Office about your paper, as they might be interested in promoting its publication. This will allow them time to prepare an accurate and satisfactory press release. Include your manuscript tracking number (NI-A34325C) and the name of the journal, which they will need when they contact our office.

About one week before your paper is published online, we shall be distributing a press release to news organizations worldwide, which may very well include details of your work. We are happy for your institution or funding agency to prepare its own press release, but it must mention the embargo date and Nature Immunology. Our Press Office will contact you closer to the time of publication, but if you or your Press Office have any enquiries in the meantime, please contact press@nature.com.

Also, if you have any spectacular or outstanding figures or graphics associated with your manuscript - though not necessarily included with your submission - we'd be delighted to consider them as candidates for our cover. Simply send an electronic version (accompanied by a hard copy) to us with a possible cover caption enclosed.

If you have not already done so, we strongly recommend that you upload the step-by-step protocols used in this manuscript to the Protocol Exchange. Protocol Exchange is an open online resource that

allows researchers to share their detailed experimental know-how. All uploaded protocols are made freely available, assigned DOIs for ease of citation and fully searchable through nature.com. Protocols can be linked to any publications in which they are used and will be linked to from your article. You can also establish a dedicated page to collect all your lab Protocols. By uploading your Protocols to Protocol Exchange, you are enabling researchers to more readily reproduce or adapt the methodology you use, as well as increasing the visibility of your protocols and papers. Upload your Protocols at www.nature.com/protocolexchange/. Further information can be found at www.nature.com/protocolexchange/about .

Please note that we encourage the authors to self-archive their manuscript (the accepted version before copy editing) in their institutional repository, and in their funders' archives, six months after publication. Nature Portfolio recognizes the efforts of funding bodies to increase access of the research they fund, and strongly encourages authors to participate in such efforts. For information about our editorial policy, including license agreement and author copyright, please visit www.nature.com/ni/about/ed_policies/index.html

Kind regards,

Laurie

Laurie A. Dempsey, Ph.D.
Senior Editor
Nature Immunology
l.dempsey@us.nature.com
ORCID: 0000-0002-3304-796X